# Ice bridges and ridges in the Maxwell-EB sea ice rheology

Véronique Dansereau[1], Jérôme Weiss[1], Pierre Saramito[2], Philippe Lattes[3], and Edmond Coche[4]

[1]Institut des Sciences de la Terre, CNRS UMR 5275, Université de Grenoble, Grenoble, France
[2]Laboratoire Jean Kuntzmann, CNRS UMR 5224, Université de Grenoble, Grenoble, France
[3]TOTAL S.A. - DGEP/DEV/TEC/GEO, Paris, France
[4]TOTAL S.A. - DGEP/AF/CG-TEPCG/DBD/DDP/DMHB, Pointe Noire-Poincare, Congo

*Correspondence to:* Véronique Dansereau (veronique.dansereau@univ-grenoble-alpes.fr)

**Abstract.**

   This paper presents a first implementation of a new rheological model for sea ice on geophysical scales. This continuum model, called Maxwell-Elasto Brittle (Maxwell-EB), is based on an Maxwell constitutive law, a progressive damage mechanism that is coupled to both the elastic modulus and apparent viscosity of the ice cover and a Mohr-Coulomb damage criterion that allows for pure (uniaxial and biaxial) tensile strength. The model is tested on the basis of its capability to reproduce the complex mechanical and dynamical behaviour of sea ice drifting through a narrow passage. Idealized as well as realistic simulations of the flow of ice through Nares Strait are presented. These demonstrate that the model reproduces the formation of stable ice bridges as well as the stoppage of the flow, a phenomenon occurring within numerous channels of the Arctic. In agreement with observations, the model captures the propagation of damage along narrow arch-like kinematic features, the discontinuities in the velocity field across these features dividing the ice cover into floes, the strong spatial localization of the thickest, ridged ice, the presence of landfast ice in bays and fjords and the opening of polynyas downstream of the Strait. The model represents various dynamical behaviours linked to an overall weakening of the ice cover and to the shorter lifespan of ice bridges, with implications in terms of increased ice export through narrow outflow pathways of the Arctic.

## 1   Introduction

The formation of ice bridges is a common phenomenon in the Amundsen Gulf, Bering Strait as well as in many narrow passages of the Canadian Arctic Archipelago (Sodhi, 1977). Commonly referred to as ice *arches* because of their curved and concave shape, these structures can remain stable for several weeks or months and stop the flow of ice through outflow channels (Kwok, 2005; Kwok et al., 2010; Münchow, 2016). Downstream of ice bridges, expanses of ice-free water and polynyas open, which strongly impacts the local atmosphere-ocean heat exchanges and promotes the generation of new ice (Smith et al., 1990). Upon breakup of a bridge, the outflow of ice, stored in the basin upstream, drastically increases. Therefore, the capability of representing adequately the complex dynamical behaviour of ice drifting through narrow passages might constitute a key asset when using numerical models to asses the seasonal and interannual variability in the circulation and export of fresh water and sea ice in the Arctic.

Figure 1b shows an example of such an ice arch present on July 2nd, 2010, at the Lincoln Sea entrance to Nares Strait, in the Canadian Arctic Archipelago (see Fig. 1a), one of the most extensively studied outflow pathways for seasonal and multi-year Arctic sea ice (Barber et al., 2001; Kwok, 2005; Kwok et al., 2010; Münchow, 2016; Ryan and Münchow, 2017). The annual mean ice volume flux through this channel, only a few tens of kilometers wide ($30-40$ km) in some places, is thought to be equivalent to about $7\%$ of the annual mean flux through Fram Strait ($\sim 130-140$ km$^3$, Kwok, 2005; Kwok et al., 2010). Poleward of the Strait, ice converges towards the coast of Ellesmere Island and Greenland, where multiyear ice coverage is known to be high ($> 80\%$, Kwok, 2006). Convergence leads to the formation of pressure ridges and the thickness of the ice cover there reaches values among the highest encountered in the Arctic Ocean (Wadhams, 1994; Haas et al., 2006). Analyses of RADARSAT imagery have shown that the ice flux out of Nares Strait stops seasonally after the formation of stable ice bridges in mid- to late-winter, allowing the wide North Water polynya to open downstream of the Strait (Barber et al., 2001; Ingram et al., 2002), and resumes upon breakup of the bridges in summer (Kwok, 2005; Kwok et al., 2010).

In this paper, these unique flow and ice coverage conditions are used as a benchmark for testing a new rheological framework developed as an alternative to the traditional Viscous-Plastic (VP) rheology to represent accurately the deformation and drift of sea ice in continuum models at regional ($\sim 100$ km) to global ($\sim 1000$ km) scales (Dansereau et al., 2016). This framework, called Maxwell-Elasto-Brittle, combines the concepts of elastic memory, viscous-like relaxation of the internal stress and progressive damage mechanics. Highly idealized simulations have demonstrated that the Maxwell-EB model reproduces the important characteristics of sea ice deformation revealed by the analysis of available ice buoy and satellite data : anisotropy, high localization in both space and time and the associated scaling laws (Dansereau et al., 2016; Weiss and Dansereau, 2017). Here, this rheological framework is implemented on geophysical scales and in a realistic context. This work focusses on two main aspects. We aim to establish the capability of the model to represent (1) the localization of the ice deformation along arch-like features in the vicinity and within a channel as well as the formation of stable ice bridges and (2) the strong localization of the thickest ice along narrow oriented features representing pressure ridges. Based on our simulation results, we also discuss how the mechanical weakening of the ice cover estimated over the period 2002-2008 relative to the period 1979-2001 (Gimbert et al., 2012a) can be linked to a shorter lifespan of ice arches and consequently, to an increased ice export through Nares Strait.

## 2 Background

### 2.1 Ice bridges

Granular materials have been known for a long time to form concave stress-free surfaces and exhibit self-obstruction to flow under certain conditions (e.g., Richmond and Gardner, 1962; Walker, 1966). By assimilating sea ice as a 2-dimensional continuum material obeying Coulomb's failure criterion, Sodhi (1977) applied the concepts of granular models developed to describe the formation of stress-free arches in hoppers and chutes to the formation of stable ice bridges in the Bering Strait and Amundsen Gulf and obtained good agreement with ice deformation patterns as observed via satellite (Landsat) imagery.

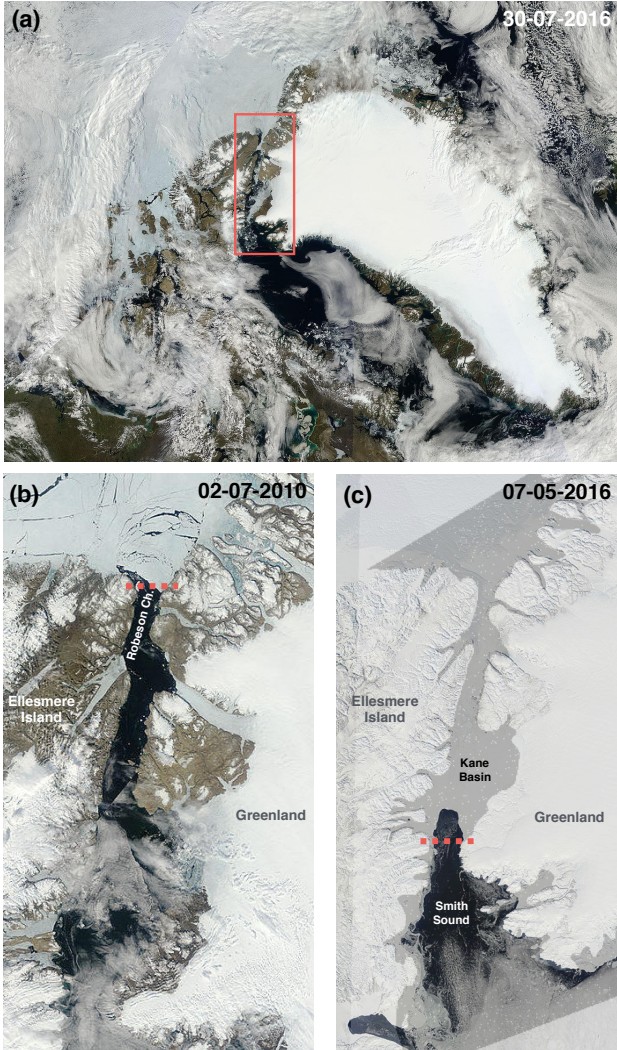

**Figure 1.** (a) Moderate Resolution Imaging Spectroradiometer (MODIS) reflectance image indicating (a) the location of Nares Strait (red rectangle), (b) the presence of an ice bridge prior to a partial breakup event, on July 2, 2010, with multiple arch-like leads upstream of Robeson Channel (red dotted line) (c) a stable ice bridge at the constriction between Kane Basin and Smith Sound, indicated by the red dotted line, with the North Water polynya open on May 7, 2016. The superimposed grey shading indicates the coverage of the domain used in the realistic Nares Strait simulations. *NASA/GSFC MODIS Rapid Response at http://rapidfire.sci.gsfc.nasa.gov/imagery/ .*

Stable ice bridges have also been successfully reproduced by plastic-type sea ice models, providing the prescribed plastic yield criterion allowed for some cohesive strength, that is, the capability to sustain uniaxial (tensile and/or compressive) stresses (e.g., Ip, 1993; Hibler et al., 2006). Recently, Dumont et al. (2009) were able to simulate the formation of ice arches in both idealized and realistic representations of Nares Strait using a dynamic Elastic-Viscous-Plastic (EVP) model (Hunke, 1997), the

ice mechanics component of which is based on the VP rheology and elliptical yield curve of Hibler (1979). This rheological framework typically does not account for uniaxial or biaxial (i.e., pure) tensile strength (see Fig. 2, stress states 0 and 1). In the case of Dumont et al. (2009), stable ice bridges and flow stoppage were obtained by decreasing the ellipticity of the yield curve below its original value (2, Hibler, 1979) to increase the shear and *uniaxial compressive strength* of the ice (see Fig. 2, stress state 2), which increases its cohesive strength. Rasmussen et al. (2010) performed numerical simulations of the sea ice dynamics in Nares Strait and the North Water Polynya using the dynamic-thermodynamic CICE sea ice model, based on the EVP rheology. The authors noted a lack of stability and shorter lifespan of the simulated ice bridges, leading to a slower opening and lower extent of the North Water polynya and to an earlier draining of Nares Strait compared to estimates from satellite imagery. They attributed this deficiency to a too low ice strength in their model, either caused by a too thin ice cover or to the inability of their rheology to reproduce the correct internal strength of sea ice.

Channel flow simulations have not yet been performed using elastic-brittle models. Hence such experiments constitute an interesting test case of their mechanical behaviour. Moreover, while other rheological models have been shown to simulate both the occurrence of ice bridges and flow stoppage, it is not at all clear if these models, even with a fine spatial resolution (e.g., 4 km in the Lincoln Sea, about 7 km at the constriction between Kane Basin and Smith Sound and 10 km in Baffin Bay in the model of Rasmussen et al. (2010) and about 3 km by 4 km in the realistic simulations of Nares Strait of Dumont et al. (2009)), are also able to account for the presence of *multiple* arch-like leads within and upstream of the channel, as observed from satellite imagery (e.g., see Fig. 1b). In coupled thermodynamic and dynamic models, a high density of leads is expected to impact the simulated heat fluxes between the atmosphere, the ice and the ocean (Smith et al., 1990). With its capability to represent the extreme localization of damage and deformation (Dansereau et al., 2016), the Maxwell-EB model might be suited to simulate these fine features.

Ice drift and coverage conditions within a channel moreover represent a severe test of the numerical scheme in terms of handling discontinuities within the simulated fields, as once a stable ice bridge forms, the ice downstream detaches from the bridge and is driven out of the channel without mechanical resistance. At this point, extremely sharp gradients in ice velocity, thickness and concentration are expected to arise (see Fig. 1b and 3a).

## 2.2 Ice ridges

Sea ice models are most often compared to each other and to observations in terms of the spatial distribution of the simulated ice thickness (e.g., Johnson et al., 2012). An equally important, and perhaps more appropriate, metric to investigate the mechanical behaviour of the sea ice cover is the probability density function (PDF) of the ice thickness, of which some valuable information have been available for some time from drill-hole, submarine-mounted sonar and airborne electromagnetic sounding measurements (Lindsay, 2003). In particular, the tail of PDFs represents the ice that has thickened, not only due to thermodynamic but also to mechanical redistribution processes. This ice is incorporated in pressure *ridges*, which are long, linear rubble piles of ice meters or tens of meters wide formed under convergent and shearing motions. A recurrent statistical property of the tail of the PDF of ice thickness is that it appears to fit a negative exponential function (Wadhams, 1994; Haas,

2009). To this day, it is not clear why it takes this particular form. The important point however is that it is the signature of the tendency of mechanical redistribution to "create extremes" (Thorndike et al., 1975). In other words, it characterizes the strong localization of the thickest ice in space.

Over the years, there have been different attempts to represent the formation of pressure ridges in numerical simulations
of the sea ice cover (e.g., Parmerter and Coon, 1972; Kovacs and Sodhi, 1980; Hopkins, 1994), but no model is yet capable of describing the entire process. Continuum sea ice models typically have a spatial resolution of a few kilometers to tens of kilometers and hence do not resolve ridges per se. In such models, two main approaches are taken to handle the redistribution of ice thickness associated with ridge building.

The so-called multi-thickness categories scheme based on the pioneering work of Thorndike et al. (1975) and Rothrock
(1975) is widely used in current sophisticated sea ice and coupled models. This scheme introduces an areal thickness distribution function which evolves in time due to both thermodynamics and dynamics processes. Ridging is treated "explicitly" by allowing a prescribed thin ice portion of the thickness distribution to be redistributed into thicker ice categories in response to the simulated deformation. The main advantage of this scheme is that it allows accounting for variations in the ice thickness at the sub-grid scale. The relation between the redistribution process and the strength of the ice (often characterized by
a pressure, $P$) in this modelling framework is based on energy conservation principles : the deformational work is equated to the work done in building ridges, which is partitioned between potential energy changes (Thorndike et al., 1975), the frictional dissipation in ridging (Rothrock, 1975) and dissipation in shearing deformation (Pritchard, 1981), all of which are very hard to estimate. This theory does not take into account other mechanisms such as crushing, buckling, flexural breakage, inelastic contacts and frictional sliding contacts between rubble ice blocks (Hopkins, 1998). As the simulated strain rate tensor does
not provide directly the information on the relative amount of opening and ridging (and also sliding) within the ice cover, expressions for the modes of redistribution need to be assumed, the correct form of which remains uncertain to this day (Hunke et al., 2010). These redistribution functions can be set in an ad-hoc manner (Thorndike et al., 1975), estimated empirically from strain rates observations (Stern et al., 1995) or, in the case of plastic models such as the VP and EVP models, determined based on the prescribed form of the yield criteria and flow rule (Rothrock, 1975; Hibler, 1980; Flato and Hibler, 1995). The multi-
categories scheme introduces several additional parameters (e.g., a frictional dissipation coefficient, a prescribed percentage of the thickness distribution participating in the ridging, ...), which are all poorly constrained and to which the simulated thickness distribution and patterns can be highly sensitive (Flato and Hibler, 1995; Bitz et al., 2001). This framework necessitates solving an additional evolution equation for the thickness distribution function as well as thermodynamics and transport equations for multiple ice thicknesses, which increases the cost of numerical schemes as the number of ice categories is increased.
The second approach is the simpler two-level model suggested by Hibler (1979) in which the simulated ice cover falls into two thickness categories: the effective thickness, representing the average ice thickness of the ice-covered portion of a model grid cell, and zero thickness, or open water. As opposed to the multi-category scheme, ridge building is treated implicitly based on a volume conservation principle. Known shortcomings of this model when coupled to a VP (or EVP) rheological framework are the underestimation of ice thickening in regions of convergent and shearing ice motion, which has been attributed to the

unresolving of thin ice that participates in ridge building. This process has been more adequately simulated at the cost of increasing the number of ice thickness categories in a multi-categories redistribution scheme (e.g., Bitz et al., 2001).

Here, we use a very simple redistribution scheme to test the capability of the Maxwell-EB rheological framework to reproduce the observed strong localization of thick ice in space.

## 3    The Maxwell-EB sea ice model

The Maxwell-EB model builds on the continuum Elasto-Brittle (EB) rheology, which has been used to model the fracturing of rocks (e.g., Amitrano et al., 1999) and was implemented for sea ice modelling by Girard et al. (2010) and Bouillon and Rampal (2015); Rampal et al. (2015). As the EB model is based on a linear-elastic constitutive law, it does not solve simultaneously for both the elastic (reversible) deformations associated with the fracturing of the ice pack and the permanent (irreversible) deformations occurring once the ice pack is fractured and ice floes move relative to each other. Therefore, it does not allow estimating ice drift velocities unambiguously. The Maxwell-EB model was developed to deal with these intrinsic shortcomings of the EB framework. This mechanical model has been described in full detail by Dansereau et al. (2016). Here we only review its essential features and present the full system of equations and its numerical treatment in Appendix A.

In this augmented rheology, a transition between the small/elastic and large/permanent deformations is made possible by the addition of a viscous-like relaxation term in the linear-elastic constitutive law for a compressible, continuous solid (Dansereau et al., 2016). Associated with the linear elastic term in this constitutive equation (Eq. (A2)) is the true elastic modulus ($E$) of the material, i.e., of sea ice, at the scale of the model grid cell. The viscosity, $\eta$, associated with the viscous term, is *not* the true bulk viscosity of sea ice, but an apparent viscosity that represents the flow resistance of the fractured/fragmented ice cover averaged over the grid cell. The ratio of the two mechanical properties, $\lambda = \frac{\eta}{E}$, has the dimension of a time, and sets the rate of dissipation of the stress through permanent viscous-like deformations. Alternatively, it quantifies the capability of the ice pack to retain the memory of elastic deformations. In the Maxwell-EB model, these three mechanical parameters vary with the local level of damage of the material, quantified by a scalar variable $d$ that evolves between 1 for an undamaged and 0 for a "completely damaged" ice cover.

Damage occurs when the state of stress becomes overcritical with respect to a Mohr-Coulomb failure criterion or a tensile cut-off. The combined criteria are represented in Fig. 2 (thick black lines) in the principal stresses space ($\sigma_1$, $\sigma_2$), with the convention that compressive stresses are positive. With such criterion, uniaxial compressive strength, $\sigma_c$, uniaxial tensile strength, $\sigma_t$, (as well as biaxial tensile strength) are accounted for and are directly related to a cohesion parameter $C$ as follows:

$$\sigma_c = \frac{2C}{(\mu^2 + 1)^{1/2} - \mu},\tag{1}$$

$$\sigma_t = -\frac{\sigma_c}{q},\tag{2}$$

where $\mu$ is the internal friction coefficient and $q = \left[(\mu^2 + 1)^{1/2} + \mu\right]^2$ is the slope of the Mohr-Coulomb failure envelope in the internal stress space [1]. In the model, the damage criterion varies spatially, as some disorder in $C$ is introduced at the local

---

[1]Note that there is an error in the expression for $\sigma_t$, Eq. 10, in Dansereau et al. (2016): $\sigma_t = -\frac{\sigma_c}{q}$, but $\sigma_t \neq -2C\left[(\mu^2 + 1)^{1/2} + \mu\right]$.

scale to represent the natural heterogeneity of the ice cover associated with the presence of various defects (e.g., brine pockets, thermal cracks) as well as of different ice types (first year versus multi-year ice), floe sizes and arrangments at the sub-grid scale. This heterogeneity ensures the progressive failure of an initially undamaged ice cover even under fully homogeneous forcing conditions (Dansereau et al., 2016).

5      A healing mechanism counterbalances the effects of damaging over much larger time scales and represents the refreezing of leads. This mechanism is distinct from pure thermodynamic growth as it allows the level of damage variable to re-increase and recover *at most* the undamaged value of $d = 1$ (see Dansereau et al. (2016), section 3.3.2). Both processes, damaging and healing, are combined in a single equation for the evolution of $d$ (Eq. (A3)).

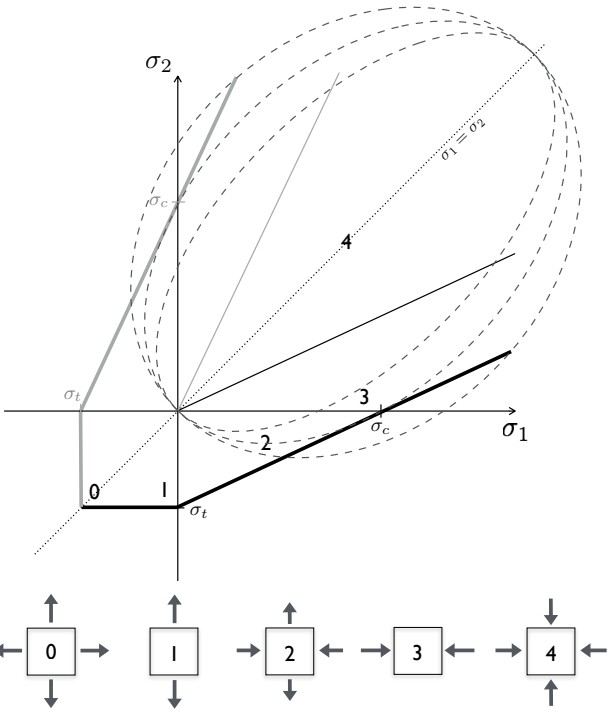

**Figure 2.** Damage criterion in the Maxwell-EB model represented symmetrically in the principal stresses plane (thick solid lines). The thin solid lines radiating from the origin represent the damage criterion in the case of no cohesion ($C = 0$). The ellipses represent the yield criterion in the standard VP model of Hibler (1979) for different aspect ratios ($< 2$, outer; 2, center; $> 2$, inner ellipse). The numbers 0 to 4 indicate the state of (0) biaxial tension, (1) uniaxial tension, (2) biaxial tension and compression, (3) uniaxial compression and (4) biaxial compression.

     The coupling of $E$, $\eta$ and $\lambda$ with $d$ (Eq. 18 to 20, Dansereau et al., 2016) is such that the mechanical strength, as well as 10   the capability of the material to retain the memory of elastic deformations, decreases with increasing damage ($d \to 0$) and increases with healing. It is this coupling of the mechanical properties with the level of damage of the ice cover that allows the model to dissipate internal stresses in large, permanent deformations along leads once the ice pack is highly damaged while

reproducing the small deformations associated with the fracturing process and retaining the memory of elastic deformations over relatively low damage areas.

Analyses of the deformation and damage fields simulated using idealized geometries, simple forcing conditions and mechanical parameters values consistent with sea ice on geophysical scales (Dansereau et al., 2016) have demonstrated that the Maxwell-EB rheological framework successfully reproduces the anisotropy of sea ice deformation as well as the strong strain localization in both space and time and associated spatial scaling laws (Stern et al., 1995; Kwok, 2001; Marsan et al., 2004; Rampal et al., 2008, 2009; Weiss, 2008). The observed spatial and temporal coupling between these scalings is also represented (Weiss and Dansereau, 2017). Sensitivity analyses on the damage parameter $\alpha$, which sets the rate of viscous dissipation of the internal stress as a function of the increasing level of damage of the ice cover, have shown that the model, with few independent variables, can represent a large range of mechanical behaviours: from a regular, predictable stick-slip with a single damaging frequency related to the prescribed rate of healing, to a marginally stable, unpredictable deformation with temporal correlations in the damaging activity at all time scales below the material's healing time (Weiss and Dansereau, 2017). Over a range of values of this parameter, the model reproduces both the persistence of creeping leads in the ice cover and the activation of new leads with different shapes and orientations (Dansereau et al., 2016; Weiss and Dansereau, 2017).

In the channel flow simulations presented here, this new rheological model is implemented in a continuum modelling framework typical of regional and global sea ice models. In such framework, the ice cover is simulated as a 2-dimensional plate. Hence plane-stresses are assumed. The motion of the ice is described by the following Navier-Stokes type equation:

$$\rho h \left[ \frac{\partial \mathbf{u}}{\partial t} + (\mathbf{u} \cdot \nabla) \mathbf{u} \right] = A(\tau_a - \tau_w) - \rho h f \mathbf{k} \times \mathbf{u} - \rho h g \nabla H + \nabla \cdot (h\sigma), \tag{3}$$

with $\mathbf{u}$, the ice velocity, $\sigma$, the stress tensor, $\rho$, the ice density, $\tau_a$ and $\tau_w$, the air and water drags, $-\rho h f \mathbf{k} \times \mathbf{u}$, the Coriolis pseudo-force, $f$, the Coriolis parameter, $\mathbf{k}$, the upward unit vector normal to the ice surface, $g$, the gravitational acceleration and $-\rho h g \nabla H$, the force due to gradients in the sea surface dynamic height ($H$), which can be expressed in terms of the geostrophic ocean current velocity, $\mathbf{u}_w$ (e.g., Thomson et al., 1988).

In this 2-dimensional momentum equation, the variables $h$ and $A$ represent respectively the mean ice thickness and ice concentration over a model grid cell. The mean thickness, $h$, is the weighted sum of the average ice thickness over the ice-covered portion of the grid cell, $h_{thick}$, often referred to as the thickness of thick ice (e.g., Hibler, 1979), and the remaining open water (zero thickness). Hence $h = h_{thick}A$.

In the present uncoupled implementation of the Maxwell-EB model, thermodynamic processes are not accounted for. The ice density is considered constant and for $0 \leq A \leq 1$, mass conservation is ensured by the following evolution equations for the ice concentration and mean thickness of the ice-covered portion of the grid cell:

$$\frac{\partial A}{\partial t} + \nabla \cdot (A\mathbf{u}) = 0, \tag{4}$$

$$\frac{\partial h_{thick}}{\partial t} + (\mathbf{u} \cdot \nabla) h_{thick} = 0. \tag{5}$$

Ice thickening through mechanical redistribution, i.e., pressure ridge formation, is accounted for in a very simple manner. This is done on purpose, to test the input of the new rheology in the representation of the thickness distribution. If as a result of convergent ice motion, the ice coverage $A$ over a given model element exceeds unity, the excess concentration, $\max[0, (A-1)]$, is used to increment the ice thickness over that element, and the ice concentration is reset to 1 (see Appendix A, section A2.1). The mechanical sink of $A$ in this case reads:

$$A^- = -\max[0, (A-1)] \tag{6}$$

and the associated mechanical source of $h_{thick}$, equivalent to $h$ for $A = 1$, is

$$h^+_{thick} = \max[0, (A-1)]h_{thick}. \tag{7}$$

This redistribution scheme implies that ridging does not occur for $A < 1$. In the absence of quantitative observational support for the dependance of the amount of ridging on sea ice concentration, we chose the simplest possible approach.

Finally, the mechanical parameters $E$ and $\eta$ are coupled to the ice concentration $A$, as follows :

$$E = f_1(E^0, d) \exp\left[-c^*(1-A)\right], \tag{8}$$

$$\eta = f_2(\eta^0, d) \exp\left[-c^*(1-A)\right], \tag{9}$$

with $E^0$ and $\eta^0$, the elastic modulus and apparent viscosity of an undamaged ice cover, $f_1$ and $f_2$, the functional dependence of $E$ and $\eta$ on $d$, given respectively by Eq. (18) and (19) in Dansereau et al. (2016), and $c^*$ a non-dimensional, constant parameter. The dependence on the ice concentration follows the one suggested by Hibler (1979) and widely employed in VP models for the pressure term ($P$), which sets the ice strength in compression. The exponential function of $A$ simply allows the value of both $E$ and $\eta$ to be maximal when the ice concentration is $100\%$ ($A = 1$) and to decrease rapidly when leads open and $A$ drops ($\sim 10\%$ at $A = 90\%$, representing essentially a free drift state). It is employed to characterize the dependence of the elastic modulus (or effective elastic stiffness) on $A$ in the elasto-brittle models of Girard et al. (2010) and of Bouillon and Rampal (2015). In the case of $\eta$, this parametrization is compatible with the rapid decay of the apparent viscosity of granular media when decreasing their packing fraction from the close-packed limit (Aranson and Tsimring, 2006). In the present implementation of the Maxwell-EB model, this simple parametrization as well as the value of the non-dimensional parameter $c^*$ is the same for both mechanical parameters, but this could be refined in future developments of the rheology.

## 4   Channel flow simulations

The drift of sea ice within Nares Strait is thought to be primarily driven by the prevalence of northerly winds associated with the strong pressure gradient between the Lincoln Sea to the north and Baffin Bay to the south, and which are orographically channelled by the steep coastal topography of Ellesmere Island and Greenland (Ingram et al., 2002; Gudmandsen, 2004; Samelson and Barbour, 2008; Münchow, 2016). The most recurrent location for an ice bridge is in the southern Kane Basin (see Fig. 1c) (Kwok et al., 2010), where a stable arch is observed to form almost every year between November and March

(Barber et al., 2001) as a result of the convergence of a mixture of first and multi-year ice into Kane Basin (Gudmandsen, 2004; Kwok et al., 2010). Disintegration of the ice cover downstream of this arch leads to the opening of the North Water polynya (see Fig. 1c) in Smith Sound (Barber et al., 2001; Ingram et al., 2002).

In both the idealized and realistic simulations presented here, we aim to reproduce the formation of such an ice bridge. In the idealized case, the domain consists in a 120 km wide rectangular basin that converges into a 40 km wide, 40 km long channel (see Fig. 3b). The geometry, similar to that used in the idealized simulations of Dumont et al. (2009), is conceived to be roughly consistent with the shape of the constriction between Kane Basin and Smith Sound (dashed box, Fig. 3a). This simple configuration, symmetric with respect to the $y-$axis, facilitates the analysis of the dynamical behaviour of the model and, in particular, of the simulated states of stress. The dynamics described in the next section is however not specific to this geometry : simulations were also performed over different domains (narrower, longer channels, smaller basins) and produced similar results. The realistic domain covers the entire Strait down to Smith Sound (see Fig. 1c, grey shading). This configuration allows investigating the formation of secondary arches in various locations as well as other phenomena related to the presence of topographic features such as islands and fjords. In this case the mesh was built using the Gmsh mesh generator (Geuzaine and Remacle, 2009) and the GSHHG high resolution shoreline data (http://www.soest.hawaii.edu/pwessel/gshhg/) between 73 and 85° N and 280 and 320° W. This data is available in geodetic longitude/latitude on the WGS-84 ellipsoid. Here it is converted to Cartesian coordinates using a stereographic projection. The projection is centered on the North Pole, with the Greenwich meridian aligned on the positive $x$-axis and the Strait roughly oriented along the $y$-axis (see Fig. 3a).

The prescribed wind forcing is made as simple as possible to facilitate the analysis. Consistent with observations of orographic channelling, an along-channel, i.e., southward, wind stress, $\tau_a$, is applied. The stress is spatially uniform and increased steadily between 0 and 1 Nm$^{-2}$ over a period of 24 hours, and then held constant, to simulate the passage of a storm (Kwok, 2006; Samelson et al., 2006; Samelson and Barbour, 2008). Considering a simplified quadratic wind drag based on the absolute, instead of the relative, wind speed of the form

$$\tau_{\mathbf{a}} = \rho_a C_{da}|\mathbf{u}_a|\mathbf{u}_a, \tag{10}$$

with $\mathbf{u}_a$, the wind velocity, $\rho_a = 1.3 \, \mathrm{kg\,m^{-3}}$, the surface air density and $C_{da}$, the air drag coefficient, commonly set to $1.2\cdot10^{-3}$ in sea ice models following Hibler (1979), this corresponds to a maximum wind speed of $\sim 22 \, \mathrm{ms^{-1}}$ ($\sim 82 \, \mathrm{km\,h^{-1}}$). The ocean is at rest ($\mathbf{u}_w = 0$), hence the oceanic drag is given by the following quadratic formula

$$\tau_{\mathbf{w}} = \rho_w C_{dw}|\mathbf{u}|\mathbf{u}, \tag{11}$$

where $\rho_w = 1027 \, \mathrm{kg\,m^{-3}}$ is the density of sea water and $C_{dw}$ is the drag coefficient, set to $5.5 \cdot 10^{-3}$ (McPhee, 1980).

In both the idealized and realistic simulations, the Coriolis term in the the momentum equation (3) is discarded to retain symmetry in forcing conditions. The ocean is at rest, hence the force associated with gradients in the sea surface dynamic height is also zero. As the goal of these numerical experiments is to investigate the dynamical behavior of the Maxwell-EB model, thermodynamic processes are not accounted for. Simulations are therefore analyzed over a short period of time (3 days).

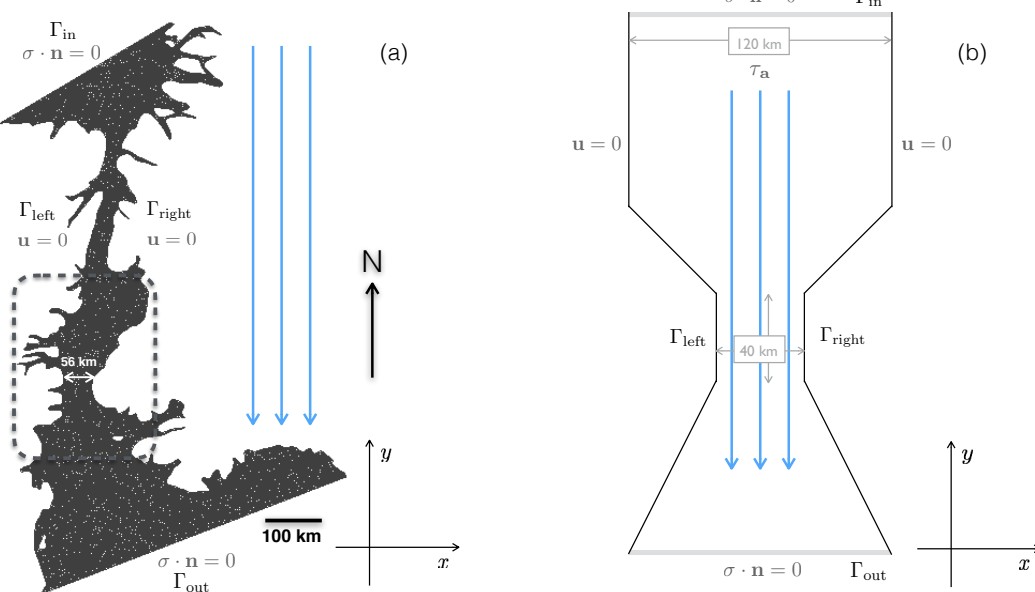

**Figure 3.** Domain and boundary conditions for (a) the realistic simulations of Nares Strait and (b) the idealized simulations of the constriction point between Kane Basin and Smith Sound (dashed box, panel a).

Mechanical parameter values are based on measurements within sea ice. An undamaged elastic (Young's) modulus of $E^0 = 5.85 \cdot 10^8$ Pa, on the order of that used by Girard et al. (2010) and consistent with an elastic shear wave speed of $500 \text{ ms}^{-1}$ in an heterogeneous ice pack (Marsan et al., 2011) is considered. Poisson's ratio is set to $\nu = 0.3$ (Timco and Weeks, 2010). The undamaged relaxation time, $\lambda^0$, is set to $10^7$ s ($\sim 115$ days), a value that allows the numerical scheme to converge, while also

5    ensuring that non-physical viscous dissipation over low damage areas of the ice cover is insignificant (Dansereau et al., 2016). The characteristic time for the healing process, $t_h$, which corresponds to the time required for the local level of damage $d$ to re-increase from 0 to 1, is set to $5 \cdot 10^5$ s ($\sim 5.7$ days) based on estimates of ice growth within open leads: Petrich et al. (2007) reported a time for the growth of 1 m of ice within an opening of 10 cm under air temperatures of $-15°$C between $10^5$ and $10^6$ seconds. A constant healing rate is used (see Eq. (A3)). As thermodynamic processes are not accounted for and as healing

10    is meant to represent only the local recovery of the ice mechanical strength within refreezing leads, not the thermodynamic growth of ice within polynyas, healing is turned off as soon as the ice concentration locally drops below $75\%$ in the simulations, which occurs when and where the ice detaches from a stable ice bridge or from the coast. This avoids unphysical situations where healing would cause $d$ to reincrease towards the undamaged value of 1 over low ice concentration or even open water areas. As the dependancy of the mechanical parameters on the ice concentration (Eq. (8) and Eq. (9)) ensures that the rheology

15    term drop to less than $1\%$ of its undamaged value for $A = 75\%$, including this threshold has no significant impact on the results presented here. Table 1 summarizes all model parameter values employed in the simulations presented here.

As a determining factor for the formation of stress-free surfaces is the cohesive strength of the material, simulations with a different range of values of cohesion were compared. The field of $C$ was set as follows. First, its spatial distribution for all simulations using the idealized or realistic domain was obtained by randomly drawing a value in the non-dimensional interval $[1, 2]$ over each model element. This noise was then multiplied by a minimum value of cohesion, $C_{min}$, such that

$C \in [C_{min}, 2 \times C_{min}]$. This minimum cohesion was varied between $2, 5, 10, 20, 30$ kPa. Hence, the same *spatial distribution* of $C$ was used in all simulations, but the magnitude of $C$ was varied between simulations. In all simulations, the disorder introduced in the field of cohesion is quenched (Herrmann and Roux, 1990): it is set once, at the beginning of each simulation, and is passively advected with the ice flow. It is important to note that this disorder, because it is set randomly, does not introduces spatial correlations in the model. Therefore, it does not prescribe the location of the simulated ice leads and bridges.

Model simulations using the same value of $C_{min}$ but different random spatial distributions of the disorder on $C$ produced similar results, comparable in all aspects to those discussed below. The largest values of $C$ employed are consistent with in-situ stress measurements in the Beaufort Sea reported by Weiss et al. (2007) and Weiss and Schulson (2009) (see Fig. 8a). According to the presumed scale effect on shear strength, set by the size of the defects/heterogeneities present in the ice cover (Schulson, 2004; Weiss et al., 2007), lower values of $C$ are consistent with larger defects/heterogeneties sizes and a lower shear

strength.

| Parameters | | Values |
|---|---|---|
| Internal friction coefficient | $\mu$ | 0.7 |
| Ice density | $\rho$ | 900 kg m$^{-3}$ |
| Undamaged elastic modulus | $E^0$ | $E^0 = 5.0 \cdot 10^8$ Pa |
| Undamaged apparent viscosity | $\eta^0$ | $10^7 \times E^0$ Pa s |
| Undamaged relaxation time | $\lambda^0$ | $10^7$ s |
| Minimum cohesion | $C_{min}$ | $2, 5, 10, 20, 30$ kPa |
| Damage parameter | $\alpha$ | 4 |
| Characteristic time for damage | $t_d$ | 4 s (idealized sim.), 6 s (realistic sim.) |
| Characteristic time for healing | $t_h$ | $5 \cdot 10^5$ s |
| Mean model resolution | $\Delta x$ | 2 km (idealized sim.), 3 km (realistic sim.) |
| Model time step | $\Delta t$ | 4 s (idealized sim.), 6 s (realistic sim.) |
| | $c^*$ | 20 |
| Air drag coefficient | $C_{da}$ | $1.5 \cdot 10^{-3}$ |
| Air density | $\rho_a$ | 1.3 kg m$^{-3}$ |
| Water drag coefficient | $C_{dw}$ | $5.5 \cdot 10^{-3}$ |
| Water density | $\rho_w$ | 1027 kg m$^{-3}$ |

**Table 1.** Model parameters for the idealized and realistic channel flow simulations.

All simulations are initialized with a uniform ice thickness of 1 m. This value is consistent with the median ice draft in Nares Strait over the 2003-2012 period reported by Ryan and Münchow (2017). Moreover, we find that the evolution of the simulated fields described in the following sections does not depend on the specific value of the initial ice thickness, as long as the coverage is initially uniform. The domain is initially completely covered with undamaged ice ($A = 1$, $d = 1$), so that the location of ice bridges is not prescribed. Simulations are started from rest. A no-slip condition ($\mathbf{u} = 0$) is applied at the lateral boundaries $\Gamma_{\text{left}}$, $\Gamma_{\text{right}}$, representing the coasts (see Fig. 3a and b). The channel is open at its top ($\Gamma_{\text{in}}$) and bottom ($\Gamma_{\text{out}}$) boundaries with the Neumann condition $\sigma \cdot \mathbf{n} = 0$. The value of all transported quantities is prescribed on $\Gamma_{\text{in}}$ and $\Gamma_{\text{out}}$ and represent undamaged ice entering the channel, i.e., with $d = 1$, $A = 1$, $h = 1$ m, $\sigma = 0$, and with $C$ randomly drawn from the same uniform distribution prescribed as initial conditions.

The model is entirely developed within the C++ environment RHEOLEF (Saramito, 2013a). The numerics is based on finite elements and variational methods. The equations of motion are cast in the Eulerian frame and discontinuous Galerkin methods are used to handle advective processes (Saramito, 2013b), as well as the non-linear terms arising in the objective derivative of the stress tensor (see Appendix A). Polynomial approximations of degree 1 are used for the velocity field and all advected fields ($\sigma$, $A$, $h$, $d$, $C$) are piecewise constant. Appendix A presents the details of the numerical scheme.

Unstructured meshes with triangular elements are used. The average spatial resolution, $\Delta x$, is constant. It is of 2 km in the idealized simulations and of 3 km in the realistic simulations. The time step is of 4 s in the idealized and of 6 s in the realistic simulations. The results obtained here are not conditional to the choice of spatial resolution, as long as it allows resolving the flow of ice through the narrowest point of the channel with the no-slip boundary condition.

## 5    Results

### 5.1    Dynamical behaviour

Here we investigate the formation of arches and stable ice bridges in the Maxwell-EB model and analyze the evolution of the simulated level of damage, ice velocity and internal stresses.

Idealized and realistic simulations with a different range of values of $C$ (see section 4) were compared. The evolution of the applied wind forcing (dashed line) and of the damage rate (solid grey line), defined as the number of damaged elements per model time step times their respective distance to the Mohr-Coulomb or tensile damage criterion (see Dansereau et al. (2016)), is represented for one idealized and one realistic simulation using $C_{min} = 20$ kPa in Fig. 4a and Fig. 6a respectively. The spatial distributions of the level of damage and ice concentration are shown at three different stages of these two simulations on Fig. 4b, c and Fig. 6b, c. We first discuss the idealized simulations results and then comment on the realistic case.

### 5.1.1 Idealized simulations

In all simulations using different values of $C_{min}$, high deformation rates first concentrate along narrow, concave damaged features that form in the interior and downstream of the channel (Fig. 4b, panel 1). The profile of the $y$-component of the ice velocity, $u_y$ for the simulation with $C_{min} = 20$ kPa shown in Fig. 4 is plotted along the central meridional axis of the idealized domain (red curve, Fig. 5a). It shows that after the onset of damaging (indicated by the first peak in damage rate on Fig. 4a), the ice over this portion of the domain is set in motion while the undamaged ice upstream remains motionless (Fig. 5a, stage 1). The sharp no-flow transition coincides with the constriction point that defines the entrance to the channel, across which an ice arch has formed (see Fig. 4c). It is important to note that here, "no-flow" or, as later mentioned, "flow stoppage" is not defined as a zero drift speed, but rather as a drift velocity on the order of that associated with strictly elastic deformations within an undamaged ice cover (here, $|\mathbf{u}|$ is on the order of $10^{-5}$ ms$^{-1}$ before the onset of damage).

As the wind forcing is further increased, damage propagates upstream of the channel, which corresponds to the second peak in the damage rate seen on Fig. 4a. Linear converging features form along both sides of the channel, leaving some stagnant ice with low damage near the coasts in the converging part of the basin (see Fig. 4b, panel 2). This creates an inner flow channel, inside of which the ice is set in motion everywhere and the drift is almost uniform (see Fig. 5, panel 2). Similar results were also obtained by Dumont et al. (2009) and are consistent with the observed jamming of grains and sand along the walls of a silo (Munch-Andersen, 1986; Munch-Andersen et al., 1992) as well as with previous discrete elements (Morrissey et al., 2013) and continuum (Wang and Ooi, 2015) model simulations of the discharge of a granular solid in a silo.

Up to that point, the overall spatial and temporal evolution of damage is similar between the simulations using different ranges of cohesion. However, as the local value of $C$ sets the local damage criterion in the Maxwell-EB model, i.e., the local value of $\sigma_c$ and $\sigma_t$, the minimum value of cohesion over the domain controls the timing of the onset of damaging in the simulations, with damaging occurring sooner as $C_{min}$ is lower.

Beyond this point, drift velocities in the middle of the domain progressively decrease as ice converges into the inner ice channel (not shown). The simulated dynamics then differ between the simulations using different values of $C$. For the smaller values of cohesion used ($C_{min} = 2, 5, 10$ kPa) ice arches that form at the opening of the channel eventually collapses as the wind forcing is further increased. In the simulations with higher values of cohesion ($C_{min} \geq 20$ kPa), a stable bridge forms and the flow of ice within and upstream of the channel effectively stops (see Fig. 5a, panel 3).

We further investigate the mechanism behind the formation of the ice arches by analyzing the simulated states of stress for the idealized simulation with $C_{min} = 20$ kPa. Similar results were obtained for $C_{min} = 30$ kPa. The symmetry of the idealized domain facilitates the analysis, but the same general conclusions also apply to the realistic simulations. Figure 5b represents the instantaneous profiles along the central meridional axis of the principal stresses $\sigma_1$ and $\sigma_2$ at the times indicated by the numbers 1, 2 and 3 on Fig. 4a.

Internal stresses within the initially undamaged ice cover are compressive ($\sigma_1, \sigma_2 > 0$) over the basin, change sign at the middle of the channel (at $y = 0$) and are tensile ($\sigma_1, \sigma_2 < 0$) downstream of the channel (not shown). At the onset of damage,

$\sigma_2$ becomes negative (i.e., tensile, see Fig. 2) over the converging part of the basin ($y < 50$ km) and in the interior of the channel. Within and downstream of the channel, *multiple* minima in $\sigma_2$ are observed. These minima are collocated with either a maximum (positive) value of $\sigma_1$ (shearing state of stress) or a minimum (negative) value of $\sigma_1 < 0$ (tensile state of stress) and correspond to the location of arch-like features (see Fig. 4b, panel1). Figure 5c, panel 1 shows the elements that have exceeded either the local Mohr-Coulomb (in blue), the tensile (in red) criterion or both criteria (in black) from the beginning up to this point in the simulation and confirms that these features were formed by a shear, tensile or a combination of both failure mechanisms.

Later in the simulation, the ice cover fails preferentially in shear upstream of the channel, while tensile failure progressively becomes predominant within and near the exit of the channel (see Fig. 5c, panels 2 and 3). Once the stable ice bridge is formed, its location is clearly seen on the field of ice concentration (Fig. 4c, panel 3) and on the profile of vertical ice velocity (Fig. 5a, panel 3). The profile of the principal stress components just upstream of the bridge gives evidence that this structure sustains *biaxial tensile* stresses (Fig. 5b, panel 3). This is an important point, as models based on the standard elliptical yield curve do not account for uniaxial or biaxial tensile strength and hence would not be able to reproduce the formation of a stable ice arch with self-obstruction to flow under the stress conditions simulated here. Consistent with the formation of a stable stress-free surface and with the detachment of the ice from the bridge, both principal stress components decrease to zero downstream of the bridge (Fig. 5b, panel 3). Velocity profiles (Fig. 5a, panel 3) are uniform downstream of the no-flow transition with $u_x \approx 0$ and $u_y \approx 0.43 \text{ ms}^{-1}$ corresponding to the free drift velocity

$$u_y = \sqrt{\frac{\tau_a}{\rho_w C_{dw}}}, \tag{12}$$

for $\tau_a = 1 \text{ Nm}^{-2}$. Simulations with a spatial resolutions of 2, 4 and 8 km were compared and produced similar results (not shown), demonstrating that the mechanical and dynamical behaviour of the Maxwell-EB model described here does not depend on its spatial resolution.

### 5.1.2 Realistic simulations

The temporal evolution of damage, velocity and internal stress in the realistic simulations is similar to the idealized case. The ice first fails near the exit of Kane Basin and rapidly weakens in Smith Sound (see Fig. 6b, $t = 6$ hours). Damage then progressively propagates within and upstream of Kane Basin along closed arches up to the entrance of Robeson Chanel and eventually along open arches and more linear features upstream of the Strait (Fig. 6b, $t = 24$ and 72 hours). The opening of multiple arch-like leads under the effect of the wind forcing appears clearly on the corresponding fields of ice concentration (Fig. 6c). Comparison of these fields and corresponding snapshots of the ice drift velocity over the domain indicates that these leads divide the ice into relatively undamaged plates, or floes. Drift speeds are piecewise constant over the floes and discontinuous across the floes (Fig. 7, $t = 6$ hours and left inset), a feature that is also reproduced in the idealized simulations, as shown by the "stair-case" like profiles of $u_y$ (see Fig. 5a). This behaviour is consistent with the motion of the Arctic ice

cover as revealed by Synthetic Aperture Radar imagery analysis and RGPS motion products (Kwok, 2001; Moritz and Stern, 2001).

Also evident from the fields of ice concentration and corresponding fields of ice drift speed are regions where the ice remains landfast even under strong wind forcing, either enclosed in bays, inlets and fjords or between islands and the nearby shore. This is consistent with observations of landfast first year ice in the Strait in protected areas along the coasts of Ellesmere and Greenland and within Kane Basin and Smith Sound in winter and spring (Mundy and Barber, 2001; Yackel et al., 2001). Fig. 6c shows that the ice progressively detaches from the coast or landfast ice covered areas. The sharp gradients in ice velocity between these motionless regions and the fast flowing ice downstream are well simulated (see Fig. 7, $t = 72$ hours, right inset).

For $C_{min} \geq 20$ kPa, stable ice arches form in the realistic simulations. Consistent with observations, the main bridge is
located at the constriction point between Kane Basin and Smith Sound (see Fig. 6c and 7). Downstream of this bridge, the ice concentration rapidly drops, as the ice detaches and is driven out the channel at the free drift speed, leading to the opening of the North Water polynya (see Fig. 6c, $t = 72$ hours). The model also simulates the detachment of the ice from the leeward side of islands (see Fig. 7, $t = 72$ hours) and, as the wind forcing is further increased, from a secondary bridge at the entrance of Kane Basin (see Fig. 6c, $t = 72$ hours), which is also observed on satellite imagery (Kwok et al., 2010).

Internal stresses in the realistic simulations also evolve similarly to the idealized case. Figure 8a shows the evolution of the proportion of the simulated (uniaxial and biaxial) tensile, biaxial tensile-compressive and biaxial compressive stresses over the domain in the realistic simulation shown in Fig. 6 and 7. The instantaneous states of stress are represented in the principal stresses plane in Fig. 8c ($t = 72$ hours) after the formation of a stable ice bridge, i.e., after the proportion of the three stress types and the repartition of stresses in the principal stresses plane has stabilized. The states of stresses are compared to in situ stress
measurements from the Beaufort Sea reported by Weiss and Schulson (2009) (Fig. 8b[2]). Consistent with these observations, biaxial tensile stresses occur in a large number both prior and after the formation of a stable ice bridge in the Strait. This again supports the relevance of accounting for some resistance in pure tension in sea ice models. Biaxial tensile-compressive, i.e., shear, stresses are the dominant type of stresses at all times during the simulation. Also consistent with in-situ measurements, *large* biaxial compressive stresses do not appear so frequent compared to pure tensile and biaxial tensile-compressive states
(Weiss et al., 2007; Weiss and Schulson, 2009), although convergent ice motion occurs over a significant portion of the domain. It is important to note that isotropic stresses ($\sigma_1 = \sigma_2$) are frequent in the observations and absent in the model. Theses are associated with thermal processes, i.e., thermal expansion/contraction (Richter-Menge et al., 2002), which are not represented in the present Maxwell-EB framework.

### 5.1.3   Consequences in terms of sea ice export

While studies have highlighted the strong interannual variability of the flux of ice through Nares Strait (Kwok et al., 2010), observations have also shown a tendency for ice bridges to form later and break earlier in the 1990s than in the 1980s (Barber

---

[2]The value of $q$ used on this figure by Weiss and Schulson (2009) is based on a value of the internal friction coefficient $\mu$ of 0.9 estimated by Schulson (2006a) from laboratory tests on first-year Arctic sea ice and is slightly higher than the value used in the model (0.7).

et al., 2001), suggesting that an increased ice outflow through the Strait should be expected in a future climate. Except for a 2 months period, no ice bridge formed between Kane Basin and Smith Sound in the winters of 2007/2008 to 2009/2010. During that time, the (southward) ice and ocean velocities in the Strait were observed to increase and surface waters became fresher (Münchow, 2016). The year 2007 in particular was characterized by the absence of any ice bridge and flow stoppage in the Strait, which resulted into a record ice area and volume export, equivalent to twice their estimated average value over the 1997 to 2009 period (Kwok et al., 2010) and into a maximum observed median draft of the drifting ice over the 2003 to 2012 period (Ryan and Münchow, 2017). The following question arises: would a tendency towards shorter lifespan of ice bridges and associated increased ice export through straits be the consequence of a thinning and/or of a mechanical weakening of the ice cover?

In a recent study, Gimbert et al. (2012b) analyzed in the frequency domain the drift of Arctic sea ice over the period 1979-2008 as estimated from the International Arctic Buoy Data Programme dataset and identified an increase of the inertial motion of the ice cover. With the use of a simple ice-ocean boundary layer, dynamically coupled model, Gimbert et al. (2012a) were able to relate this evolution to a *genuine* mechanical weakening of the ice pack in the Arctic and peripheral zone over the period 2002-2008 relative to the previous 22 years, that is, a weakening that is independent of a concomitant sea ice thinning and could

be due to an evolution towards a more fractured ice cover and smaller ice floes. In this section we analyze a supplemental set of simulations which allows investigating the model's behaviour when changing few of its mechanical parameters to represent this mechanical weakening scenario.

In accordance with Gimbert et al. (2012a), who estimated a decrease in mechanical strength by a factor of about 1.5 to 1.75 between the periods 1979-2001 and 2002-2008 in both summer and winter, we compare realistic simulations of Nares Strait

in which both the minimum uniaxial compressive strength and minimum tensile strength, set in the model by the cohesion, $C$, are reduced by a factor of 2 compared to the simulation with $C_{min} = 20$ kPa (in which a stable ice arch does form under the specific forcing conditions applied here, see section 5.1.2). The initial prescribed ice thickness ($h = 1$ m) is unchanged. Summer conditions are distinguished from winter conditions by increasing the prescribed healing time to $t_h = 365$ days, so that the recovering of mechanical strength associated with the refreezing of leads is negligible over the short duration of the

simulations. Figure 9a shows the time series of the meridional component of the simulated ice drift velocity, averaged zonally across the Strait ($\overline{u_y}$) near the exit of Kane Basin, i.e., just upstream of the Kane Basin ice bridge. Four scenarios are compared: a stronger ($C_{min} = 20$ kPa) and a weaker ($C_{min} = 10$ kPa) ice cover, corresponding respectively to 1979-2001 and 2002-2008 conditions, in both winter and summer. Figure 9b shows instantaneous fields of the ice drift speed, $|\mathbf{u}|$, over the domain at $t = 48$ hours in these four cases.

In all cases, arch-like leads are still clearly defined and occur in similar locations as in the stronger ice (1979-2001) winter case discussed in section 5.1.2 (not shown). In this winter case, the stable ice arch that forms at the exit of Kane Basin causes the meridional ice flow to stop completely within 48 hours (see figure 9a). In all of the weaker ice cover scenarios (2002-2008 period and/or summer), none of the ice arches formed near the exit of Kane Basin nor secondary arches formed elsewhere sustain the applied wind forcing and all ice bridges eventually collapse. While the flow of ice can be significantly slower over

narrower than over larger portions of the Strait (see Fig. 9b), complete flow stoppage does not occur in any of these weaker ice

cover scenarios over the time period analyzed here (see Fig. 9a), which allows a much larger portion of ice to be flushed out of the Strait. Comparison of the fields of $|\mathbf{u}|$ in the summer and winter cases moreover suggests that the healing of the fractured ice cover might play a significant role in supporting stable ice bridges. The reduction of the extent of landfast ice between the summer and winter cases implies that healing also helps maintaining the stagnant ice within bays and fjords.

It is important to note that these numerical experiments are by no means an attempt to determine the physically appropriate value of the cohesion of the ice cover in either of these scenarios, as the presence or not of a stable ice bridge also depends on other mechanical parameter values (in particular, the elastic modulus), on the magnitude and form of the applied wind forcing as well as on the prescribed initial ice thickness. Instead, this exercise serves to show that by varying few mechanical strength parameters in a range of values which, based on observations, seems physically appropriate, the Maxwell-EB model is able to reproduce a wide range of dynamical behaviours. Conversely, the simulations suggest that mechanical weakening, independent of thinning of the ice cover, can play a role in increased ice exports through narrow outflow pathways of the Arctic. This has implications in terms of ice export because the ice cover is expected to continue weakening, i.e., being more fragmented in the future.

## 5.2 Ice thickness distribution

We now investigate the ice thickness distribution simulated by the Maxwell-EB model and, in particular, the distribution of ridged ice, i.e. the ice that has thickened through mechanical redistribution. In the simulations analyzed for this purpose we set $C_{min} = 10$ kPa. In these conditions, a stable ice bridge does *not* form over the 3 days period analyzed here: the flow of ice upstream the channel slows but does not stop, which allows the ice to be more rapidly redistributed under the applied wind forcing.

Figure 10 shows the instantaneous fields of the mean ice thickness over a grid cell, $h$, after 3 days of simulation in an idealized case using $C_{min} = 10$ kPa and the probability density functions, $P(h)$, corresponding to instantaneous fields of $h$ at different times after the onset of damaging in this simulation. As the spatial resolution is approximately constant over the domain, $P(h)$ here is estimated by the frequency histogram of $h$ and normalized by the total number of model grid cells (i.e., $h$ values are not weighted by the areal fraction of the corresponding grid cell). This PDF is dominated by a strong mode at 1 m that corresponds to the initial prescribed value of $h$. The ice thickness increases over time within the channel in the converging part of the basin, defining oriented regions of ridged ice that correspond to the areas of shear failure identified on Fig. 4c. This part of the basin is essentially represented by the tail of the distribution ($h > 1$ m), which at all times is well described by a negative exponential (the coefficient of determination for the goodness of the fit varies between $90\%$ and $98\%$). This function is of the form

$$P(h) \sim \exp^{-\frac{h}{h^*}}, \tag{13}$$

with $h^*$ increasing in time until the convergence and thickening of ice at the entrance of the channel reduces the flow of ice (see Fig. 10). The flattening of the PDF is a signature of the thickening of the ice along these highly localized features.

The part of the distribution with $h < 1$ is characterized by a second mode near $h = 0$ associated with the presence of open water (e.g., Wadhams, 1981, 1994; Haas et al., 2006; Haas, 2009), which becomes more important as ice is driven out of the domain. At $t = 3$ days (orange curve), the ice downstream of the channel has nearly left the domain and a large patch of ice is almost completely detached from the interior of the channel. The fact that $P(h)$ is not zero for values of $h \in ]0, 1[$ at this point is mostly attributable to numerical diffusion. For polynomial approximations of degree 0 of advected quantities (see section 4), the Galerkin discontinuous method coincides with a finite volume scheme with upwinding, which is known to be diffusive. As expected, numerical diffusion is most important at the edge of the detached ice, where mechanical stresses vanish, gradients between the ice and opening water are the strongest and drift velocities the highest. It is important to note however that the present continuum model is not meant to represent the dynamical behaviour of sea ice in regions of low ice concentration dominated by free drift conditions such as the marginal ice zone, but rather that of the ice pack. A Lagrangian scheme would perhaps be a more natural approach to simulate the edge of the detached ice, although some diffusion would be unavoidable in free drift mode as some remeshing would be required. Where ice concentration is high ($A > 90\%$), mechanical stresses are significant and constantly redistributed under damaging. In the Maxwell-EB model, the associated deformation is highly localized in both space and time, which acts to mitigate numerical diffusion and re-increase gradients in all fields.

The realistic simulations show a similar evolution of the ice thickness in the channel. In the funnel-like entrance to Nares Strait, the converging part of Kane Basin and upstream of coasts and islands, the ice builds-up along narrow and oriented features (see Fig. 11a). The effect of numerical diffusion in smoothing these features and reducing the localization of the thickest ice increases with ice drift velocities downstream of the domain. Nevertheless, at all times the simulated probability density function is strongly asymmetric, consistent with thickness distributions estimated for sea ice with little history of melting in the open Arctic ocean (e.g., Haas, 2009) and at the entrance of Nares Strait (Haas et al., 2006). As in the idealized case, the strong localization of the ridged ice translates into an exponential tail for $P(h)$ of the form of (13) with $h^*$ increasing in time (see Fig. 11b).

According to Eqn. (5) and to the simple redistribution scheme employed here, the evolution of $h$ is a function of the mechanical redistribution term given by Eqn. (7), which itself is a function of the flux of ice concentration $\nabla \cdot (A\mathbf{u}) = \mathbf{u} \cdot \nabla A + A \nabla \cdot \mathbf{u}$. The spatial distribution of $h$ in the present Maxwell-EB model therefore depends essentially on the simulated velocity field. Furthermore, both simulations, idealized and realistic, show a similar exponential decrease for the tail of $P(h)$. This suggests that the localization of the thickest ice does not arise from the complexity of the domain geometry. The shape of the tail of $P(h)$ is also conserved when using a lower spatial resolution, for instance, with $\Delta x = 4$ and 8 km in idealized simulations (orange, dotted and dotted-dashed curves on Fig. 10b), suggesting that this property of $P(h)$ is not resolution-dependant either. This is consistent with the fact that there is no characteristic scale for the localization of damage and deformation in the model beyond the scale of the model element (see Dansereau et al., 2016, sections 6.1 and 6.2), i.e., that at all spatial resolutions, the simulated deformation is highly localized. Additional simulations with different ranges of cohesion (not shown) indicate that the value of $C$ not impact the shape of the ice thickness distribution either, but only the rate at which the exponential tail flattens, i.e., the rate at which the ice cover thickens. In brief, the strong localization of ridged ice in the model therefore

30  appears to be only the consequence of its capability to reproduce the extreme localization of ice deformation and associated sharp gradients in the ice velocity field.

## 6  Conclusions

In this paper we have presented the results of a first implementation of the Maxwell-EB rheology for modelling sea ice on geophysical scales. Idealized and realistic simulations of the flow of ice through Nares Strait have shown that the Maxwell-EB

sea ice model is able to reproduce

- the formation of multiple arch-like leads within the ice cover downstream, in the interior and upstream of the channel, in agreement with observations of ice conditions in narrow straits of the Arctic (Sodhi, 1977; Kwok et al., 2010). As these features appear in high numbers in both highly idealized and realistic simulations, they are not attributable to the complexity of the domain geometry or boundary conditions but to the mechanical behaviour of the Maxwell-EB model

itself.

- the formation of a stable, concave ice bridge, the associated stoppage of the ice flow upstream and the opening of a polynya downstream of the bridge. In the realistic simulations of Nares Strait, the location of the main bridge is consistent with the stable arch observed to form seasonally downstream of Kane Basin.

- regions of relatively undamaged ice with uniform, plate-like motion, and extreme gradients in ice velocity across the

leads that delimit these regions, consistent with RGPS observations (see Kwok (2001); Moritz and Stern (2001)). In particular, the model reproduces the very sharp gradients in ice velocity, concentration and thickness associated with the edge of the ice bridge while remaining numerically stable.

- the presence of landfast ice along portions of the coast of Nares Strait, in agreement with observations (Mundy and Barber, 2001; Yackel et al., 2001). In the model, this landfast ice resists the strong applied wind forcing. The associated

sharp gradients in ice velocity between this stagnant and the nearby fast flowing ice are well represented.

- states of stress that are overall in good agreement with in-situ measurements in Arctic sea ice. In particular, the simulations have shown that uniaxial/biaxial tensile stresses are rather frequent, hence pointing out the importance of accounting for some resistance in pure tension in sea ice models.

- the strong localization of ridged ice and an associated thickness distribution with an exponential tail, in agreement with

probability density functions calculated from sea ice thickness measurements.

This last point is an important outcome of this first implementation of the Maxwell-EB rheology in a realistic context, as the treatment of the mechanical redistribution of the ice thickness, in particular of the ridging process, has been the subject of numerous studies since the 1970's and continues to challenge the sea ice modelling community to this day. Although the very simple redistribution scheme used here does not include multiple ice categories nor allows prescribing the thickness of the ice

involved in the ridging, the Maxwell-EB model seems to allow ice to thicken in regions of strong convergence and shear. In the Maxwell-EB model, this capability of accounting for a sufficient thickening of the ice as well as the spatial localization of extreme thickness values arises from the appropriate description of extreme strain localization. On a mechanical point of view, this may therefore question the relevance of using multi-categories redistribution schemes.

In terms of the simulated thermodynamic fluxes and ice energy balance however, a number of studies have highlighted the sensitivity of simulation results to the number of ice thickness categories used (e.g., Castro-Morales et al., 2014), and have pointed to an overall underestimation of ice thickness due the under-representation of the thermodynamic growth of thin ice using a two-level versus a multi-categories model (e.g., Walsh et al., 1985). As opposed to mechanical redistribution processes, thermodynamic processes are expected to "seek the mean" (Thorndike et al., 1975) and smooth the distribution by allowing ice to grow over open water, thinner ice to thicken faster than thick ice and thicker ice to melt comparatively faster (Haas, 2009). A thorough comparison of simulated ice thickness distributions model against observations therefore calls for larger scale, longer term simulations including thermodynamic processes. The coupling of the Maxwell-EB rheology with a thermodynamic component is under-way and will allow evaluating the impact of the representation of multiple, fine ice leads in the Maxwell-EB model on the simulated heat fluxes between the atmosphere, ice and ocean.

Besides numerical efficiency, other advantages of using a simple redistribution scheme such as the one employed here is that no thickness redistribution function needs to be assumed and the redistribution is not directly tied to the prescribed failure strength of the ice. In the Maxwell-EB model, the prescribed strength is instead based on in-situ stress measurements, which point to a Mohr-Coulomb failure criterion and directly provide information on the relative amount of shear and tensile strength. In particular, both the observations and numerical simulations here suggest that prescribing a cut-off for biaxial compressive strength (equivalent to the pressure, $P$, in VP models) is unnecessary (see figure 8). Instead, the uniaxial compressive strength, $\sigma_c$, and maximum uniaxial tensile strength, $\sigma_t$, appear to be more relevant to represent adequately the strength of the ice cover. The Maxwell-EB model presents the advantage that both these quantities are set through a single parameter, the cohesion, $C$. The channel flow simulations performed here moreover showed that varying this single parameter in the limit of physically acceptable values and in agreement with the suggested evolution of the failure strength of the ice pack in recent years (Gimbert et al., 2012a), while keeping the initial thickness unchanged, is sufficient to reproduce different dynamical behaviours of the ice cover (the presence of stable or unstable bridges, the opening of different size polynyas and the associated ice fluxes). This therefore suggests that the Maxwell-EB model could be an efficient tool for modelling the deformation and drift of sea ice in complex flow regimes and under different climate scenarios.

## Appendix A:  Equations and numerical scheme

### A1    System of equations

With the Coriolis term neglected, the ocean at rest and the mean ice thickness over the grid cell given by $h = h_{thick}A$, the momentum equation, reads

$$\rho h_{thick} A \left[ \frac{\partial \mathbf{u}}{\partial t} + (\mathbf{u} \cdot \nabla)\mathbf{u} \right] = A \left( \tau_{\mathbf{a}} - \rho_w C_{dw} |\mathbf{u}|\mathbf{u} \right) + \nabla \cdot (h_{thick} A \sigma). \tag{A1}$$

Following Connolley et al. (2004), both the air and water drag terms are weighted by the local ice concentration to account for the fraction of open water within a grid cell.

The Maxwell-EB constitutive law writes

$$\frac{1}{E} \left[ \frac{\partial \sigma}{\partial t} + (\mathbf{u} \cdot \nabla)\sigma + \beta_a(\nabla \mathbf{u}, \sigma) \right] + \frac{1}{\eta}\sigma = \mathbf{K} : \dot{\varepsilon}.$$

where $\dot{\varepsilon}$ is the strain rate tensor, here equivalent to the rate of strain tensor, and the term $\beta_a$ in the objective Gordon-Schowalter

derivative of $\sigma$ accounts for the effect of rotation and deformation of the stress tensor. This term reads

$$\beta_a(\nabla \mathbf{u}, \sigma) = \sigma W(\mathbf{u}) - W(\mathbf{u})\sigma - (\sigma D(\mathbf{u}) + D(\mathbf{u})\sigma)$$

in the upper convected form, with $D(\mathbf{u}) = \frac{\nabla \mathbf{u} + \nabla \mathbf{u}^T}{2}$ and $W(\mathbf{u}) = \frac{\nabla \mathbf{u} - \nabla \mathbf{u}^T}{2}$, the symmetric and anti-symmetric parts of the velocity gradient. In plane stress conditions, the dimensionless stiffness tensor, $\mathbf{K}$, is defined in terms of $\nu$, Poisson's ratio, such that for all symmetric tensors $\epsilon = \epsilon_{ij} \ \forall \ i,j; \ 1 \leq i,j \leq 2$, $(\mathbf{K} : \epsilon)_{ij} = \frac{\nu}{1-\nu^2}\mathrm{tr}(\epsilon)\delta_{ij} + 2\frac{1}{2(1+\nu)}\epsilon_{ij}$. With the following coupling

of the mechanical parameters with respect to the level of damage and ice concentration,

$$E = E^0 d \exp[-c^*(1 - A)],$$

$$\eta = \eta^0 d^\alpha \exp[-c^*(1 - A)],$$

$$\lambda = \frac{\eta^0}{E^0} d^{\alpha-1} = \lambda^0 d^{\alpha-1},$$

(Eq. 18-20, 24 and 25 Dansereau et al., 2016), where $\alpha$ is a constant greater than one such that the relaxation time for the

stress, $\lambda$, decreases with increasing level of damage and increases with healing (see Dansereau et al. (2016), section 4.1.5), the constitutive equation can be alternatively written in the following form

$$\lambda^0 d^{\alpha-1} \left[ \frac{\partial \sigma}{\partial t} + (\mathbf{u} \cdot \nabla)\sigma + \beta_a(\nabla \mathbf{u}, \sigma) \right] + \sigma = \eta^0 d^\alpha \exp[-c^*(1 - A)]\mathbf{K} : D(\mathbf{u}). \tag{A2}$$

The damage evolution equation reads

$$\frac{\partial d}{\partial t} + (\mathbf{u} \cdot \nabla)d = \left( \min\left[ 1, \frac{\sigma_t}{\sigma_2}, \frac{\sigma_c}{\sigma_1 - q\sigma_2} \right] - 1 \right)\frac{1}{t_d}d + \frac{1}{t_h}, \ \ 0 < d \leq 1, \tag{A3}$$

with $t_d$, the characteristic time for damaging, set by the ratio $\frac{\Delta x}{c}$ with $\Delta x$, the spatial resolution and $c$, the speed of propagation of elastic waves in the ice cover which carry the damage information (see Dansereau et al. (2016), sections 3.3.1 and 4.4.1),

$t_h$, the characteristic time for healing (see Dansereau et al. (2016), sections 3.3.2 and 4.4.2), $\sigma_c$ and $\sigma_t$ defined in terms of $C$ as in Eq. (1) and Eq. (2) respectively and $\sigma_1$ and $\sigma_2$ the principal stresses, defined as

$$\sigma_1 = -\frac{(\sigma_{11}+\sigma_{22})}{2} + \sqrt{\left[\frac{\sigma_{11}-\sigma_{22}}{2}\right]^2 + \sigma_{12}^2}, \quad \sigma_2 = -\frac{(\sigma_{11}+\sigma_{22})}{2} - \sqrt{\left[\frac{\sigma_{11}-\sigma_{22}}{2}\right]^2 + \sigma_{12}^2}.$$

As the internal stress is assumed to be homogeneously distributed over the thickness of the ice cover, the momentum, constitutive and damage equations are written in terms of the *internal stress* rather than the *vertically integrated internal stress*. This allows a direct comparison between the local state of stress and the damage criterion.

As thermodynamic processes are not accounted for in the present implementation of the model, the conservation equations for the thickness of the ice-covered portion of the grid cell, $h_{thick} = \frac{h}{A}$, and ice concentration read

$$\frac{\partial h_{thick}}{\partial t} + (\mathbf{u} \cdot \nabla) h_{thick} = 0, \tag{A4}$$

$$\frac{\partial A}{\partial t} + (\mathbf{u} \cdot \nabla) A = -A(\nabla \cdot \mathbf{u}). \tag{A5}$$

for $0 \le A \le 1$. Adjustements to the concentration and thickness are applied when and where $A > 1$ and are given by expressions (6) and (7).

An additional equation handles the transport of the passive field of cohesion, which sets the local value of the damage criterion, with the ice flow:

$$\frac{\partial C}{\partial t} + (\mathbf{u} \cdot \nabla) C = 0. \tag{A6}$$

Together, Eq. (A1) (2 horizontal components), (A2) (3 components), (A3), (A4), (A5), (A6) form a system of nine equations that are solved for the nine variables $\sigma$ (3 components), $\mathbf{u}$ (2 components), $d$, $A$, $h_{thick}$ and $C$.

## A2   Numerical scheme

The system of Eq. (A1) to (A6) is closed by suitable initial and boundary conditions and solved over a domain $\Omega$ and a time $t \in [0, +\infty[$. Initial conditions are given by

$$\mathbf{u}(t=0) = 0 \text{ ms}^{-1} \text{ in } \Omega,$$

$$\sigma(t=0) = 0 \text{ Pa in } \Omega,$$

$$d(t=0) = 1 \text{ in } \Omega,$$

$$A(t=0) = 1 \text{ in } \Omega,$$

$$h_{thick}(t=0) = 1 \text{ m in } \Omega,$$

$$C(t=0) = C_0 \text{ Pa in } \Omega,$$

where $C_0$ is the initial field of cohesion, set as described in section 4. The domain boundary $\partial\Omega$ is partitioned as $\partial\Omega = \Gamma_{in} \cup \Gamma_{left} \cup \Gamma_{out} \cup \Gamma_{right}$. A no-slip condition is applied at the lateral boundaries ($\Gamma_{left}$, $\Gamma_{right}$). The channel is open at its top ($\Gamma_{in}$)

and bottom ($\Gamma_{out}$) boundaries with the Neumann condition $\sigma \cdot \mathbf{n} = 0$. The value of the transported quantities $\mathbf{u}$, $\sigma$, $d$, $A$, $h_{thick}$

and $C$ are prescribed on the *upstream* part of the top and bottom boundaries, $\Gamma_-$, defined as

$$\Gamma_- = \{x \in \Omega; \mathbf{u}(x) \cdot \mathbf{n}(x) < 0\}.$$

These are chosen to represent inflowing undamaged ice, with $d = 1$, $A = 1$, $h_{thick} = 1$ m, $\sigma = 0$ and $C$ randomly drawn from the same uniform distribution prescribed as initial condition. The velocity of the inflowing ice is taken as the ice velocity at the nearest upstream boundary node, $\mathbf{u}_-$. The complete set of boundary conditions writes:

$$\mathbf{u}(t) = 0 \text{ ms}^{-1} \text{ on } \Gamma_{\text{left}} \times ]0, +\infty[ \text{ and } \Gamma_{\text{right}} \times ]0, +\infty[,$$

$$\sigma(t) \cdot \mathbf{n} = 0 \text{ Pa on } \Gamma_{\text{in}} \times ]0, +\infty[ \text{ and } \Gamma_{\text{out}} \times ]0, +\infty[,$$

$$\mathbf{u}(t) = \mathbf{u}_-(t) \text{ Pa on } \Gamma_- \times ]0, +\infty[,$$

$$\sigma(t) = 0 \text{ Pa on } \Gamma_- \times ]0, +\infty[,$$

$$d(t) = 1 \text{ on } \Gamma_- \times ]0, +\infty[,$$

$$A(t) = 1 \text{ on } \Gamma_- \times ]0, +\infty[,$$

$$h_{thick}(t) = 1 \text{ m on } \Gamma_- \times ]0, +\infty[,$$

$$C(t) = C_0 \text{ Pa on } \Gamma_- \times ]0, +\infty[.$$

### A2.1 Time discretization

Let $\Delta t > 0$ be the model time step and $t_n = n\Delta t$, $n \geq 0$. The system of equations is discretized in time as follow.

– A semi-implicit scheme is used for the momentum equation: the internal stress term is discretized using an implicit scheme, the water drag term is linearized as $\tau_{\mathbf{w}} = \rho_w C_{dw} |\mathbf{u}^n| \mathbf{u}^{n+1}$ and ice thickness and concentration are both taken

at the $n^{th}$ time step. The total derivative of the ice velocity $\frac{D\mathbf{u}}{Dt} = \frac{\partial \mathbf{u}}{\partial t} + (\mathbf{u} \cdot \nabla)\mathbf{u}$ is approximated using a Lagrange-Galerkin method as

$$\frac{D\mathbf{u}}{Dt}(t_{n+1}, x) = \frac{\mathbf{u}(t_{n+1}, x) - \mathbf{u}(t_n, X_n(x))}{\Delta t} + O(\Delta t) \tag{A7}$$

where $O(\Delta t)$ denotes the neglected higher order terms and $X_n(x)$ is the position of a particle at time $t_n$ that is at the position $x$ at time $t_{n+1}$ and is transported by the velocity field $\mathbf{u}^n$. The first-order Euler approximation for this position

is

$$X_n(x) \approx x - \Delta t \, \mathbf{u}^n(x).$$

The second term on the right hand side of (A7) is denoted $\mathbf{u}^n \circ X^n$ (Saramito, 2013a).

– A semi-implicit scheme is used for the constitutive equation in which the advection, rotation and deformation terms are estimated using the fields of velocity and internal stress at the $n^{th}$ model time step. The ice concentration is taken at the

$n^{th}$ time step.

- The damage evolution equation is discretized using an explicit scheme in which the damage term is evaluated using the states of stress at the $(n+1)^{th}$ time step and the damage criterion ($\sigma_c$ and $\sigma_t$) at the $n^{th}$ time step.

- The cohesion, ice thickness and concentration equations use an explicit scheme and the value of the ice velocity at the $(n+1)^{\text{th}}$ time step.

Hence a semi-implicit, first order discretization of the problem reads:

$(P)$ *For all $n \geq 0$, with $\mathbf{u}^n$, $\sigma^n$, $d^n$, $A^n$, $h_{thick}^n$, $C^n$ known, find $\mathbf{u}^{n+1}$, $\sigma^{n+1}$, $d^{n+1}$, $A^{n+1}$, $h_{thick}^{n+1}$ and $C^{n+1}$ such that*

$$\rho h_{thick}^n A^n \frac{\mathbf{u}^{n+1} - \mathbf{u}^n \circ X^n}{\Delta t} = A^n \left( \tau_{\mathbf{a}} - \rho_w C_{dw} |\mathbf{u}^n| \mathbf{u}^{n+1} \right) + \nabla \cdot (h_{thick}^n A^n \sigma^{n+1})$$

$$\lambda^0 (d^{n+1})^{\alpha-1} \left[ \frac{\sigma^{n+1} - \sigma^n}{\Delta t} + (\mathbf{u}^n \cdot \nabla)\sigma^n + \beta_a (\nabla \mathbf{u}^n, \sigma^n) \right] + \sigma^{n+1}$$

$$= \eta^0 (d^{n+1})^\alpha \exp[-c^*(1 - A^n)] \mathbf{K} : D(\mathbf{u}^{n+1})$$

$$\frac{d^{n+1} - d^n}{\Delta t} + (\mathbf{u}^n \cdot \nabla)d^n = \left( \min\left[ 1, \frac{\sigma_t^n}{\sigma_2^{n+1}}, \frac{\sigma_c^n}{\sigma_1^{n+1} - q\sigma_2^{n+1}} \right] - 1 \right) \frac{1}{t_d} d^n + \frac{1}{t_h}, \ 0 < d^{n+1} \leq 1$$

$$\frac{h_{thick}^{n+1} - h_{thick}^n}{\Delta t} + (\mathbf{u}^{n+1} \cdot \nabla)h_{thick}^n = 0$$

$$\frac{A^{n+1} - A^n}{\Delta t} + (\mathbf{u}^{n+1} \cdot \nabla)A^n = -A^n (\nabla \cdot \mathbf{u}^{n+1})$$

$$\frac{C^{n+1} - C^n}{\Delta t} + (\mathbf{u}^{n+1} \cdot \nabla)C^n = 0$$

*with*

$$\mathbf{u}^{n+1} = 0 \text{ on } \Gamma_{\text{left}} \cup \Gamma_{\text{right}}$$

$$\sigma^{n+1} \cdot \mathbf{n} = 0 \text{ on } \Gamma_{\text{in}} \cup \Gamma_{\text{out}}$$

$$\mathbf{u}^{n+1} = \mathbf{u}_-^{n+1} \text{ on } \Gamma_-$$

$$\sigma^{n+1} = 0 \text{ on } \Gamma_-$$

$$d^{n+1} = 1 \text{ on } \Gamma_-$$

$$A^{n+1} = 1 \text{ on } \Gamma_-$$

$$h^{n+1} = 1 \text{ on } \Gamma_-$$

$$C^{n+1} = C_0 \text{ on } \Gamma_-.$$

These equations are coupled and hence, the problem is nonlinear. To solve it we use a decoupled semi-implicit scheme that divides it into a set of smaller subproblems at each time step as follow. (1) A fixed point algorithm is used in which the momentum and constitutive equations are first solved simultaneously, and the damage equation solved exactly. (2) The value of $d$ is then updated in the constitutive equation and the two computations are iterated until the residual of the constitutive

25 equation drops below a prescribed tolerance or a prescribed maximum number of iterations. (4) The conservation equations for the ice thickness and concentration are solved using $\mathbf{u}^{n+1}$ and both fields are adjusted for mechanical redistribution. (5) The cohesion transport equation is then solved using $\mathbf{u}^{n+1}$ and the local damage criterion is updated.

Using the superscript $k \geq 0$ for the fixed point sub-iterations, the fixed point algorithm for decoupling the momentum, constitutive and damage equations reads:

*When $k = 0$, let $\left(\sigma^{n+1,0}, \mathbf{u}^{n+1,0}, d^{n+1,0}\right) = (\sigma^n, \mathbf{u}^n, d^n)$,*

*For all $k \geq 0$, by recurrence, assume that $(\sigma^{n+1,k}, \mathbf{u}^{n+1,k}, d^{n+1,k})$ are known.*

– *(P1) Find $\sigma^{n+1,k+1}$ and $\mathbf{u}^{n+1,k+1}$ such that*

$$\rho h_{thick}^n A^n \left[\frac{\mathbf{u}^{n+1,k+1} - \mathbf{u}^n \circ X^n}{\Delta t}\right] = A^n \left(\tau_{\mathbf{a}} - \rho_w C_{dw} |\mathbf{u}^n| \mathbf{u}^{n+1,k+1}\right) + \nabla \cdot (h_{thick}^n A^n \sigma^{n+1,k+1}) \tag{A8a}$$

$$\lambda^0 (d^{n+1,k})^{\alpha-1} \left[\frac{\sigma^{n+1,k+1} - \sigma^n}{\Delta t} + (\mathbf{u}^n \cdot \nabla)\sigma^n + \beta_a(\nabla \mathbf{u}^n, \sigma^n)\right] + \sigma^{n+1,k+1}$$

$$= \eta^0 (d^{n+1,k})^\alpha \exp[-c^*(1 - A^n)] \mathbf{K} : D(\mathbf{u}^{n+1,k+1}) \tag{A8b}$$

$$\mathbf{u}^{n+1,k+1} = 0 \text{ on } \Gamma_{\text{left}} \cup \Gamma_{\text{right}} \tag{A8c}$$

$$\sigma^{n+1,k+1} \cdot \mathbf{n} = 0 \text{ on } \Gamma_{\text{in}} \cup \Gamma_{\text{out}} \tag{A8d}$$

$$\mathbf{u}^{n+1,k+1} = \mathbf{u}_-^{n+1,k+1} \text{ on } \Gamma_- \tag{A8e}$$

$$\sigma^{n+1,k+1} = 0 \text{ on } \Gamma_-. \tag{A8f}$$

– *(P2) Find $d^{n+1,k+1}$, such that $0 < d^{n+1,k+1} \leq 1$ and*

$$\frac{d^{n+1,k+1} - d^n}{\Delta t} = \left(\min\left[1, \frac{\sigma_t^n}{\sigma_2^{n+1,k+1}}, \frac{\sigma_c^n}{\sigma_1^{n+1,k+1} - q\sigma_2^{n+1,k+1}}\right] - 1\right)\frac{1}{t_d} d^n, \tag{A9a}$$

$$d^{n+1,k+1} = 1 \text{ on } \Gamma_-. \tag{A9b}$$

– *Stopping criterion : compute*

$$\text{res}_\sigma = \left| \lambda^0 (d^{n+1,k+1})^{\alpha-1} \left[\frac{\sigma^{n+1,k+1} - \sigma^n}{\Delta t} + (\mathbf{u}^n \cdot \nabla)\sigma^n + \beta_a(\nabla \mathbf{u}^n, \sigma^n)\right] + \sigma^{n+1,k+1} \right.$$

$$\left. - \eta^0 (d^{n+1,k+1})^\alpha \exp[-c^*(1 - A^n)] \mathbf{K} : D(\mathbf{u}^{n+1,k+1}) \right|$$

*If $\text{res}_\sigma < \text{tol}$ then*

*set $\left(\mathbf{u}^{n+1}, \sigma^{n+1}, d^{n+1}\right) = \left(\mathbf{u}^{n+1,k+1}, \sigma^{n+1,k+1}, d^{n+1,k+1}\right)$.*

*stop iteration in index $k$.*

Subproblem (P1) is solved in two steps. From (A8b), the following explicit expression for $\sigma^{n+1,k+1}$ in obtained in terms of the unknown $\mathbf{u}^{n+1,k+1}$ and other known variables:

$$\sigma^{n+1,k+1} = K\left[\lambda^0 \left(d^{n+1,k}\right)^{\alpha-1}\left(\frac{\sigma^n}{\Delta t} - (\mathbf{u}^n \cdot \nabla)\sigma^n - \beta_a(\nabla\mathbf{u}^n, \sigma^n)\right) + \eta^0\left(d^{n+1,k}\right)^{\alpha}\exp[-c^*(1-A^n)]\mathbf{K} : D(\mathbf{u}^{n+1,k+1})\right],$$

(A10)

where $K = \left[\lambda^0\left(d^{n+1,k}\right)^{\alpha-1}\frac{1}{\Delta t} + 1\right]^{-1}$. Substituting for this expression in (A8a) leads to an expression of (A8b) with $\mathbf{u}^{n+1,k+1}$ as the only unknown. This linear equation is solved, after discretization with respect to space, by a direct method. Then, $\sigma^{n+1,k+1}$ is computed explicitly from (A10). Subproblem (P2) leads to an explicit computation of $d^{n+1,k+1}$.

The machine used for these simulations allows reaching a value of the residual of the constitutive equation, $res_\sigma$, on the order of $10^{-12}$. In the simulations presented here, the tolerance, tol, was set to $10^{-7}$, which allowed $res_\sigma$ to drop by 4 to 9 orders of magnitude relative to its initial value through the iterative fixed point algorithm. A maximum number of sub-iteration of 10 was imposed. In these simulations, the number of instances for which the residual $res_\sigma$ did not drop below the fixed tolerance in the prescribed number of fixed point iteration represent 2% or less of the total number of model time steps. Imposing a larger maximum number of subiterations (20, 40, 50) reduces this percentage but has shown to have no effect on the results.

Using the superscript $'$ for the variables *before* mechanical redistribution, the ice thickness and concentration are obtained from

(P4) Find $A'$ and $h'_{thick}$ such that

$$\frac{h'_{thick} - h^n_{thick}}{\Delta t} + (\mathbf{u}^{n+1} \cdot \nabla)h^n_{thick} = 0$$
$$\frac{A' - A^n}{\Delta t} + (\mathbf{u}^{n+1} \cdot \nabla)A^n = -A^n(\nabla \cdot \mathbf{u}^{n+1})$$
$$h'_{thick} = 1 \text{ on } \Gamma_-$$
$$A' = 1 \text{ on } \Gamma_-.$$

and

$$h^{n+1}_{thick} = h'_{thick} + \max[0, (A'-1)]h'_{thick},$$
$$A^{n+1} = A' - \max[0, (A'-1)].$$

The transport equation for the cohesion is solved as

(P5) Find $C^{n+1}$ such that

$$\frac{C^{n+1} - C^n}{\Delta t} + (\mathbf{u}^{n+1} \cdot \nabla)C^n = 0,$$
$$C^{n+1} = C_0 \text{ on } \Gamma_-.$$

The damage criterion, i.e., $\sigma_c^{n+1}$ and $\sigma_t^{n+1}$, is then obtained directly from Eq. (1) and (2) and the time step is complete.

While the advection, rotation and deformation terms in the momentum, constitutive and damage equations could all be updated in the fixed point iteration, we find that this has no impact on the results presented here since the time step employed in the simulations is taken very small (6 s for $\Delta x = 3$ km and $4$ s for $\Delta x = 2$ km) in order to simulate the propagation of damage over the ice cover with the highest possible resolution (see Dansereau et al. (2016)). Similarly, the cohesion, ice thickness and concentration could also be solved for using an implicit scheme and could be updated as part of the fixed point iteration. However, because variations in the mechanical parameters associated with changes in $C$, $h$ and $A$ are slow compared to the changes due to damaging, we find that this does not significantly impact our model solution either.

### A2.2  Space discretization

The velocity field is discretized in space by a continuous first order polynomial finite element method while all others variables are discretized by a piecewise constant discontinuous approximation. The transport operator $\mathbf{u}.\nabla$ (as well as the terms for the rotation and deformation of the stress tensor) are discretized by a discontinuous Galerkin method, that coincides, when using piecewise constant approximation, with the usual upwind finite volume scheme on unstructured meshes.

*Author contributions.* Véronique Dansereau wrote the paper, developed the Maxwell-EB model and performed the simulations. This work was supervised by former Ph.D. advisors Jérôme Weiss, Pierre Saramito, Philippe Lattes and Edmond Coche.

*Acknowledgements.* The financial support of TOTAL EP RECHERCHE DEVELOPPEMENT is gratefully acknowledged. We also thank Christian Haas and Kaj Riska for useful discussions on this work.

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

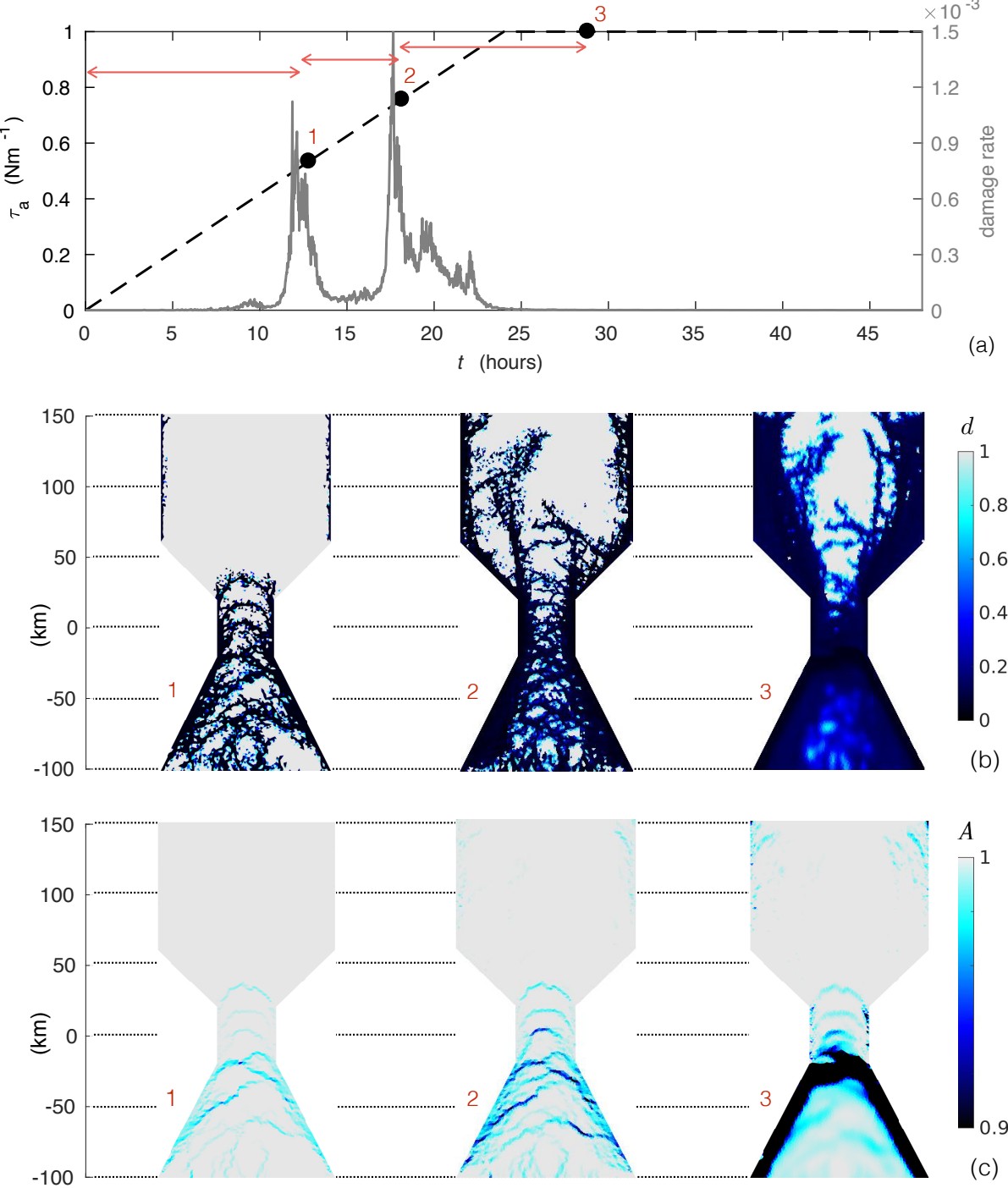

**Figure 4.** (a) Time series of the wind forcing (dashed curve) and of the damage rate (solid grey curve) in an idealized channel simulation using $C_{min} = 20$ kPa. Instantaneous spatial distribution of (b) the level of damage and (c) ice concentration at the times indicated by the numbers 1, 2 and 3 on the time series of panel (a).

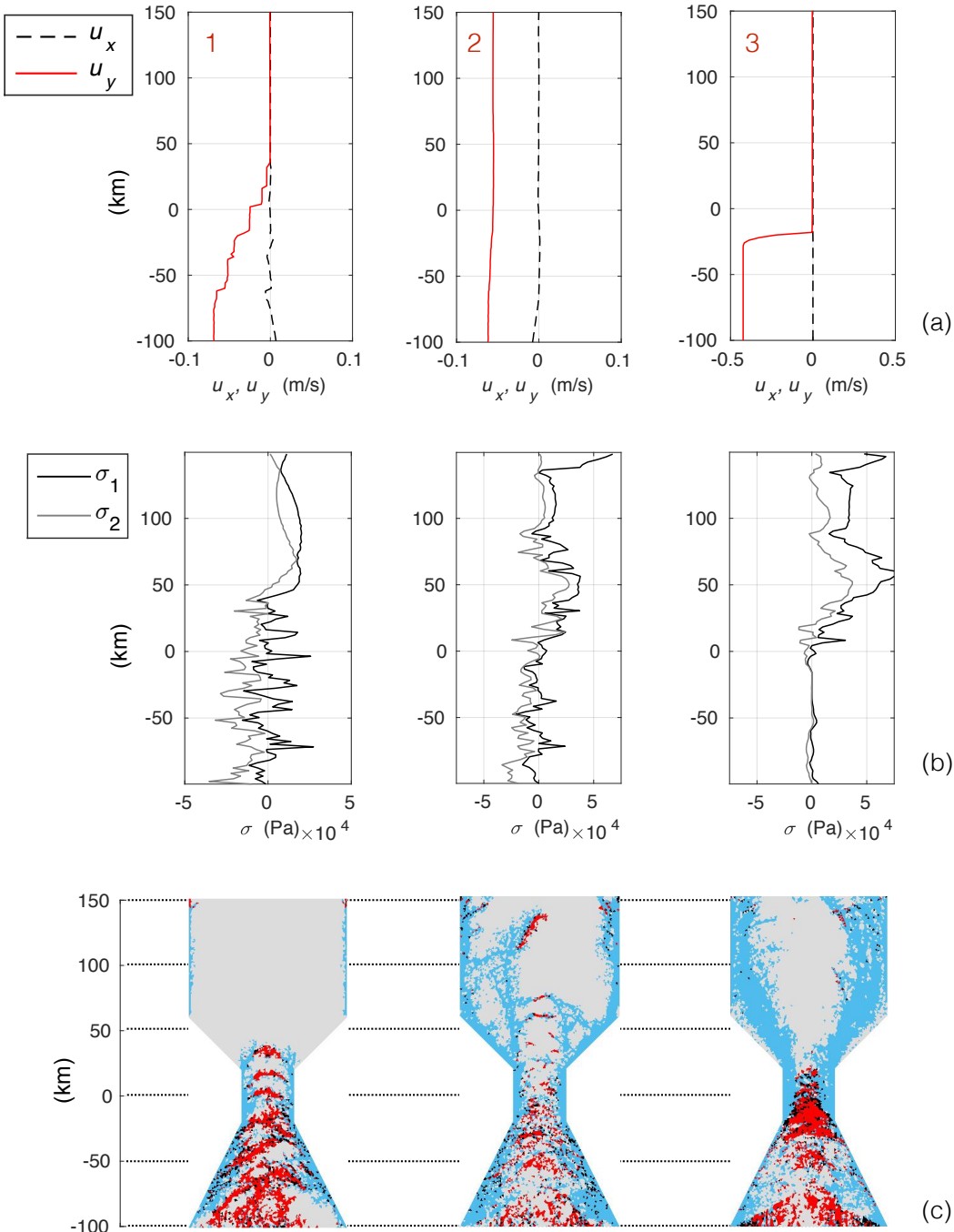

**Figure 5.** Profiles of (a) the instantaneous $x-$component (dashed curves) and $y-$component (red curves) of the ice velocity and (b) the instantaneous principal stress components $\sigma_1$ (black curves) and $\sigma_2$ (grey curves) along the central meridional axis of the channel at the three stages indicated on Fig. 4a ($C_{min} = 20$ kPa). (c) Spatial distribution of all the elements that have exceeded the Mohr-Coulomb (blue) the tensile (red) or both (black) damage criteria during the 3 periods of time, between the beginning of the simulation and times numbered 1, 2 and 3, indicated by the red arrows on 5a.

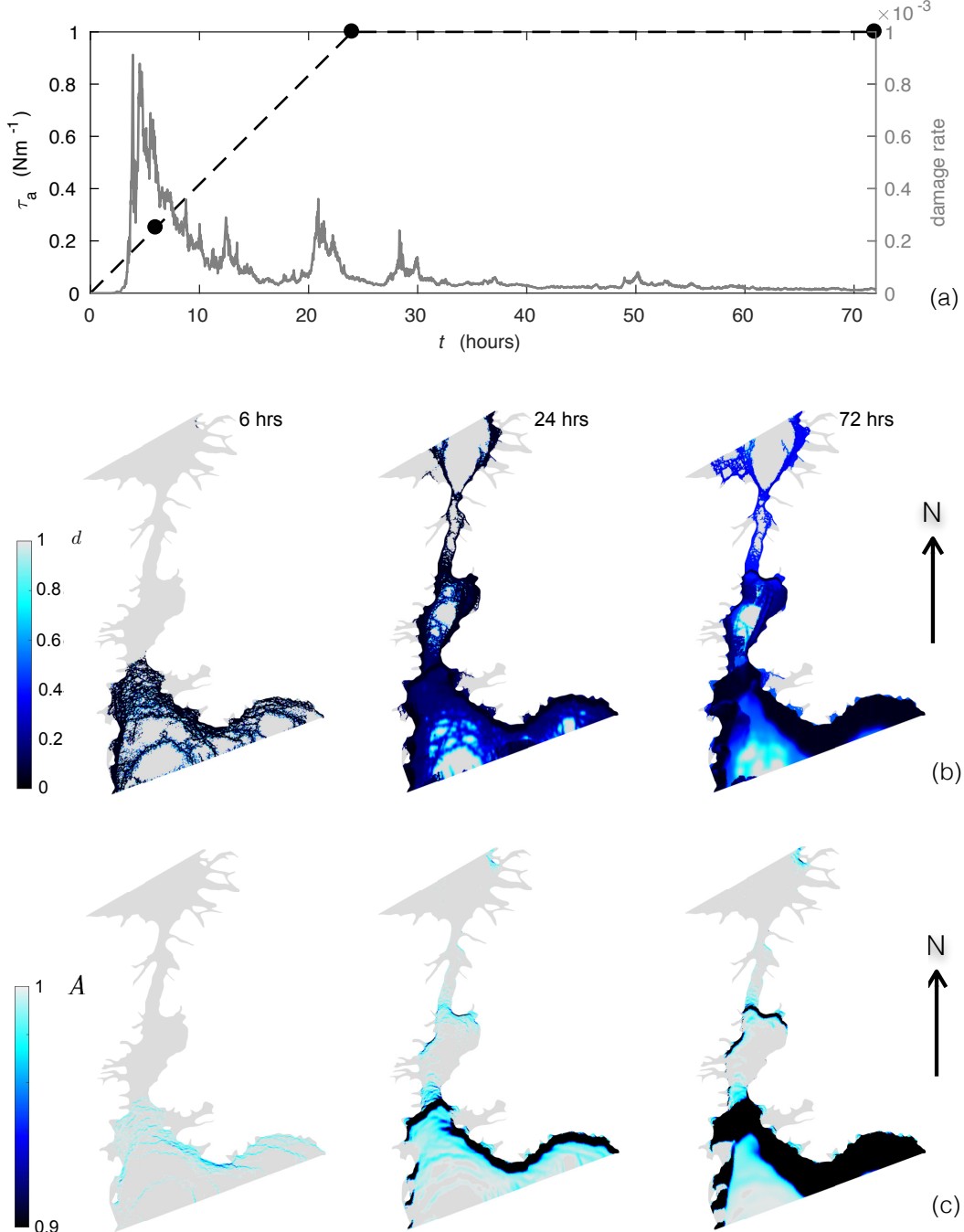

**Figure 6.** (a) Time series of the wind forcing (dashed curve) and of the damage rate (solid grey curve) over the realistic Nares Strait in a simulation using $C_{min} = 20$ kPa. Instantaneous fields of the simulated (b) level of damage and (c) ice concentration at $t = 6, 24$ and $72$ hours.

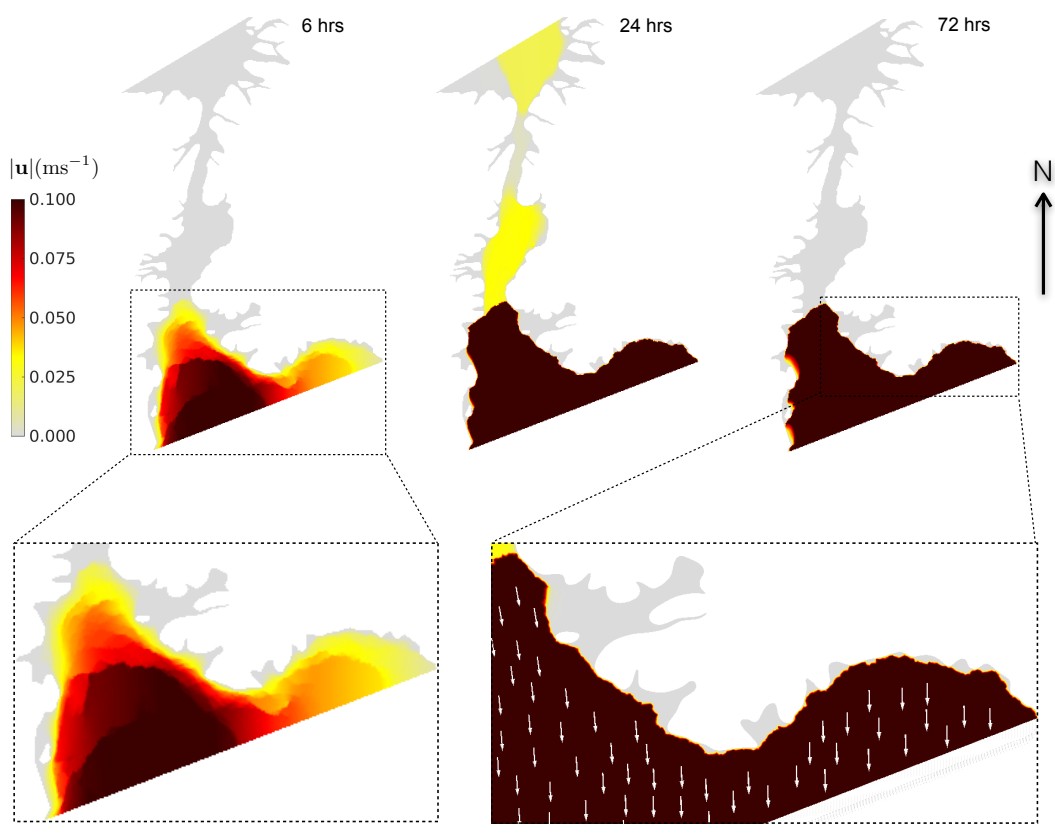

**Figure 7.** . Instantaneous fields of the simulated ice drift speed over the realistic Nares Strait at $t = 6, 24$ and 72 hours ($C_{min} = 20$ kPa). The lower panels are zoom-ins of the 6 and 72 hours drift speed fields, showing (left panel) the division of the ice cover downstream of Kane Basin into ice floes with piecewise constant velocities and (right panel) the sharp gradient in ice velocity between free drifting ice and regions of stagnant, landfast ice attached to the coast in narrow fjords and upwind of islands (the white arrows indicate the drift direction at evenly-spaced element nodes).

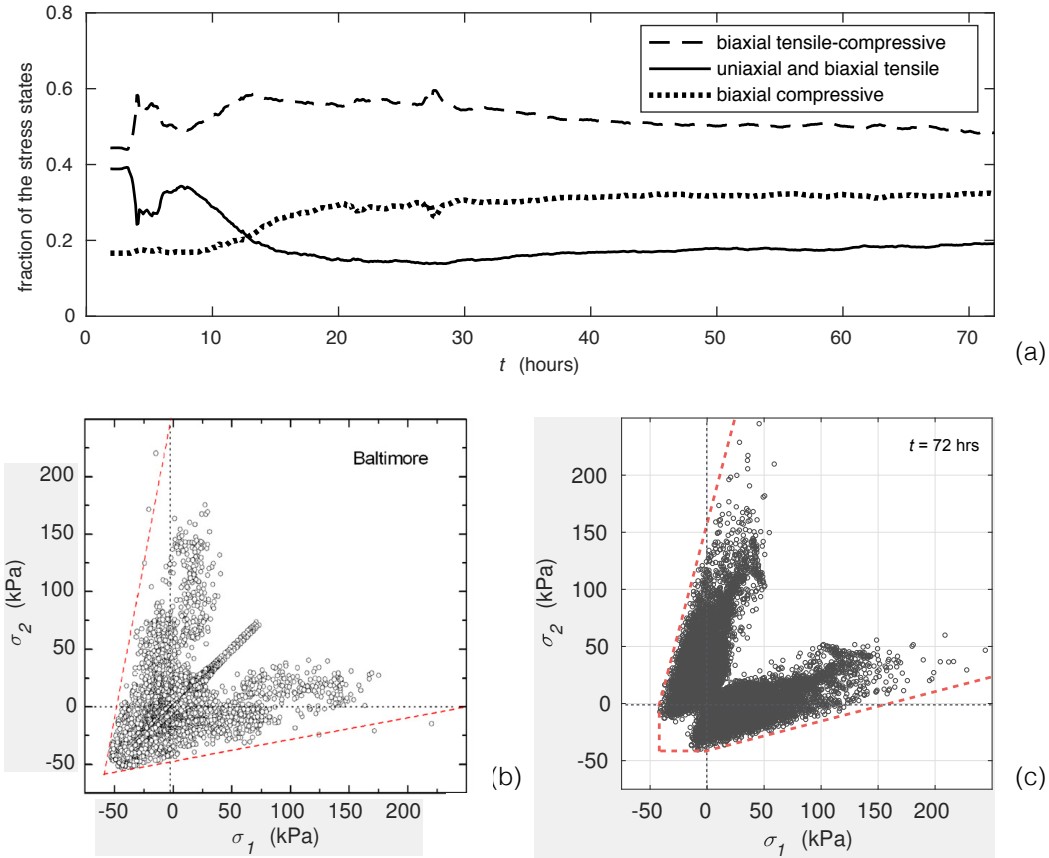

**Figure 8.** (a) Time series of the proportion of tensile (uniaxial and biaxial, solid curve), biaxial tensile-compressive (dashed curve) and biaxial compressive (dotted curve) stresses over the realistic Nares Strait in the simulation using $C_{min} = 20$ kPa. (b) Stress states in principal stress space recorded by one stress-meter (one measurement per hour) during the SHEBA experiment in the Beaufort sea (from mid-October, 1997 to end of June, 1998). The dashed red lines represent the Coulombic branches of a failure envelope for $\sigma_c = 250$ kPa and $\mu = 0.9$. *From Weiss and Schulson (2009).* (c) Instantaneous stress states simulated with the Maxwell-EB model using $C_{min} = 20$ kPa after the formation of the stable ice bridge downstream of Kane Basin ($t = 72$ hours). The dashed red lines represent the damage criterion corresponding to the highest value of cohesion over the domain ($\sigma_c \approx 154$ kPa, $\sigma_t = 42$ kPa, $\mu = 0.7$). In both cases, $(\sigma_1, \sigma_2)$ and $(\sigma_2, \sigma_1)$ are plotted, resulting in symmetry about the axis $\sigma_1 = \sigma_2$.

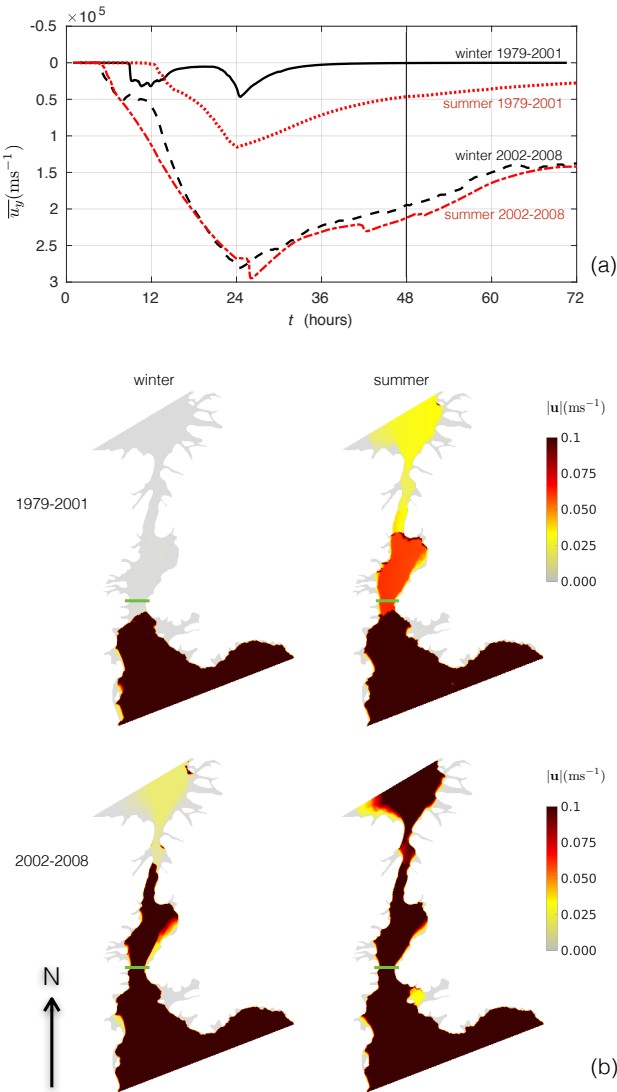

**Figure 9.** (a) Time series of the meridional component of the simulated ice drift velocity, $u_y$, averaged zonally across the constriction point between Kane Basin and Smith Sound, in the 1979-2001 (stronger ice cover), 2002-2008 (weaker ice cover) and winter ($t_h \approx 5.7$ days) and summer ($t_h = 365$ days) scenarios. (b) Instantaneous field of the ice drift speed, $|\mathbf{u}|$, over Nares Strait after 48 hours of simulation in the four scenarios. The green lines show the location of the zonal cross-section used for the calculation of $\overline{u_y}$ in (a). Everywhere over the domain and at all times, the drift velocity is dominated by its meridional component and is southward.

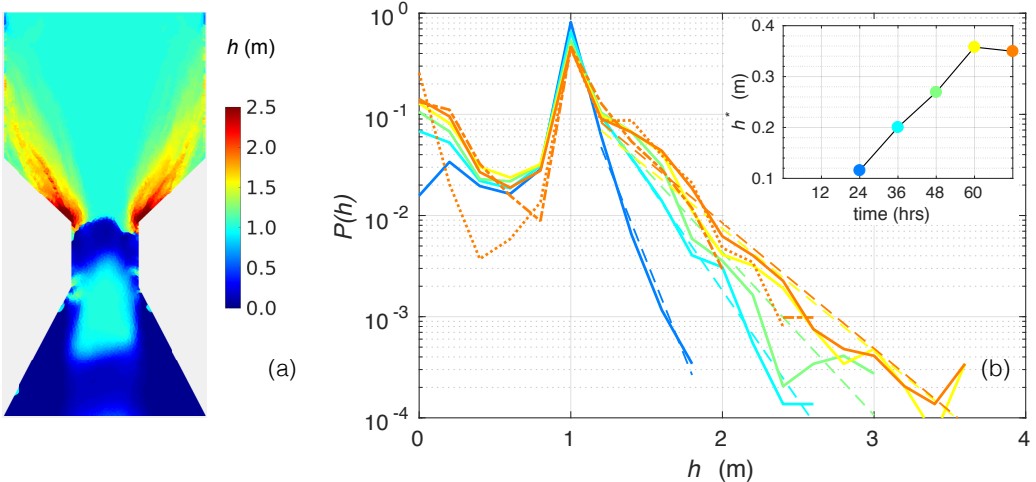

**Figure 10.** (a) Instantaneous field of the mean ice thickness, $h$, after 3 days of simulation in an idealized channel simulation. The value of $C_{min}$ is 10 kPa and self-obstruction to flow (i.e., a stable ice bridge) does not occur. (b) Probability density function of the instantaneous simulated ice thickness $P(h)$ at different times (coloured lines). A bin width of 0.2 m is used. Note the semi-logarithmic axis. The (orange) dotted and dotted-dashed lines represents $P(h)$ calculated at $t = 3$ days in simulations using the same idealized domain and boundary conditions but a decreased spatial resolution of $\Delta x = 4$ km and 8 km respectively. The inset shows the values of $h^*$ calculated at different times using a linear regression of $\ln P(h)$ for $h > 1.0$ m.

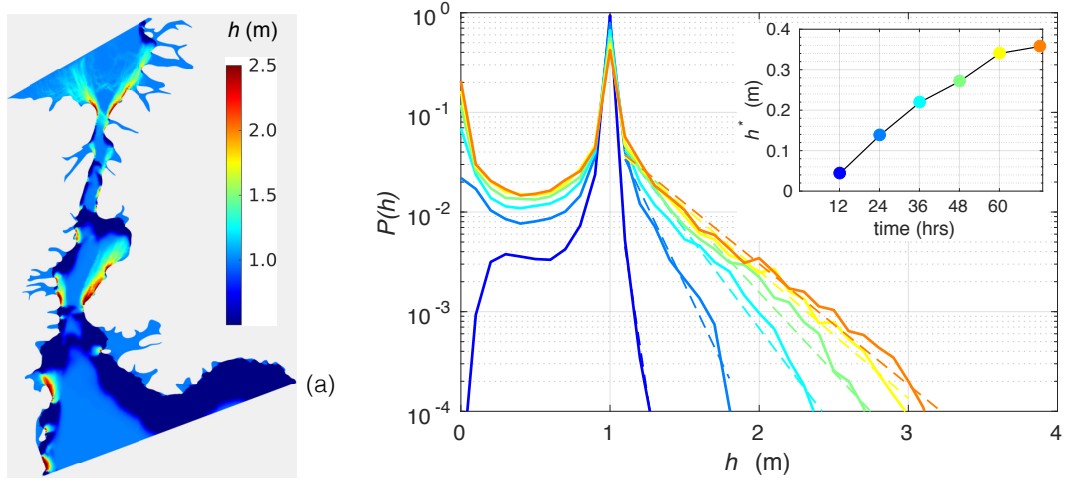

**Figure 11.** (a) Instantaneous field of the mean ice thickness, $h$, after 3 days in the realistic simulation of Nares Strait. The value of $C_{min}$ is 10 kPa and a stable ice bridge does not occur. (b) Probability density function of the instantaneous simulated ice thickness, $P(h)$ at different times (coloured lines). A bin width of 0.2 meters is used. Note the semi-logarithmic axis. The inset shows the values of $h^*$ calculated at different times using a linear regression of $\ln P(h)$ for $h > 1.0$ m. The percentage of the variance ($R$) explained by this fit is $> 95\%$.