# Peer review of "Ice bridges and ridges in the Maxwell-EB sea ice rheology"

_The Cryosphere, 2016_

## Referee Comment (RC1) · Anonymous Referee #1 · 18 Jan 2017

This paper describes numerical experiments with a sea-ice model that uses the "Maxwell Elasto-Brittle" rheology. The details of the Maxwell E-B rheology are described in a previous paper (Dansereau et al, The Cryosphere, 2016). In this paper, two simulations are run: one with idealized geometry of flow through a constriction, and one with the actual geography of Nares Strait. The model reproduces the ice arch across the southern end of Kane Basin that is often observed in satellite imagery, leading to the formation of the North Water polynya to the south. The model also reproduces the presence of landfast ice in small inlets and fjords, sharp gradients in ice velocity across leads, the exponential tail of the sea-ice thickness distribution, and stress states in reasonable agreement with observations.

This paper is clearly written (minor exceptions noted below) and describes results that should be of interest to sea-ice modelers and general circulation modelers. I recom-

mend publication after minor (mainly technical) revisions noted below. These comments are in page order, not in order of importance.

Minor and Technical Comments

Abstract, first sentence. You need to tell the reader what "EB" is and what the model does before launching into a discussion of its implementation. Like this: "The Maxwell Elasto-Brittle (EB) model uses a sea-ice rheology that allows tensile stress... blah blah [one sentence about the model]. This paper presents a first implementation of the..."

Abstract, lines 6-8. This sentence is a bit awkward. I suggest: "In agreement with observations, the model captures the propagation of damage..." etc. (and delete "are all represented" at the end of the sentence).

Abstract, last sentence. "weakening of the ice cover" and "shorter lifespan of ice bridges" – over what time period? You should add something like "in the 2000s relative to the 1980s and 1990s". And then "with implications in terms of increased ice export" – say that this would be expected because sea ice is expected to continue thinning in the future. In fact, maybe this is a positive feedback on the loss of ice: as it weakens, it drains from the Arctic faster, which weakens it further.

Page 1, line 15. "expenses" should be "expanses"

Page 1, line 21. "May 2005" should be July 2010.

Page 2, line 12. Change "allowed demonstrating" to "demonstrated"

Page 2, lines 18-20. "shorter lifespan of ice arches" and "increased ice export" – over what time period?

Figure 1. The dates on the images are in the format DAY-MONTH-YEAR. Is that standard for The Cryosphere? It might confuse U.S. readers, who often use MONTH-DAY-YEAR.

Figure 1, caption. Put "(MODIS)" before the word "reflectance", not after.

Page 4, line 10. 0.32 degrees is not 4-10 km.

Page 4, lines 21-24. For the statement about the sea-ice thickness PDF you can cite: Lindsay, R. W. 2013. Unified Sea Ice Thickness Climate Data Record, 1975-2012. Boulder, Colorado USA: National Snow and Ice Data Center. http://dx.doi.org/10.7265/N5D50JXV. Web site: http://nsidc.org/data/docs/noaa/g10006-unified-sea-ice/

Page 5, line 3. Change "pioneer" to "pioneering"

Page 5, line 9. "dubiously" – Why it is dubious to base the modelling framework on energy conservation?

Page 5, line 18. There is an error in the citation of Hibler's 1980 paper. William D. Hibler III should be "Hibler", not "III". Also on page 23, line 10, the author is Hibler, not III.

Page 5, line 19. "sensible" should be "sensitive"

Page 6, line 5. Instead of "recall", better to write "review"

Page 6, line 7. Instead of "passage", I think "connection" is the intended meaning

Page 6, line 9. After "elastic modulus" write "(E)" because it is used three lines later in the definition of lambda.

Page 6, equations (1) and (2) – you need to say that "mu" is the internal friction coefficient.

Page 6, lines 27-28. About healing: healing allows the level of damage to DECREASE – more healing, less damage. But the damage parameter d INCREASES as healing increases (d=1 is undamaged). So you have to be careful when describing healing. Verbally, healing means a SMALLER AMOUNT of damage. But numerically, healing means a LARGER VALUE of d.

Page 7, Figure 2, caption. Several things: – It is awkward to have complicated equations in the caption. Furthermore, the components sigma11, sigma12, and sigma22 are not used anywhere else in the paper, so I think it would be OK to delete the equations entirely. Otherwise, they should probably go in the main text. That's just a suggestion, there is nothing wrong with the current presentation. – Second sentence: "The thin solid lines represent the damage criterion in the case of C=0." To make this clearer, consider: "The thin solid lines radiating from the origin represent the damage criterion in the case of no cohesion (C=0)." – What is the shaded region in the figure?

Page 8, line 7. Change "assimilated to" to "simulated as"

Page 8, line 27. I think max[0,(1-A)] should be max[0,A-1]. This is supposed to describe "the excess concentration" when A > 1. If A > 1 then max[0,1-A] = 0, which doesn't make sense. The excess concentration should be max[0,A-1].

Page 9, equation 7. This equation doesn't make sense either. When A > 1, h+ = 0. I think this equation should be h+ = max[0,A-1]h.

Page 9, line 18. Change "of being" to "to be". Change "sturdy" to "steady"

Page 9, line 19. Capitalize "Lincoln Sea"

Page 10, line 27. Acceleration and advection terms are neglected in equation 3. Does this mean the solution is steady state, with no time dependence?

Page 11, Figure 3 caption. Delete the first "(a)" because the caption later refers to (a) and (b).

Page 11, line 11. "quenched disorder" – this needs a reference.

Page 11, line 16. Beaufort SEA. Also, Weiss et al (2007) is cited in connection with Figure 8(a), but the figure caption cites Weiss and Schulson (2009).

Page 12, line 10. Delete "the upstream part of", so the sentence reads, "...prescribed on GAMMAin and GAMMAout..."

Page 12, lines 14-15. "The equations of motion... Galerkin methods are used to handle advective processes" – but the advection terms in equation 3 are neglected. So I don't understand this. Does it refer to advection in equations 4 and 5?

Page 14, line 2. "deformation rates" – I think this should be "high deformation rates"

Page 14, line 3. "downstream of the channel" should be "downstream of the constriction"

Page 14, Figure 4. This figure should be bigger, so that it fills the full width of the page. It's a bit difficult to see the peaks of wind stress in panel (a) – bigger would be better.

Page 15, line 23. I think "Fig 4b and 4c" should be "Fig 5b and 5c"

Page 15, line 27. I think "Fig 4b" should be "Fig 5b"

Page 16, line 24. The text says "Weiss et al (2007)" but the Figure 8 caption says "Weiss and Schulson (2009)"

Page 18, line 2. "ice landfast" should be "landfast ice"

Page 18, line 8-9. How does the ice become weaker without thinning? Is it because the ice strength is temperature-dependent, so increasing the temperature would make it weaker? Is it because the composition of the ice could change, such as increased salinity?

Page 23, line 10. The first author is Hibler, W.D. III

Page 25, line 27. Weiss and Dansereau 2016 – is the DOI correct?

Page 26, Figure 5 caption. At the end of the last sentence, "red arrows on (a)" should be "red arrows on 4(a)"

Page 27, Figure 6 caption. Delete the first "(a)" because the caption later refers to (a) and (b).

Page 29, Figure 8 caption. Delete the first "(a)". Capitalize "Beaufort Sea"

[Figure]

Page 31, Figure 10 caption:

– line 2: "is of 10" should be "is 10"

– lines 5 and 6: "insert" should be "inset"

– I recommend deleting these two sentences: "The tail of the distribution..." and "The temporal evolution..." because the last sentence of the caption explains the same thing as these sentences.

Page 31, Figure 11 caption:

– line 2: "is of 10" should be "is 10"

– line 5: "insert" should be "inset"

Please note: I have not flagged all the minor grammatical corrections that should be made by an editor, for example:

Page 9, line 10. "is of 100%" – delete "of"

Page 9, line 26. such AN ice bridge

Page 10, line 8. "in Cartesian" should be "to Cartesian"

Page 10, line 9. Delete "is"

Page 11, line 8. employed IN the simulations

Page 11, line 12. "by randomly by" – delete second "by"

...and so on...

---

## Referee Comment (RC2) · Anonymous Referee #2 · 1 Feb 2017

**Review: Ice bridges and ridges in the Maxwell-EB sea ice rheology**

**Summary & General remarks:**

The paper presents simulations of ice bridges with the Maxwell-EB rheology in idealised and realistic geometries. The authors evaluate the model by analysing the simulated stress states associated with the formation and persistence of ice bridges. In different experiments, they study the effect of weaker ice in recent years on the flow storage and the sea ice export. Finally, power-law scaling of the ice thickness distribution produced by ridging ice at the margins of the bridges is presented. The paper is well written and presents interesting, new results that contribute to the current research on sea ice rheology. Nevertheless, these results generated in this idealised experiments need to be discussed in further detail to back all conclusions that authors

make. This issue lead to the "fair" grade for review criteria 2 (C). In addition, the readability of the model description could be improved. I therefore recommend a major revision on the manuscript.

**Major comments:**

There are many statements that require more discussion and clarification. In some places strong statements are made that not supported by the results of the model simulations. These statements need either be toned down, or sufficient evidence should be provided (which is probably not possible without additional experiments, especially when reference to VP models is made.) For instance, the impact of the simplified model equations, geometry, wind forcing and no thermodynamics is discussed too little, especially when comparing to observations made outside of a channel. The model set-up for the analysis of PDFs of ice thickness facilitates ridging with a high amount of coastal boundaries, low cohesion parameter and strong wind forcing (and a presumably wrong source term in Equation (7)), which is not further discussed. In the experiments with reduced mechanical strength a comparison to observations would be even possible: Are a collapse of ice bridges and no flow storage observed in the recent years? Especially, regarding the comparison to VP-models more caution is required. If no reference VP simulation is performed, the differences of the referred studies need to be considered, as most of the studies do not agree in resolution or region used in the experiments presented here. For example, it is not clear, if VP models at the same resolution would not lead to the same type of "ridging" behavior; as long as it is not clear, it shouldn't be claimed. The detailed overview which points need further discussion are listed below in the minor comments.

In Section 3, a sea-ice model using the Maxwell-EB rheology is outlined. However, due to the reference to individual equations in Dansereau, et al. (2015) the readability is reduced. A summary of the model equations cited from Dansereau et al. (2015) in an appendix would assist the reader. In addition, if you would include the simplifications that are made in your experiments into the model description, it is directly clear to the

reader what equations are solved in your model. For example, the full momentum equations are given in Equation (3); one can easily miss the fact that many terms are not solved for (p10, l.26-27). Here the reduced equations could replace Equation (3), since this is all that is used in the manuscript.

**Minor comments:**

page 1,l.7: "in" -> into

page 1,l.8: "Strait" -> strait

page 1,l.9: "different dynamical behaviours" -> various dynamical behaviour

page 2, l.26: "to" -> and?

page 2, l.32: "This rheological framework typically does not account for (uniaxial or biaxial) tensile strength." Not quite true, there is uniaxial tensile strength in Hibler elliptical yield curve (see your figure 2), but there's usually no isotropic/biaxial tensile strength; adding tensile strength is another option for the VP model and has been used to improve simulations of land fast ice (Lemieux et al 2016, Olason, 2016). Could be mentioned in this context, too.

page 2, l. 30-32: Dumont also used EVP

page 4, l. 13-14: That's a speculation. It would be nicer to actually show that this can work or not work with VP models at high resolution (better than 4km grid spacing).

page 4, l.13: "...(e.g., see Fig. 1b and Sodhi,1977)" wrong citation

page 4, l.20: Section 2.2 "Ice ridges". First paragraph is relevant for the following analysis of the ice thickness distribution. The last three paragraphs can be shortened or discarded, as neither a VP-model nor an ITD is used later on.

page 5, l.17: wrong citation, I guess Hibler, 1980 is meant?

page 5, l.19: "badly" -> poorly

page 6, l. 5: "details" -> detail

page 6, l. 5: "recall" -> repeat?

page 6, l.15: This definition is unfortunate and unintuitive. 0 should be "undamaged" (zero damage) and 1 should "completely damaged" if d is called "damage". d feels more like "integrity" for the material, but it's just terminology . . .

page 6, l. 22: Isn't Equation (2) $\sigma_t = -2C[(\mu^2+1)^{1/2}+\mu]^{-1}$ ? At least in Danserau et al. (2016) Equation (7),(8) and (10) are not consistent and I guess the exponent -1 is missing in Equation (10).

page 6, l. 27-28: Unclear, if it represents refreezing of leads, how can it be independent of pure thermodynamics, please explain/rewrite/elaborate

page 8, l.9: Equation (3); Please also state the simplified version of the equation that is actually used by the model, to prevent misunderstandings.

page 8, l. 21-22: Mechanical redistribution in your formulations is represented by the divergence term, see next comment about Equation (7).

page 9, l. 1, Equation (7) That is wrong: $h$ is defined as the mean thickness (per grid cell area), so something like a volume of ice in the cell. The ice volume does not change if $A > 1$ is reset to $A = 1$, but the (mean) thickness of the thick ice $h_{thick} = h/A$ is increased, and there is no extra contribution to $S_h$. (See also Schulkes (1995), JGR.) This should be corrected in the text and also in the model, if the model actually implements this extra (spurious) tendency in Equation (4).

page 9, l. 4-5, Equation (8) and (9): Just for clarity, a dependence on the thickness as in Hibler (1979) is not needed, as the internal stress is used in the momentum equation?

page 9, l. 8: "widely" -> is widely

page 9, l. 22: "(Kwok et al., 2010))" -> (Kwok et al., 2010)

[Figure]

page 10, l.11: "northerly, wind stress" -> not sure about this: northerly winds, but the stress is acting towards the south, should be made clearer I think.

page 10, l.24: "transport of the cohesion, C" there are no sources and sinks of cohesion? How realistic is that? Please comment and elaborate the cohesion equation in Dansereau et al. (2016), this is not discussed.

page 10, l.32: What is the reference for the Young's modulus? Same as for the Poisson's ratio?

page 11, l.5-8: The physical role of healing is unclear and needs to be explained better. It is clearly connected to the thermodynamics (in contrast to earlier statements in the manuscript) . . . Please elaborate . . .

page 12, l.7: "location of ice bridges is not prescribed" only through the random field of cohesion (the spatial pattern should be shown somewhere). I would like to see simulations with uniform cohesion; the model geometry should be irregular enough to make the model develop ice bridges, etc.

page 12, l.16 (and elsewhere): "(see (Dansereau et al., 2016))" -> (see Dansereau et al., 2016).

page 12, l.18: For the claims made in the introduction and background sections, VP simulations at this resolution are absolutely required (have not been done to my knowledge). Please tone done the statements in the appropriate places.

page 13, l.1-2: please state the range of $\Delta x$

page 13, l.16-17: "(see Dansereau et al. (2016))" -> (see Dansereau et al., 2016)

page 13, l.19-20: I think this statement requires, that you have tried a fully implicit scheme and compared the results. Have you? If not, this statement is no really supported by anything and should be changed.

page 14, l.8-9: Please say, how much the "drift velocity on the order of that associated

with strictly elastic deformations within an undamaged ice cover." really is (in m/s or cm/s or whatever) so that others can compare.

page 14, l.11: "relatively undamaged" -> rephrase to "stagnant ice with low damage" or similar

page 14, l. 20-21: "the width of the distribution of C impacts the rate of propagation of the damage, with the propagation being more progressive for a larger distribution." Since the cohesion appears to be an important parameter, it would be useful to add more information about the choice of C, i.e. the actual distribution of C that is generated (page 11, ll.10) in case the reader would like to reproduce the results.

page 15, l.2: "differs" -> differ

page 15, l.23: "(see Fig. 4b and 4c, panel 3)" Should be Fig. 5b and 5c...

page 15, l.23-26: "This is an important point, as standard viscous-plastic sea ice models do not account for pure uniaxial or biaxial tensile strength and hence would not be able to reproduce the formation of a stable ice arch with self-obstruction to flow under the conditions simulated here." I don't agree: (1) From the figure, the location of the arch is not visible if you mean it is defined by the location of black elements. (2) the details of the yield curve (Figure 2) should not matter, one can tune the elliptic yield curve to resemble the Mohr-Coulomb and tensile failure criteria (see Figure 1 in Lemieux et al, 2016). (3) even without isotropic/biaxial tensile strength, Dumont could simulate arches with VP rheology, so do Losch and Danilov (2012) in similar idealized simulations, even with "a standard VP model" for order 1000 days. (4) why do VP models not account for pure uniaxial tensile strength? I think that this statement needs to change.

page 16, l.24: In this comparison (Figure 8), one might ask why the specific failure curves where chosen differently for the model, when there are estimates for the parameters available ($\sigma_c = 250$, and $\mu = 0.9$). Should be discussed somewhere.

page 17, l.3: "later" -> more recent ?

page 17, l.16: "According with"-> In line with

page 17, l.21: "'differentiated" does not sound right, rephrase if necessary

page 17, l.29-31: "However, in all of the weaker ice cover scenarios (2002-2008 period and/or 30 summer), none of the ice arches formed near the exit of Kane Basin nor secondary arches formed elsewhere sustain the applied wind forcing and all ice bridges eventually collapse." Is there a similar behaviour in observations in this period? Please add a comment.

page 18, l.7: "widely different dynamical behaviours" -> a wide range of dynamical behaviour

page 18, l.7-9: The big question remains: how do you determine the appropriate cohesion? It appears to be vital parameter, similar to P* in Hibler's VP model.

page 19, l.5-6: "A Lagrangian model would perhaps be more suitable to simulate the edge of the detached ice"; or a better advection scheme with less numerical diffusion (i.e. higher order basis functions in your finite element method)

page 19, l.13-14: "Nevertheless, at all times the simulated probability density function is strongly asymmetric, consistent with thickness distributions estimated for sea ice with little history of melting (e.g., Haas, 2009)." Please discuss in how far this special experimental geometry with many coastlines and the low $C_{min}$ is suitable to compare to observations made for open ocean Arctic sea ice as described in Haas (2009).

page 19, l.18: This term (7) is not correct and should not be used. See e.g. Schulkes (1995), JGR, for correct equations and a nice explanation of ridging in general.

page 19, l. 23: "Fig. 11b" -> Fig. 10b 10b

page 20, l. 2-3: "In coupled thermodynamic and dynamic models, a high density of leads is expected to impact the simulated heat fluxes between the atmosphere, the ice and the ocean (Smith et al., 1990)." This is not really a conclusion, but part of a

discussion.

page 20, l. 11-13: "the presence of land fast ice along. . ." This has hardly been discussed and comes as a surprise. Needs more attention in Section 5 if you want to keep this conclusion

page 20, l. 24: "a process that is known to be underestimated in VP models using a two-level scheme" This is new to me. At correspondingly high resolution I would expect a VP model to behave in a similar manner, see also Losch and Danilov (2012), Fig6. which shows very similar ice thickness distribution in a similar channel experiment.

page 20, l.26-28: See above, I don't think, that you can say this, because you'd have to show that the same model configuration with a VP model would not have your thickness distribution. I am pretty sure that you would get a similar result.

page 21, l. 14: "later" -> recent

page 21, l. 33: "Haas, C.: Dynamics Versus Thermodynamics: The Sea Ice Thickness Distribution, p. 638, Wiley-Blackwell, 2009." Please correct citation as book chapter in Sea Ice (eds D. N. Thomas and G. S. Dieckmann)

page 22, l.10: "III, W. D. H.: Modeling a Variable Thickness Sea Ice Cover,. . ." -> wrong name

page 25, l.27: "Weiss, J. and Dansereau, V.: Linking scales in sea ice mechanics, Philosophical Transactions A, pp. –, doi:10.1098/XXXX, 2016." Is this a submitted manuscript? If so it is not properly cited.

page 26, Figure 5: What is $C_{min}$ in this simulations? Did you consider to show a sea ice concentration plot for the idealized experiments as well? That would help to see the arches directly.

page 29, Figure 8: An indication of the probability of single stress states using a colormap or transparency would be helpful, to get an impression how frequently biaxial

tensile states (and all other stress states) occur.

page 31, Figure 10: Why are the PDFs for $\Delta x$ = 4km and 8km given at t=5days, whereas the other results are shown for t=3days?

———————————————

---

## Referee Comment (RC3) · Anonymous Referee #3 · 9 Feb 2017

General comments: This paper aims at examining the validity of the Maxwell-EB sea ice rheology to reproduce the ice bridge phenomena which often appears at the narrow straits like Nares Strait. The major purpose of this study is placed on the first application of this rheology to the geophysical scale sea ice phenomena. Resultantly, they found that this rheology can reproduce the real ice bridges which occurs at the narrow straits. Overall I feel that this paper is well written based on sound science and certainly contributes to the development of sea ice dynamics, especially for the behavior of the brittle sea ice. Therefore, my comments are only minor. General comments area as follows: 1) It would be helpful to explain more about the advantage of the Maxwell-EB rheology compared with the traditional elliptic curve rheology. The authors pointed out the capability to represent the extreme localization of damage and deformation (P4L13). However, there is a possibility that it comes just from the horizontal resolu-

tion of the grid cell in the model. I mean that the traditional plastic rheology might be able to reproduce the phenomena if finer grid cells are used. Thus I want to know why they consider the continuum elasto-brittle rheology should be more appropriate than the continuum viscous plastic rheology for this phenomena, and whether this rheology can be applicable to the general sea ice conditions. 2) Intuitively my feeling is that the floe size distribution of sea ice should also play an important role in the brittle ice rheology. Therefore, in the question at P17L7-8 it would be natural that the change in floe size distribution may also contribute to the phenomena. What do you think? 3) On the whole, I am somewhat concerned about why the authors did not pay so much attention to the horizontal scale. For example, the scales of ice bridges seem to be different depending on the straits. Accordingly the mechanism might be different depending on the regions. Could you explain how the Maxwell-EB rheology influence the results depending on the scales. Although the description on the scale dependence (P19L17-25) is interesting, in general it seems that the localization of deformation depends on the grid cell size. Could you explain why this property is independent of resolution?

Specific points: *(P1L2)"on geophysical scales" I wonder if we can assume ice bridges and ridges to be on a geophysical scale. It would be preferable to describe the specific phenomena like "ice bridges on a few tens of kilometers". *(Figure 1) The red dotted line in Fig.1b is hard to see. Please make it more prominent. In Fig.1c there are two red dotted lines. I guess the northern one should be deleted. *(P5L17) Please insert "Hibler" *(P8L9, Eq.3) I think "A" is not needed. *(P17L4) Please replace "than" by "that". *(P19L10) I agree, but there are some discrepancies in the slope of the thickness pdf around 1 m. Is that a negligible problem? *(P19L18) In the equation, $h \cdot â\acute{L}\breve{G}u$ should be $hâ\acute{L}\breve{G} \cdot u$. *(P21L11-12) "prescribing a cut-off for biaxial compressive strength... appears unnecessary" I could not understand this. Can you add some additional explanation? *(P23L10) "Hibler" is missing.

That is all. Faithfully yours.

---

## Author Comment (AC1) · 15 May 2017

**Minor and Technical Comments**

**Abstract, first sentence. You need to tell the reader what "EB" is and what the model does before launching into a discussion of its implementation. Like this: "The Maxwell Elasto-Brittle (EB) model uses a sea-ice rheology that allows tensile stress... blah blah [one sentence about the model]. This paper presents a first implementation of the..."**

This sentence is now added to the abstract: " This continuum model, called Maxwell-Elasto Brittle (Maxwell-EB), is based on an Maxwell constitutive law, a progressive damage mechanism that is coupled to both the elastic modulus and apparent viscosity of the ice cover and to a Mohr-Coulomb damage criterion that allows for pure (uniaxial and biaxial) tensile strength."

**Abstract, lines 6-8. This sentence is a bit awkward. I suggest: "In agreement with observations, the model captures the propagation of damage..." etc. (and delete "are all represented" at the end of the sentence).**

Agreed: we made the suggested modifications.

**Abstract, last sentence. "weakening of the ice cover" and "shorter lifespan of ice bridges" – over what time period? You should add something like "in the 2000s relative to the 1980s and 1990s". And then "with implications in terms of increased ice export" – say that this would be expected because sea ice is expected to continue thinning in the future. In fact, maybe this is a positive feedback on the loss of ice: as it weakens, it drains from the Arctic faster, which weakens it further.**

We are not specific about a time period for these two processes in the abstract because they have not been deduced to occur at the exact same time: the weakening of the ice cover estimated by *Gimbert et. al. 2012b* occurred over the period 2002-2008 relative to the period 1979-2001 while the shorter lifespan of ice bridges reported by *Barber et al., 2001* occurred in the 1990s relative to the 1980s (see section 5.1.3). The formulation used in the abstract reflects the aim of the experiments presented here, where is to relate in a general sense a mechanical weakening of the ice cover to a shorter lifespan (or absence) of ice bridges and increased ice export, i.e., not to reproduce a trend in the lifespan of ice bridges observed in recent years. Moreover, we do not believe that speculating about a future ice thinning is relevant at this point in the paper: actually, the numerical experiments presented here aim to relate a shorter lifespan of ice bridges and increased ice export to a *genuine mechanical weakening* of the ice cover, i.e., a weakening that is *independent* of a decrease in ice thickness. Mentioning an expected thinning of the ice cover in future years in the abstract would turn the attention away from this important point.

**Page 1, line 15. "expenses" should be "expanses"**

**Page 1, line 21. "May 2005" should be July 2010.**

**Page 2, line 12. Change "allowed demonstrating" to "demonstrated"**

**Page 2, lines 18-20. "shorter lifespan of ice arches" and "increased ice export" – over what time period?**

Here we added the reference time period for the mechanical weakening reported by *Gimbert et. al. 2012b*. To clarify the point made in the response to your earlier comment, we modified this sentence as follow: "we also discuss how the mechanical weakening of the ice cover estimated over the period 2002-2008 relative to the period 1979-2001 (*Gimbert et al., 2012b*) can be linked to a shorter lifespan of ice arches and consequently, to an increased ice export through Nares Strait".

**Figure 1. The dates on the images are in the format DAY-MONTH-YEAR. Is that standard for The Cryosphere? It might confuse U.S. readers, who often use MONTH-DAY-YEAR.**

Yes, DAY-MONTH-YEAR is a standard for the *European* Geosciences Union.

**Figure 1, caption. Put "(MODIS)" before the word "reflectance", not after.**

**Page 4, line 10. 0.32 degrees is not 4-10 km.**

There was a word missing there: 0.32 degrees *of longitude* is indeed equivalent to 4 km in the Lincoln Sea and 10 km in Baffin Bay (taken from *Rasmussen et al., 2010,* page 163: "The model horizontal grid has a longitudinal grid size of 0.32° resulting in a grid size ranging from 4 km in the Lincoln Sea to 10 km in the Baffin Bay"). We now remove the mention to degrees and give the resolution in kilometres only to avoid any confusion.

**Page 4, lines 21-24. For the statement about the sea-ice thickness PDF you can cite: Lindsay, R. W. 2013. Unified Sea Ice Thickness Cli- mate Data Record, 1975-2012. Boulder, Colorado USA: National Snow and Ice Data Center. http://dx.doi.org/10.7265/N5D50JXV. Web site: http://nsidc.org/data/docs/noaa/g10006-unified-sea-ice/**

Thank you for this reference.

**Page 5, line 3. Change "pioneer" to "pioneering"**

**Page 5, line 9. "dubiously" – Why it is dubious to base the modelling framework on energy conservation?**

Basing the estimation of the strength of the ice cover, i.e., equating deformational work to the sinks of energy that are taken into account in this framework is dubious in the sense that all of these sinks are very hard to estimate, and do not account for other processes such as ice crushing, buckling, frictional contacts between ice blocks in the rubble pile, etc. However, we agree that the use of the adjective "dubious" is unnecessary and nonobjective. We rephrase this passage in an clearer and more objective manner as follow:

"The relation between the redistribution process and the strength of the ice (often characterized by a pressure, *P*) in this modelling framework is based on energy conservation principles : the deformational work is equated to the work done in building ridges, which is partitioned between potential energy changes (*Thorndike et al., 1975*), the frictional dissipation in ridging (*Rothrock, 1975*) and dissipation in shearing deformation (*Pritchard, 1981*), all of which are very hard to estimate. This theory does not take into account other mechanisms such as crushing, buckling, flexural breakage, inelastic contacts and frictional sliding contacts between rubble ice blocks (*Hopkins, 1998*)."

**Page 5, line 18. There is an error in the citation of Hibler's 1980 paper. William D. Hibler III should be "Hibler", not "III". Also on page 23, line 10, the author is Hibler, not III.**

Yes, thank you for catching this.

**Page 5, line 19. "sensible" should be "sensitive"**

**Page 6, line 5. Instead of "recall", better to write "review"**

**Page 6, line 7. Instead of "passage", I think "connection" is the intended meaning**

You are right, "passage" might not be the right word: we changed it for transition.

**Page 6, line 9. After "elastic modulus" write "(E)" because it is used three lines later in the definition of lambda.**

**Page 6, equations (1) and (2) – you need to say that "mu" is the internal friction coefficient.**

Yes, thank you.

**Page 6, lines 27-28. About healing: healing allows the level of damage to DECREASE – more healing, less damage. But the damage parameter d INCREASES as healing increases (d=1 is undamaged). So you have to be careful when describing healing. Verbally, healing means a SMALLER AMOUNT of damage. But numerically, healing means a LARGER VALUE of d.**

The formulation in lines 27-28 is correct: the *level of damage*, *d*, has been introduced in lines 15 and 16. It is not employed the same as "damage". Here it is specified that the *level of damage* re-increases at most to the (numerical) value of 1.

Here I add "the level of damage *variable*" to make the distinction clearer.

**Page 7, Figure 2, caption. Several things: – It is awkward to have complicated equations in the caption. Furthermore, the components sigma11, sigma12, and sigma22 are not used anywhere else in the paper, so I think it would be OK to delete the equations entirely. Otherwise, they should probably go in the main text. That's just a suggestion, there is nothing wrong with the current presentation. – Second sentence: "The thin solid lines represent the damage criterion in the case of C=0." To make this clearer, consider: "The thin solid lines radiating from the origin represent the damage criterion in the case of no cohesion (C=0)." – What is the shaded region in the figure?**

Agreed: the equations are now removed. We also included your comment about the cohesion. The shaded region is also removed: it has actually no meaning relative to the simulation results presented here.

**Page 8, line 7. Change "assimilated to" to "simulated as"**

OK

**Page 8, line 27. I think max[0,(1-A)] should be max[0,A-1]. This is supposed to describe "the excess concentration" when A > 1. If A > 1 then max[0,1-A] = 0, which doesn't make sense. The excess concentration should be max[0,A-1].**

Yes, thank you very much for catching this: this is a typing mistake and the right form is indeed given in Eq. 6.

**Page 9, equation 7. This equation doesn't make sense either. When A > 1, h+ = 0. I think this equation should be h+ = max[0,A-1]h.**

Yes, the same mistake was repeated there. We however insist on the fact that this term is implemented in the correct form (h+ = max[0,A-1]h) in the code (see response to reviewer's 2 comment).

**Page 9, line 18. Change "of being" to "to be". Change "sturdy" to "steady" Page 9, line 19. Capitalize "Lincoln Sea"**

**Page 10, line 27. Acceleration and advection terms are neglected in equation 3. Does this mean the solution is steady state, with no time dependence?**

No: in the originally submitted version of the paper, the acceleration and advection term were neglected only in the momentum equation. The time derivatives and advection of all other variables ($\sigma, h, A, C, d$) were accounted for. Hence the solution was therefore not steady state. However, as mentioned in our response to reviewer 2, and as expected from a dimensional analysis of the momentum equation (see *Dansereau et al., 2016*, section 4.1.4), including the advection and acceleration terms in the momentum equation does not affect the results and conclusions of this paper. In the corrected version of the paper, both terms are included in all simulations (which needed to be run again due to an error in the thickness redistribution scheme, see our response to reviewer 2). This does not impact the results.

**Page 11, Figure 3 caption. Delete the first "(a)" because the caption later refers to (a) and (b).**

**Page 11, line 11. "quenched disorder" – this needs a reference.**

You are right, and more precision was needed on this point. This paragraph is now modified and this sentence is added after line 15 : "In all simulations, the disorder introduced in the field of cohesion is quenched (*Hermann and Roux, 1990*): it is set once, at the beginning of each simulation, and is passively advected with the ice flow''.

**Page 11, line 16. Beaufort SEA. Also, Weiss et al (2007) is cited in connection with Figure 8(a), but the figure caption cites Weiss and Schulson (2009).**

*Weiss et al., 2007* are cited in connection with the in-situ stress measurements in the Beaufort Sea, but figure 8(a) does show figure 13(a) from *Weiss and Schulson, 2009* (which presents the same measurements as in *Weiss et al., 2007*, figure 2A, but with the convention that compressive stresses are positive, as in the present paper). The reference to *Weiss and Schulson, 2009*, is now added in line 16 to make the connection to figure 8(a).

**Page 12, line 10. Delete "the upstream part of", so the sentence reads, "...prescribed on GAMMAin and GAMMAout..."**

**Page 12, lines 14-15. "The equations of motion... Galerkin methods are used to handle advective processes" – but the advection terms in equation 3 are neglected. So I don't understand this. Does it refer to advection in equations 4 and 5?**

The advection of ice momentum only is neglected here. All other variables are advected with the ice flow.

**Page 14, line 2. "deformation rates" – I think this should be "high deformation rates"**

Yes, thank you.

**Page 14, line 3. "downstream of the channel" should be "downstream of the constriction"**

No: here we precise "in the interior" of the channel, hence downstream of the constriction point, and "downstream of the channel", i.e., of the opening point.

**Page 14, Figure 4. This figure should be bigger, so that it fills the full width of the page. It's a bit difficult to see the peaks of wind stress in panel (a) – bigger would be better.**

Agreed: this figure is now resized.

**Page 15, line 23. I think "Fig 4b and 4c" should be "Fig 5b and 5c" Page 15, line 27. I think "Fig 4b" should be "Fig 5b"**

Thank you for catching this.

**Page 16, line 24. The text says "Weiss et al (2007)" but the Figure 8 caption says "Weiss and Schulson (2009)"**

We changed the reference there to *Weiss and Schulson, 2009*.

**Page 18, line 2. "ice landfast" should be "landfast ice"**

**Page 18, line 8-9. How does the ice become weaker without thinning? Is it because the ice strength is temperature-dependent, so increasing the temperature would make it weaker? Is it because the composition of the ice could change, such as increased salinity?**

Here, the mechanical weakening refers to a more fragmented/fractured ice cover (*Gimbert et al., 2012b*), and so (as mentioned by reviewer 3) an evolution towards smaller and different shape (more circular) ice floes. This point is now made clearer in section 5.1.2 (page 17, line 12, see response to reviewer 3). We neglect temperature and salinity effects.

**Page 23, line 10. The first author is Hibler, W.D. III Page 25, line 27. Weiss and Dansereau 2016 – is the DOI correct?**

**Page 26, Figure 5 caption. At the end of the last sentence, "red arrows on (a)" should be "red arrows on 4(a)"**

**Page 27, Figure 6 caption. Delete the first "(a)" because the caption later refers to (a) and (b).**

**Page 29, Figure 8 caption. Delete the first "(a)". Capitalize "Beaufort Sea" C5**

**Page 31, Figure 10 caption: – line 2: "is of 10" should be "is 10" – lines 5 and 6: "insert" should be "inset"**

**I recommend deleting these two sentences: "The tail of the distribution..." and "The temporal evolution..." because the last sentence of the caption explains the same thing as these sentences.**

Agreed, these sentences are removed.

**Page 31, Figure 11 caption:**

- **line 2: "is of 10" should be "is 10"**

- **line 5: "insert" should be "inset"**

Agreed, and the repetitions mentioned in your previous comment were also removed from this caption.

**Please note: I have not flagged all the minor grammatical corrections that should be made by an editor, for example:**

**Page 9, line 10. "is of 100%" – delete "of" Page 9, line 26. such AN ice bridge Page 10, line 8. "in Cartesian" should be "to Cartesian" Page 10, line 9. Delete "is" Page 11, line 8. employed IN the simulations Page 11, line 12. "by randomly by" – delete second "by" ...and so on...**

---

## Author Comment (AC2) · 15 May 2017

**RESPONSE TO REVIEWER 3**

**General comments**

1) **It would be helpful to explain more about the advantage of the Maxwell-EB rheology compared with the traditional elliptic curve rheology. The authors pointed out the capability to represent the extreme localization of damage and deformation (P4L13). However, there is a possibility that it comes just from the horizontal resolution of the grid cell in the model. I mean that the traditional plastic rheology might be able to reproduce the phenomena if finer grid cells are used. Thus I want to know why they consider the continuum elasto-brittle rheology should be more appropriate than the continuum viscous plastic rheology for this phenomena, and whether this rheology can be applicable to the general sea ice conditions.**

The difference between the VP and Maxwell-EB rheology is two-folds: it lies in the rheology itself (i.e., the viscous-elastic-brittle versus the viscous-plastic constitutive relationship) and in the prescribed damage (or yield) criterion. Of course, both aspects impact the simulated mechanical behaviour.

First, the goal in developing the Maxwell-EB framework was to suggest an alternative to the traditional Viscous-Plastic (VP) rheology that is more physically sound, as in recent years, the viscous hypothesis and other underlying physical assumptions of the VP model have been revisited and found to be inconsistent with the observed mechanical behaviour of sea ice in many aspects, e.g., with respect to the order of magnitude of the observed strain rates (*Weiss et al., 2007*; *Rampal et al., 2008*), the anisotropic distribution of ridges and leads and associated discontinuities in the velocity field on scales both small and large ( > 100 km) (*Hibler, 2001; Schulson, 2004; Coon et al., 2007*), the relation between stresses and strain-rates (*Weiss et al., 2007*), the strength of pack ice in tension (*Weiss et al., 2007; Coon et al., 2007*) and the normal flow rule (*Weiss et al., 2007*) . The aim in building this new continuum model was to represent accurately the *deformation and drift* of sea ice. In particular, we wished to developing a modelling framework that allows representing both the small deformations associated with brittle failure and the large deformations occurring within a fractured ice cover. In the paper, these points are introduced in the last paragraph of the introduction and in the first paragraph of section 3 (which presents the model). As they are discussed in the first paper that presents the motivations for and the details of the Maxwell-EB rheology (*Dansereau et al., 2016*) and references to this paper are included both in the introduction and in section 3, we do not think repetition is needed in the present paper, which we wish to be relatively short and to focus on the implementation of the model on geophysical scales.

One particularity of the Maxwell-EB model is that the localization of deformation indeed *does not* depend on the spatial resolution, *in the sense that the tendency to localize damage and deformation at the smallest available scale, i.e., the scale of the model grid cell, is intrinsic to the rheological framework.* In other words, no matter the spatial resolution, the Maxwell-EB model reproduces a localized deformation. This point is also discussed in more details in the paper that first presents this new rheology (see *Dansereau et al., 2016*, section 6.1) hence, an in-depth discussion was not included here. The fact that the representation of ice bridges and leads does not depend on the choice of spatial resolution (over the range of spatial resolutions that allow resolving the flow of ice through the channel) is mentioned in the description of the simulation setup (p. 12, line 18 and page 13, lines 1 and 2) and discussed in terms of the representation of the thickness redistribution (figure 10b).

We also discussed the issue of spatial resolution in our response to reviewer 2. As mentioned in section 4 and discussed in section 5.2, we analyzed lower resolution simulations. These simulations show that the model reproduces a stable ice bridge, a clearly defined ice front, arch-like and linear leads upstream of the ice bridge, and a distribution of ice thickness with a tail that follows an exponential function as in the higher resolution cases. Since we did not perform VP model simulations, in the paper we did not speculate on the fact that VP models at very high resolution can or cannot reproduce ice bridges in narrow passages. However, we believe that if a model can reproduce ice bridges and other important processes *only* at high to very high resolution it is not good news, as the physics represented by a model should not be resolution-dependant.

Second, the main advantage of using a Mohr-Coulomb (MC) failure criterion instead of the elliptical yield curve for the damage criterion in the Maxwell-EB model is that the MC criterion appears in agreement with in-situ stress measurements (see figure 8 a). Also in agreement with observations, the current damage criterion allows accounting directly for some resistance of the ice in pure ( $\sigma_1 < 0$ and $\sigma_2 < 0$ ) tension. As demonstrated in the present as well as in previous papers (ex., *Dumont et al., 2009*), resistance of the ice in tension is especially important for simulating stable ice bridges. This point is discussed in section 2.1. Another advantage of the MC damage criterion is that the cohesive strength of the ice (i.e., $\sigma_t$ and $\sigma_c$ ) can be set and adjusted directly by varying the cohesion parameter, *C*, rather than indirectly, by changing the ratio of the ellipse.

To answer your last comment, the simulations performed here are indeed on a regional scale, and concern very specific flow conditions. However, the results gives us no reasons to think that the model could not be applied to more general conditions. On the contrary, the model proves to behave well in this "extreme" case, i.e., a case chosen especially to test the ability of the model to represent (1) the complex mechanical behaviour associated with the formation of an ice bridge and (2) the

discontinuities in ice velocity, concentration, thickness, etc., associated with the presence of this bridge.

2) **Intuitively my feeling is that the floe size distribution of sea ice should also play an important role in the brittle ice rheology. Therefore, in the question at P17L7-8 it would be natural that the change in floe size distribution may also contribute to the phenomena. What do you think?**

This is a good point. The paper discussed in this section (*Gimbert et al., 2012b*) identified a mechanical weakening of the ice cover that is independent of an ice thinning and suggested that this weakening is related to the degree of fragmentation of the ice cover. A more fragmented ice pack is indeed in agreement with an evolution towards smaller ice floes. We now add a mention to this effect in section 5.1.3, paragraph 2. It is also consistent with a change in the shape (circularity) of the floes, a less cohesive state ice cover, an enhanced deformation and an increased ice drift (*Rampal et al., 2009*).

Of course, continuum models by definition, whether using a VP, EVP, EB or Maxwell-EB rheology, do not resolve ice floes per se nor the mechanical interactions between individual floes. Hence it would be interesting to explore this question further using a discrete element model (ex., *Rabatel et al., 2015*; *Hopkins, 2004*; *Herman, 2011*; *Wilchinsky et al., 2011*), that is, to try relating the floe size distribution to the cohesive strength of the ice cove in a quantitative manner.

2) **On the whole, I am somewhat concerned about why the authors did not pay so much attention to the horizontal scale. For example, the scales of ice bridges seem to be different depending on the straits. Accordingly the mechanism might be different depending on the regions. Could you explain how the Maxwell-EB rheology influence the results depending on the scales.**

We are sorry we might not understand this comment fully.

The horizontal scale of ice bridges is the width of the constriction point across which it forms. In general, the limiting span that can support a stable arch between vertical walls or in a vertical tube depends on several properties of the material (its density, cohesion, internal friction) and the friction between the material and the walls (*Richmond and Gardner, 1962*). In the case of sea ice, the presence of a stable ice bridge should depend on the cohesion of the ice cover, its thickness, concentration, etc., the friction between the ice and the coast (here we prescribe a no-slip boundary condition), but also on the wind and ocean forcings. *Rallabandi et al., 2017* for instance developed a one-dimensional theory for the wind-driven formation of ice bridges in narrow straits in a VP model and investigated the formation of a stable ice bridge at a given wind stress, maximum and minimum channel width, ice thickness and compactness in this model. A study of the limiting span of ice bridges observed to form in the Arctic with a comparison to Maxwell-EB model simulations would indeed be interesting but is beyond the scope of the present paper.

However, as mentioned on page 10, lines 6 to 8, simulations with different idealized domains (narrower, longer channels, smaller basins) were performed to verify that the dynamics described in the paper is not specific to the shape and dimension of the idealized channel. Moreover, the use of a realistic domain allows investigating the formation of ice arches at different locations, hence with different spans, in the Maxwell-EB model.

**Although the description on the scale dependence (P19L17-25) is interesting, in general it seems that the localization of deformation depends on the grid cell size. Could you explain why this property is independent of resolution?**

As mentioned in our response to your major comment (above), the tendency to localize the damage and deformation at the smallest available scale is intrinsic to the Maxwell-EB rheology. Hence there is no characteristic scale for the localization of damage and deformation in the model beyond the scale of the model element (see *Dansereau et al., 2016*, sections 6.1 and 6.2). Therefore, at all spatial resolutions, the simulated deformation is highly localized. In the present simulations, this translates into a localization of the mechanically redistributed, i.e., the "ridged" ice and an exponential tail of the ice thickness PDF at the spatial resolutions explored (2 km, 4 km and 8 km in the idealized channel case). This point is now made clearer in section 5.2.

**Specific points:**

**\*(P1L2)"on geophysical scales" I wonder if we can assume ice bridges and ridges to be on a geophysical scale. It would be preferable to describe the specific phenomena like "ice bridges on a few tens of kilometers".**

The model is used here to simulate the drift of sea ice through a channel that is 500 kilometres long and a few tens to hundreds of kilometres wide. Ice bridges and ridges are smaller-scale features resulting from the associated deformation of the ice cover. We believe it would have indeed been wrong to claim that the model was used on *global* scales, but the setup used here does qualifies this application as to apply on "geophysical" scales.

**\*(Figure 1) The red dotted line in Fig.1b is hard to see. Please make it more prominent. In Fig.1c there are two red dotted lines. I guess the northern one should be deleted.**

Yes, thank you for catching this.

**\*(P5L17) Please insert "Hibler"**

**\*(P8L9, Eq.3) I think "A" is not needed.**

The air and water drag terms in the momentum equation are indeed both multiplied by the ice concentration. This approach was suggested by *Gray and Morland, 1994* and *Connolley et al., 2004*, to account for the contribution of the ice-free and ice-covered fraction of a grid cell to the wind and water stress. *Connolley et al., 2004*, explains the necessity of introducing this weighting to maintain physical consistency in the free-drift limit. Without it, the free-drift solution of the momentum equation (when including the Coriolis term)

depends on ice concentration, i.e., ice floes with the same thickness would not be drifting at the same velocity based on their concentration, even in the limit of negligible mechanical interactions. Here, this "correction" is included for the sake of physical consistency, even if not strictly necessary since the Coriolis term is neglected in the present implementation of the model. We now add a reference to the work of Connolley et al., 2004 when introducing the form of the momentum equation solved here (Eq. A1).

This weighting approach is quite standard and was used for instance in the sea ice models of *Tremblay and Mysak, 1997, Lieataer et al., 2009, Danilov et al., 2015* (FESIM), and others. Interestingly, in the present model, it has effectively little effect on the simulation results, a point also noted by *Connolley el al., 2004* and *Tremblay and Mysak, 1997*.

**\*(P17L4) Please replace "than" by "that".**
Yes, thank you.

**\*(P19L10) I agree, but there are some discrepancies in the slope of the thickness pdf around 1 m. Is that a negligible problem?**
This discrepancy is explained by the fact that a uniform thickness of $h = 1.0$ m is prescribed as the initial condition in all simulations presented here. Hence we naturally expect a mode to stand out at $h = 1.0$ m. The tail of the PDF, which represents the ridged ice, is therefore the part of the distribution with $h > 1.0$ m.  Here, the PDF was effectively fitted with an exponential function for all values of $h > 1.0$ m. The presence of the mode indeed results in a systematic misfit near $h = 1.0$ and fitting the distribution for larger values of $h$ only gives a somewhat better fit. Nevertheless, the values of the coefficient for the goodness of the fit obtained here vary between 90% and 98% in the idealized and are > 95% in the realistic case.

**\*(P19L18) In the equation, h cdot nabla u should be h nabla cdot u.**
Yes, thank you for catching this.

**\*(P21L11-12) "prescribing a cut-off for biaxial compressive strength. . . appears unnecessary" I could not understand this. Can you add some additional explanation?**
As suggested both by in-situ stress measurements (see figure 8a) and the realistic numerical simulations performed here (see figure 8b), large biaxial compressive stresses seldom occur in the sea ice cover. This is an interesting result, since the flow conditions here are convergent over a large part of the domain. The stress states measured and reproduced by the model indicates that the ice fails frequently under pure tensile and biaxial tensile-compressive (i.e., shear) stresses (which is also illustrated in figure 5c). This point is further discussed in the response to reviewer 2.
Because large biaxial compressive stresses and pure biaxial compressive stresses, i.e., compressive states of stress involving little shear ( $\sigma_1 \sim \sigma_2$ ), are marginal, imposing a biaxial compression damage criterion, would not significantly affect the number of damage events and propagation of damage in the Maxwell-EB model. The addition of such a cutoff is not supported (and not well constrained) by the observations. Instead, in-situ stress measurements suggest that the uniaxial (unconfined) compressive strength, $\sigma_c$ and maximum tensile strength (or $\sigma_t$) are more relevant parameters to describe the failure strength of the ice cover.

To make this point clearer, we modify this paragraph as follow:
"Besides numerical efficiency, other advantages of using a simple redistribution scheme such as the one employed here is that no thickness redistribution function needs to be assumed and the redistribution is not directly tied to the prescribed failure strength of the ice. In the Maxwell-EB model, the prescribed strength is instead based on in-situ stress measurements, which point to a Mohr-Coulomb failure criterion and directly provide information on the order of magnitude of the shear strength and tensile strength. In particular, both the observations and numerical simulations here suggest that prescribing a cut-off for biaxial compressive strength (equivalent to the pressure, *P*, in VP models) is unnecessary. Instead, the uniaxial (unconfined) compressive strength, or sigma_c and maximum tensile strength, sigma_t appear to be more relevant to represent adequately the strength of the ice cover. The Maxwell-EB model presents the advantage that both these quantities are set through a single parameter, the cohesion *C*."

**\*(P23L10) "Hibler" is missing.**

---

## Author Comment (AC3) · 21 May 2017

**RESPONSE TO REVIEWER 2**

**Major comments:**

These major comments are also addressed through the response to your other minor comments, below.

**There are many statements that require more discussion and clarification. In some places strong statements are made that not supported by the results of the model simulations. These statements need either be toned down, or sufficient evidence should be provided (which is probably not possible without additional experiments, especially when reference to VP models is made.)**

We stress the point that no direct comparison was made between the present Maxwell-EB and VP/EVP model simulations. This is further discussed in the responses to minor comments.

**For instance, the impact of the simplified model equations, geometry, wind forcing and no thermodynamics is discussed too little, especially when comparing to observations made outside of a channel. The model set-up for the analysis of PDFs of ice thickness facilitates ridging with a high amount of coastal boundaries, low cohesion parameter and strong wind forcing (and a presumably wrong source term in Equation (7)), which is not further discussed.**

We have addressed issues with the thickness distribution in the revised version of the paper. These issues are discussed in the responses below. The PDFs of ice thickness produced are not the result of a "low cohesion" (see our response to your comments about cohesion below : there is no physical or observational basis to claim that the value of cohesion used for these particular simulation are "low), nor of the amount of coastal boundaries (as idealized simulations have also been used). The strong wind forcing used here has the effect of redistributing the ice thickness starting from a uniform ice cover in a shorter amount of time. A lower wind forcing gives similar results over a longer time.

**In the experiments with reduced mechanical strength a comparison to observations would be even possible: Are a collapse of ice bridges and no flow storage observed in the recent years?**

As mentioned in the paper, the absence of stable ice bridges in Nares Strait was indeed observed in the year (September to August) 2006/2007. For that year, *Kwok et al., 2010* estimated an annual ice areal and volume flux equivalent to twice their average value over the 1997 to 2009 period. For the same year, *Ryan et al., 2017* observed the maximum of the median ice draft (measured across Kennedy channel, upstream Kane Basin) over the 2003 to 2012 period. *Munchow et al., 2016* reported that the ice arch between Kane Basin and Smith Sound failed to form in the winters 2006/07, 2007/08, and 2009/10, and collapsed after less than 2 months in 2008/09. They estimated an increase of 45% of the volume flux, of 69% of the ocean freshwater flux and of 46% of the freshwater flux through Nares Strait over the 2007-2009 period relative to the 2003-2006 period, during which a stable arch did form at that location. We now add references to the recent studies of *Ryan et al., 2017* and *Munchow et al., 2016* in the text (section 5.1.3).

A direct comparison of the model simulations to the ice area or volume flux estimated for instance by *Kwok et al., 2010* is not trivial: it would require at least the knowledge of the temporal and spatial evolution of the wind forcing (and perhaps of ocean currents) as well as of the coverage and thickness of ice over Nares Strait over that time period. In the absence of these informations, we compared time series of the meridional component of the simulated ice drift velocity averaged across the constriction point between Kane Basin and Smith Sound for different ice cohesion scenarios to illustrate the impact of ice strength on the simulated outflow.

**Especially, regarding the comparison to VP-models more caution is required. If no reference VP simulation is performed, the differences of the referred studies need to be considered, as most of the studies do not agree in resolution or region used in the experiments presented here. For example, it is not clear, if VP models at the same resolution would not lead to the same type of "ridging" behavior; as long as it is not clear, it shouldn't be claimed. The detailed overview which points need further discussion are listed below in the minor comments.**

2 EVP studies were mentioned (*Dumont and Gratton, 2009* and *Rasmussen et al., 2010*), in the context of simulating ice bridges, not ridges, and the resolutions of their simulations are indeed comparable to the one used here, at least in the case of *Dumont et al., 2009*. As mentioned in the response to a later comment, the Maxwell-EB model produce similar results at lower resolutions. Moreover, as discussed below, we insist on the fact that no direct comparison was made here between the Maxwell-EB and VP/EVP model simulations.

Again, we insist on the fact that it was not claimed in the original version of the paper that the VP model couldn't reproduce the same type of "ridging". What was stated is that the Maxwell-EB model can reproduce characteristics of the observed ice thickness distribution with a very simple redistribution scheme. As mentioned in the response to your later comment, *rheologies* and *thickness redistribution schemes,* are *distinct* components of a sea ice model.

**In Section 3, a sea-ice model using the Maxwell-EB rheology is outlined. However, due to the reference to individual equations in Dansereau, et al. (2015) the readability is reduced. A summary of the model equations cited from Dansereau et al. (2015) in an appendix would assist the reader. In addition, if you would include the simplifications that are made in your experiments into the model description, it is directly clear to the reader what equations are solved in your model. For example, the full momentum equations are given**

**in Equation (3); one can easily miss the fact that many terms are not solved for (p10, l.26-27). Here the reduced equations could replace Equation (3), since this is all that is used in the manuscript.**

We completely agree on the need for more information about the equations and numerical scheme. We now include a detailed description of both in an Appendix. We further comment on the simplifications to the equations which are listed in the paper (p. 10, lines 26 to 29) here:

- *No thermodynamics coupling.* The present implementation of the Maxwell-EB rheology is not coupled to a thermodynamics model. The goal of these numerical experiments is to investigate its dynamical behaviour. Including thermodynamics processes would complicate this investigation. Simulations are analyzed over a short time-period (3 days maximum). This point is now made clearer in section 4. In terms of referring to previous VP (EVP) simulations at comparable resolution, this simplification is justified as thermodynamic processes were also neglected in the study of *Dumont et al., 2009*.

- *No Coriolis acceleration and ocean tilt terms.* As mentioned in section 4, second and third paragraphs, forcing conditions are made as simple as possible to facilitate the analysis of the dynamical behaviour of the model. Hence the ocean is at rest and the wind forcing is uniform over the channel in both the idealized and realistic cases. The Coriolis term is neglected with the intention to retain symmetry in the forcing conditions. Because the Coriolis acceleration plus sea surface tilt term is smaller in magnitude than the air and water drag and rheology terms (*Steele et al., 1997*) this is not expected to have a significant impact on the results. Note that *Dumont et al., 2009* also neglected the Coriolis acceleration in the idealized case. Also, in the realistic case, the ocean in *Dumont et al., 2009* is initially static.

- *No acceleration and advection term in the momentum equation.* Scaling analysis show that these terms are small (see *Dansereau et al., 2016*, section 4.1.4) Most sea ice models now include the $\partial \upsilon / \partial \tau$ term, however the advection term is still neglected in a number of, if not most, sea ice and ice-ocean coupled models (e.g., LIM3, FESIM/FESOM, ...). Additional simulations have shown that including both terms in the momentum equation does not affect the simulation results reported here. To correct the error made in the thickness redistribution scheme (see your later comment), all simulations needed to be run again for the resubmission of the paper. Both the acceleration and the advection terms are included in the corrected version of the model used to perform these simulations. The results are in all aspects very similar to the previously reported results such that all the conclusions of the paper remain unchanged.

We corrected the reported spelling and structure mistakes and respond to your minor comments below.

**Minor comments:**

**page 1,l.7: "in" -> into page 1,l.8: "Strait" -> strait page 1,l.9: "different dynamical behaviours" -> various dynamical behaviour**

**page 2, l.26: "to" -> and?**

**page 2, l.32: "This rheological framework typically does not account for (uniaxial or biaxial) tensile strength." Not quite true, there is uniaxial tensile strength in Hibler elliptical yield curve (see your figure 2), but there's usually no isotropic/biaxial tensile strength; adding tensile strength is another option for the VP model and has been used to improve simulations of land fast ice (Lemieux et al 2016, Olason, 2016). Could be mentioned in this context, too.**

As shown on figure 2, there is no uniaxial tensile strength in Hibler's elliptical yield curve. Instead, there is only *biaxial tensile-compressive strength* (accounted by the portion of the curve in the second and fourth quadrants, see stress state 2, Fig. 2). Uniaxial tensile strength is represented on Fig. 2 by stress state number 1 : $\sigma_1 < 0$ and $\sigma_2 = 0$ (or the inverse, by symmetry with the $\sigma_1 = \sigma_2$ axis). Biaxial tensile strength implies resistance of the material for $\sigma_1 < 0$ *and* $\sigma_2 < 0$. This state of stress is now represented schematically as stress state number 0 on figure 2.

Thank you for these references. We are aware that the elliptical yield curve has been modified especially in the context of modelling landfast ice, which has been related to the phenomenon of arching between islands. However, as both these papers are really concerned with the phenomenon of landfast ice, and landfast ice is also influenced by other phenomenon such as the grounding of keels, we beleive that including these reference at this point in the paper (in the discussion of the phenomenon of ice bridges, in particular in Nares Strait) would take the reader away the main point, which is that we are testing a new rheology on the basis of its capacity to represent the phenomenon of arching.

**page 2, l. 30-32: Dumont also used EVP**

Thanks for catching this. This is now corrected.

**page 4, l. 13-14: That's a speculation. It would be nicer to actually show that this can work or not work with**

**VP models at high resolution (better than 4km grid spacing).**

"It is not clear" is not speculation. We however remove the work "better" in this sentence, which indeed implies a comparison to VP models.

**page 4, l.13: ". . .(e.g., see Fig. 1b and Sodhi,1977)" wrong citation**

**page 4, l.20: Section 2.2 "Ice ridges". First paragraph is relevant for the following analysis of the ice thickness distribution. The last three paragraphs can be shortened or discarded, as neither a VP-model nor an ITD is used later on.**

We do not agree with this comment: this discussion of ice thickness redistribution schemes is relevant and necessary to interpret the results discussed in this paper. We also stress the fact the thickness distribution scheme is a component *independent* of the rheological framework implemented in a sea ice model. VP rheology models actually use both the schemes described here. The only sentence that discusses the VP model in these paragraphs is the following:

"Nevertheless, it is still unclear to this day if, when incorporated in viscous-plastic type models, either of the two-level scheme or the multi-categories scheme, even when tuned, is able to reproduce the form of tail of the PDFs calculated from Arctic sea ice thickness measurements (e.g., *Flato et al., 1995*)."

This statement refers to a 22 years-old paper. Other more recent VP or EVP model studies in which an ice *thickness distribution* was represented (otherwise only thickness fields are discussed) were not found, which adds to the fact that the point made in this sentence "is still unclear". We however agree that the mention of VP models here is unnecessary since *the goal of the paper is to demonstrate that the Maxwell-EB model, with a very simple redistribution scheme, is capable of reproducing the exponential tail of the ice thickness distribution*. We therefore remove this sentence and add another one at the end of this section to stress this later point.

**page 5, l.17: wrong citation, I guess Hibler, 1980 is meant? page 5, l.19: "badly" -> poorly**

**page 6, l. 5: "details" -> detail**

**page 6, l. 5: "recall" -> repeat?**

We changed it for "review", as suggested by reviewer 1.

**page 6, l.15: This definition is unfortunate and unintuitive. 0 should be "undamaged" (zero damage) and 1 should "completely damaged" if d is called "damage". d feels more like "integrity" for the material, but it's just terminology . . .**

Indeed, in solid mechanics conventions *d* in this case would have the meaning of "continuity". However, this just terminology that helped us simplify the writing of some equations while developing the model. As this is the definition included in the paper describing the Maxwell-EB rheology (*Dansereau et al., 2016*) we do not modify it in the present paper.

**page 6, l. 22: Isn't Equation (2) t = 2C[(μ2 + 1)1/2 + μ] 1 ? At least in Danserau et al. (2016) Equation (7),(8) and (10) are not consistent and I guess the exponent -1 is missing in Equation (10).**

You are right: thank you for catching this. This is also a mistake in *Dansereau et al., 2016*. Here we correct this mistake and write $\sigma_t$ as it is implemented in the code, that is, $\sigma_t = -\sigma_\chi/q$, and add the definition of $q$, the slope of the damage criterion. We also add a footnote to report the mistake made in *Dansereau et al., 2016,* Eq. 10.

**page 6, l. 27-28: Unclear, if it represents refreezing of leads, how can it be independent of pure thermodynamics, please explain/rewrite/elaborate**

You are right, this formulation is confusing. Healing is "distinct" but not "independent" from thermodynamics processes, as obviously, in a coupled dynamic-thermodynamic model, the rate of healing should depend on the air and ocean temperature. In the present, uncoupled, implementation of the model, it is constant. What was meant is that, on a modelling point of view, healing is not the same process as thermodynamic growth because it allows the level of damage variable to increase *at most* to its undamaged value (*d* = 1) and does *not* allow the mean ice thickness nor the ice concentration to increase. The sentence is rephrased as "This mechanism is distinct from pure thermodynamic growth (...)" and the reader is referred to *Dansereau et al., 2016* for more precision about healing.

**page 8, l.9: Equation (3); Please also state the simplified version of the equation that is actually used by the model, to prevent misunderstandings.**

This is now incorporated in the Appendix, together with the description of the numerical scheme.

**page 8, l. 21-22: Mechanical redistribution in your formulations is represented by the divergence term, see next comment about Equation (7).**

We address this point in the response to your next comment, below.

**page 9, l. 1, Equation (7) That is wrong: h is defined as the mean thickness (per grid cell area), so something like a volume of ice in the cell. The ice volume does not change if A > 1 is reset to A = 1, but the (mean) thickness of the thick ice hthick = h/A is increased, and there is no extra contribution to Sh. (See also Schulkes (1995), JGR.) This should be corrected in the text and also in the model, if the model actually implements this extra (spurious) tendency in Equation (4).**

There was indeed an error made in the redistribution scheme, which lied in the fact that for $A > 1$, $h$ could increase both through convergence and through the prescribed redistribution (equation 7). To correct this mistake, we have modified the parameterization as follow: the thickness of thick ice, $h_{thick} = h/A$, is advected passively with the flow for $A < 1$. This means that under convergent motion, there is no ridging if $A < 1$, but the ice volume ($h$) can effectively increase if the ice concentration over a grid cell increases. If $A > 1$ over a given grid cell, it is reset to $A = 1$, and the mean ice thickness, $h$, (equal to the thickness of thick ice when $A = 1$) is increased (equation 7): hence the ice volume in that grid cell increases.

As mentioned in this section, in the present implementation of the model we seek to account for mechanical redistribution of the ice thickness in the simplest possible manner, so that to test the input of the rheological framework, i.e., its representation of ice deformation, on the thickness distribution. *In Schulkes et al., 1995* the divergence term is weighted as a function of $A$. In the ice concentration equation, this term is penalized as $A$ increases from 0 to 1 and is zero for $A = 1$. Conversely, in the thickness equation, it is penalized as $A$ decreases from 1 to 0 and is zero for $A = 0$. As opposed to the scheme presented in *Schulkes et al.,1995,* we do not suppose ice ridging occur for $A < 1$ and assume ice riding occurs only for $A \geq 1$. This avoids using any weighting/penalty function based on sea ice concentration, which would imply introducing additional parameters and which is not well constrained by observations.

Our approach is therefore simpler and more similar to that of *Hibler 1979*, in which the adjustment to the conservation equations for $A$ and $h$ occurs abruptly when and where $A = 1$, as discussed by *Schulkes, 1995*. For $A < 1$, our scheme is equivalent to that of Hibler 1979. For $A \geq 1$, the schemes differ. Our approach is as follow: the same differential equation is still solved for $A$, with a manual adjustment to $A = 1$, and ice thickness is adjusted for the excess ice concentration. This redistribution scheme conserves the ice volume and, compared to the Hibler 1979 scheme, has the advantage of not creating any spurious oscillations in the solution (which happened due to the abrupt change in the differential equations at $A = 1$, see *Schulkes et al., 1995*).

In correcting the paper, we have also addressed some issues with the presentation of the conservation equations for $A$ and $h$ and thickness redistribution scheme in the original version of the paper.

- First, the reference to *Hibler, 1979* for the parameterization of the ice thickness redistribution was a mistake. We drop the reference to *Lietaer et al., 2008* (who seem to have made the same error as we initially did with their redistribution scheme based on $h$). We also drop the reference to *Thompson et al., 1988*, as we suspect their redistribution scheme might also not be coherent.

- As pointed out by reviewer 1, there is a typo in equation (7), also found in the text (p. 8, line 27), and the right form, now corrected, reads:

  $h^+ = \max[0, (A-1)]\, h.$

  This is the formulation used in the code, hence this typo does not impact the results reported here.

- As pointed out by reviewer 3, there was also a typo in the equation appearing on line 18, page 19. The correct expression, implemented in the model, is
  $$\nabla \cdot (h\, u) = u \cdot \nabla\, h + h \nabla \cdot u$$

- The adjustment on the excess ice concentration and associated redistribution of ice thickness (given by equation 7) is made a second numerical step, after solving the conservation equation for the ice thickness. This was discussed on page 13, lines 12 and 13, but this might not have been clear in the original version of the paper because the numerical scheme was not described in details. The inclusion of the appendix now clarifies this point.

The model simulations presented in the newly submitted version of the paper have all been corrected for this error in the thickness redistribution scheme. This correction has no significant effect on the reported results. The simulated ice thickness is somewhat lower than in the previous simulations, as expected from the removal of the extra growing tendency on $h$. However, both the PDF of the mean ice thickness in the idealized and realistic case show a similar shape and evolution, such that the main conclusions drawn form these numerical experiments remain unchanged.

**page 9, l. 4-5, Equation (8) and (9): Just for clarity, a dependence on the thickness as in Hibler (1979) is not needed, as the internal stress is used in the momentum equation?**

Yes, this is right. We now add a mention to this effect when introducing the form of the momentum equation solved in the simulations (A1) in the Appendix. The mechanical parameters ($E$, $\eta$, $C$) are intrinsic properties of the ice cover, as a material, and are independent of its thickness. For instance, $E$ is an elastic modulus ($Nm^{-2}$), not a rigidity of the ice plate ($Nm^{-1}$). In particular, the fact that $C$ is independent of ice thickness is important here as the contribution from thickness and cohesion to the strength of the ice cover are differentiated (in section 5.1.3).

**page 9, l. 8: "widely" -> is widely page 9, l. 22: "(Kwok et al., 2010))" -> (Kwok et al., 2010)**

**page 10, l.11: "northerly, wind stress" -> not sure about this: northerly winds, but the stress is acting towards the south, should be made clearer I think.**

This sentence and the next are rephrased as: "Consistent with observations of orographic channelling, an along-channel, i.e., southward, wind stress, $\tau_{a}$ is applied. The stress is spatially uniform and increased steadily between (...)".

**page 10, l.24: "transport of the cohesion, C" there are no sources and sinks of cohesion? How realistic is that? Please comment and elaborate the cohesion equation in Dansereau et al. (2016), this is not discussed.**

There is no sources or sinks of cohesion in the model. The field of cohesion is set at $t = 0$, i.e., as other initial conditions, and is advected passively with the flow. As discussed on page 12, lines 3 to 12, for ice entering the channel through open boundaries, the cohesion is set over each model element as it is set for the initial conditions, that is, by drawing a value randomly from a given uniform distribution.

As mentioned in our response to your previous comment, the cohesion is an intrinsic property of the material which sets its mechanical strength (its resistance to pure shear). Here $C$ is a grid-cell averaged quantity and is allowed to vary locally to represent the natural homogeneity/heterogeneity of the material (various defects of different scales, for instance brine pockets at the small scale or the presence of different types of ice, e.g, a mixture of first year and older floes, smaller and larger floes, etc., at large scales). A comment to this effect in now added at the end line 24, p. 10. The noise introduced on $C$ could alternatively be applied to another mechanical parameter, for instance, the elastic modulus (*Amitrano et al., 1999 and others*).

As this property is independent of ice thickness, there is no source of $C$ due to ice thinning/thickening. In progressive damage models, cohesion could be made to depend on the level of damage. Simulations have shown that this causes an even more extreme localization of the deformation and damage in a material (*Lucas Girard*, *Ph.D. thesis*). We cannot think of other sources or sinks of cohesion.

The cohesion equation (a transport equation) is now included in the appendix and referred to in the text.

**page 10, l.32: What is the reference for the Young's modulus? Same as for the Poisson's ratio?**

The value of the Young's modulus (0.585 GPa here) is of the same order of that used by *Girard et al., 2011* (0.35 GPa) and implies with an elastic shear wave speed of 500 $ms^{-1}$, consistent with that reported by *Marsan et al., 2011* (440 $ms^{-1}$). These references have been added in the text. Estimates of the Young's modulus are highly variables. The value used here is close to the lower bound of the range of reported value. Using a higher value (2.34 GPa), consistent with a shear wave speed of 1000 $ms^{-1}$ and on the order of in-situ seismic measurements as reported by *Timco and Weeks, 2010* (between 1.7 and 9.1 GPa, with higher values for low brine volumes, i.e., fresher ice) however does not change the mechanical behaviour of the model. This has been verified in the context of the present channel flow simulations. As mentioned in our response to your later comment, a higher value of $E_0$ allows stable ice bridges to form in the channel for somewhat lower values of cohesion than the ones reported here. The exact values of $E_0$ and $C$ to employ in the model at a given spatial resolution are therefore not strictly constrained.

**page 11, l.5-8: The physical role of healing is unclear and needs to be explained better. It is clearly connected to the thermodynamics (in contrast to earlier statements in the manuscript) . . . Please elaborate . . .**

Healing is linked to the level of damage of the ice cover, $d$, which represents the density of cracks/leads within a model grid cell and the impact of these features on the sea ice rheology. In the present model, this variable is independent of the ice concentration, $A$. Healing represents the refreezing within these cracks/leads and allows a damaged ice cover to recover at most its undamaged mechanical strength. As explained in the response to your earlier comment, healing is theoretically not independent from thermodynamics, as the rate of healing should depend on the difference in temperature between the atmosphere above and that of the ocean below. In the present model, healing is not coupled to a thermodynamics component and the healing rate is constant in both space and time.

Because of the absence of thermodynamic-dynamic coupling in the present model, $d$ can increase locally due to healing, but the ice concentration, $A$, is not allowed to re-increase by the same process. Where the ice cover is highly fragmented but dense (high concentration), allowing the ice to heal without re-increasing the ice concentration is physically sound. However, where the ice concentration drops such that mechanical interactions (i.e., the rheology term) becomes insignificant, this absence of thermodynamicdynamic coupling leads to a situation where $d$ can re-increase up to its undamaged value (1), but $A$ can drop to 0, representing open water.

To deal with this unphysical situation, in the present simulations we impose a cutoff on healing when and where $A < 0.75$, which essentially occurs when the ice detaches from a bridge or a coast. As when $A < 0.75$, the rheology term in the momentum equation becomes negligible and the ice is in a free drift state, no matter the value of $d$, we find that imposing this cutoff and its specific value of $A$ has no significant impact on the simulated dynamics.

This point is now clarified on page 11.

**page 12, l.7: "location of ice bridges is not prescribed" only through the random field of cohesion (the spatial pattern should be shown somewhere). I would like to see simulations with uniform cohesion; the model geometry should be irregular enough to make the model develop ice bridges, etc.**

We do not agree with this comment. As explained in the text, the field of cohesion is *random*, hence, by definition, there are no spatial correlations introduced by the field of $C$ in the model. Cohesion therefore does not prescribe the location of ice leads and bridges, only the mechanical behaviour of the model and the domain geometry does. A sentence is added to stress this point in the last paragraph of page 11. Simulations, both idealized and realistic, started from different fields of $C$, set as described on page 11, line 10 to page 12, line 2, have indeed been performed, and have reproduced the same location of the ice bridge.

The disorder introduced in the field of cohesion causes the *progressive failure* of the ice cover, *even under homogeneous forcing conditions* (see *Dansereau et al., 2016*). A sentence is added to clarify this point on page 6, after line 25.

These two points can be demonstrated by comparing the propagation of damage in a simulation in which noise is initially introduced in the field of cohesion (see figure 6) and a simulation started with a uniform field of $C$ (see below). Highly damaged features emerge in both cases in similar locations. In both cases also, ice bridges develop in the same locations, which is therefore not attributable to a pattern in the field of cohesion but to the flow conditions and domain geometry. A notable difference between the simulation is the width of the first damaged features simulated by the model, that is, the features formed in initially undamaged ice (field b, $t = 6$ hrs). In the uniform cohesion case, these features are wider, due to the fact that all model elements can become over-critical and trivially fail, at the same time. This is also visible in the field of ice concentration (c) and translates into higher value of the damage rate (a) compared to the noisy cohesion case (see figure 6a). However, as discussed in *Dansereau et al., 2016* (see section 6.1), as soon as there are some damage present in the ice cover, the heterogeneities introduced in the stress field by these damaged features contribute and, over time, prevail over the noise in $C$ in setting the location and timing of subsequent events. This explains why damage in an non-intact ice cover becomes highly localized even in the uniform cohesion case ($t = 24$ hrs, 48 hrs). In the present simulation, a highly homogeneous wind forcing is used and simulations are started from uniform ice conditions. If simulations were started from realistic, heterogeneous ice conditions, with non-uniform thickness and concentration, and used realistic, time and space-dependant wind forcing, the first damage events would probably be highly localized, independently of the degree of disorder introduced through the cohesion field (eg., *Bouillon and Rampal*, 2015).

[Figure]

[Figure]

*Left panel: noise on the field of cohesion. This field is multiplied by $C_{min}$, such that $C \subset [C_{min}, 2 \times C_{min}]$. Right panel: distribution of the noise on the field of cohesion shown in the right panel.*

[Figure]

*(a) Time series of the wind forcing (dashed curve) and of the damage rate (solid grey curve) over the realistic Nares Strait in a simulation using a uniform field of cohesion, C. Instantaneous fields of the simulated (b) level of damage and (c) ice concentration at t = 6, 24 and 48 hours. This simulations was run for about 50 hours instead of 72 hours as in figure 6 of the paper.*

As the setting of the noise in the field of cohesion in both the idealized and realistic cases is described in the text (page 11, line 10-17), so that the reader can reproduce the results, and as a figure does not provide more information, we do not believe that including a figure of the field of C in the paper is necessary. An example of the random noise on the cohesion field and distribution of this noise is shown above for the realistic case.

**page 12, l.16 (and elsewhere): "(see (Dansereau et al., 2016))" -> (see Dansereau et al., 2016).**

**page 12, l.18: For the claims made in the introduction and background sections, VP simulations at this resolution are absolutely required (have not been done to my knowledge). Please tone done the statements in the appropriate places.**

The EVP simulations of *Dumont et Gratton, 2009* have a resolution of approximately 3 by 4 km, which is comparable to the resolution used here in our realistic experiments (we note that there is an error on the horizontal resolution, i.e., an inversion between the latitude and longitude, reported in their table 1, otherwise their model resolution is something like 17 km by 1 km). The authors state that there

are 14 grid cells across the narrowest point (46 km) of their channel in the idealized experiment which corresponds to the narrowest point between Kane Basin and Smith Sound. In the present realistic and idealized experiments, there are about 19-22 grid cells at the narrowest point between Kane Basin and Smith Sound (56 km on our grid) where the main ice bridge form, which is again comparable. The other EVP simulation of ice bridges in Nares Strait mentioned here (*Rasmussen et al.*, 2010) indeed use a coarser resolution (between 4 km in the Lincoln sea, 83 N, and 10 km in Baffin Bay, < 74 N, and about 7 km between Kane Basin and Smith Sound). However, as mentioned in section 5.2 about the ice thickness distribution in the idealized case, lower resolution (4 km, which gives 13-14 grid cells across the constriction point of the idealized channel, as in *Dumont et al., 2009*, and 8 km, which gives 6-7 grid cells across the constriction point, as in *Losch and Danilov, 2012*) idealized simulations produced similar results. It is also the case for other variables (level of damage, ice concentration, velocity profiles, etc., see figures below) and for the realistic experiments at lower resolution (not shown) which demonstrate that the results obtained here do not depend on the model resolution.

[Figure]

*(a) Time series of the wind forcing (dashed curve) and of the damage rate (solid grey curve) in an idealized channel simulation using $C_{min}$ = 20 kPa. Instantaneous spatial distribution of (b) the level of damage and (c) ice concentration at the times indicated by the numbers 1, 2 and 3 on the time series of panel (a). Instantaneous profiles of the vertical and horizontal velocities at the times indicated by the numbers 1, 2 and 3 on panel (a). The horizontal resolution is of 4 km.*

[Figure]

*(a) Time series of the wind forcing (dashed curve) and of the damage rate (solid grey curve) in an idealized channel simulation using $C_{min}$ = 20 kPa. Instantaneous spatial distribution of (b) the level of damage and (c) ice concentration at the times indicated by the numbers 1, 2 and 3 on the time series of panel (a). Instantaneous profiles of the vertical and horizontal velocities at the times indicated by the numbers 1, 2 and 3 on panel (a). The horizontal resolution is of 8 km.*

Modifications have been made in the text regarding the comparison between the Maxwell-EB and the VP/EVP rheology (see our responses to your earlier comments). We stress the point that this paper was never about making a comparison between the two types of models, *but to demonstrate the capabilities of the Maxwell-EB model.* Moreover, we believe that the capability of a model to represent a given physical phenomenon should *not* depend on model resolution, as long as the resolution is sufficient to resolve the relevant processes.

**page 13, l.1-2: please state the range of x**

**page 13, l.16-17: "(see Dansereau et al. (2016))" -> (see Dansereau et al., 2016)**

**page 13, l.19-20: I think this statement requires, that you have tried a fully implicit scheme and compared the results. Have you? If not, this statement is no really supported by anything and should be changed.**

Yes, we have tried a fully implicit scheme, in which all variables were updated as part of the fixed point iteration. This did not have significant impact on the simulation results both in highly idealized and realistic cases, as mentioned in the text.

**page 14, l.8-9: Please say, how much the "drift velocity on the order of that associated with strictly elastic deformations within an undamaged ice cover." really is (in m/s or cm/s or whatever) so that others can compare.**

This reference has been added ($u$ is on the order of $10^{-5}$ ms$^{-1}$ maximum for strictly elastic deformations).

**page 14, l.11: "relatively undamaged" -> rephrase to "stagnant ice with low damage" or similar**

Ok.

**page 14, l. 20-21: "the width of the distribution of C impacts the rate of propagation of the damage, with the propagation being more progressive for a larger distribution." Since the cohesion appears to be an important parameter, it would be useful to add more information about the choice of C, i.e. the actual distribution of C that is generated (page 11, ll.10) in case the reader would like to reproduce the results.**

Because this comment is not relevant to understand the results presented here, it is now removed. The main point of using different values of cohesion in mentioned in the previous sentence, which is that the minimum value of cohesion over the domain controls the timing of the onset of damaging in the simulations.

Idealized simulations exploring the specific role of disorder (i.e., the width of the distribution of $C$ here) in elasto-brittle models are now being performed, and show that this statement, "the propagation being more progressive for a larger distribution" is not exactly correct. We therefore believe that removing this sentence will avoid any confusion on this point. Besides, channel flow simulations with a uniform cohesion have produced results similar to that reported here (see our response to your earlier comment), demonstrating that the width of the distribution of cohesion is not an important factor in these simulations.

The distributions of $C$ that are generated for these simulation are explained on page 11, line 10-17 (see our response to your earlier comment).

**page 15, l.2: "differs" -> differ**

**page 15, l.23: "(see Fig. 4b and 4c, panel 3)" Should be Fig. 5b and 5c. . .**

**page 15, l.23-26: "This is an important point, as standard viscous-plastic sea ice models do not account for pure uniaxial or biaxial tensile strength and hence would not be able to reproduce the formation of a stable ice arch with self-obstruction to flow under the conditions simulated here." I don't agree: (1) From the figure, the location of the arch is not visible if you mean it is defined by the location of black elements. (2) the de- tails of the yield curve (Figure 2) should not matter, one can tune the elliptic yield curve to resemble the Mohr-Coulomb and tensile failure criteria (see Figure 1 in Lemieux et al, 2016). (3) even without isotropic/biaxial tensile strength, Dumont could simulate arches with VP rheology, so do Losch and Danilov (2012) in similar idealized simulations, even with "a standard VP model" for order 1000 days. (4) why do VP models not account for pure uniaxial tensile strength? I think that this statement needs to change.**

(1) The location of the ice bridge is not defined by the location of black elements. In the text, the location of the ice bridge is associated to *the collocation of a minimum/maximum in the second and first principal stresses*. The location of the ice arch is clearly visible from the profile of ice velocity and (now included) the field of ice concentration. This last point is now mentioned in the text. Also, in the following sentence, there was a mistake : "downstream" should be "upstream".

(2) First, it is important to stress the point that the yield/damage criterion and the rheology (i.e., the constitutive law) are separate components of a mechanical model. The details of the yield curve do matter because to sustain ice bridges, the ice needs to have some cohesive strength (see *Dumont et al., 2009* and *Lemieux et al., 2016*). *Lemieux et al., 2016* refers to the standard elliptical yield curve as accounting uniaxial tensile strength (2$^{nd}$ page, 3$^{rd}$ paragraph). This wording is false. The standard elliptical yield curve accounts for some biaxial tensile-compressive strength (see our response to your earlier comment), uniaxial

*compressive* strength but no uniaxial tensile strength. In this paper, the authors have modified the standard elliptical yield curve to account for uniaxial and biaxial tensile strength for a better representation of landfast ice in VP models, hence implying that the details of the yield curve do matter.

(3)  As mentioned in our response to your earlier comment, the elliptical yield curve used by *Dumont et al., 2009* and *Losch and Danilov, 2012*, does not include biaxial (or uniaxial) tensile strength, but biaxial compressive-tensile strength and uniaxial compressive strength. Therefore ice in these models can not sustain biaxial tensile stresses. Here, as shown by the profile of the principal stress components, the state of stress just upstream of the ice bridge is *biaxial* tensile, which demonstrates that the bridge sustains biaxial tensile stresses. In the paper, we thus make the point that models that do not account for biaxial tensile strength would not be able to reproduce a stable ice bridge in the conditions simulated here, i.e., in which the states of stress are biaxial tensile.

(4)  We do not understand this question fully because of your earlier comment, which states that there is uniaxial tensile strength in the standard elliptical yield curve. There is indeed no uniaxial nor biaxial tensile strength in the standard, Hibler elliptical yield curve. This yield curve was chosen based on the early AIDJEX assumptions that sea ice did not exhibit pure tensile strength (see *Coon et al., 2007*).

We made some adjustment to this paragraph (and figures) to indicate the location of the ice bridge as well as the states of stresses upstream of this bridge more clearly. We also made modifications to section 2.1 and figure 2 to better explain what is cohesion and the difference between uniaxial/biaxial tensile, biaxial tensile-compressive and uniaxial compressive strength. We also modified the statement concerned by this comment as "This is an important point, *as models based on the standard elliptical yield curve* do not account for uniaxial or biaxial tensile strength and hence would not be able to reproduce the formation of a stable ice arch with self-obstruction to flow under the *stress* conditions simulated here" and believe that otherwise it does not need to change.

**page 16, l.24: In this comparison (Figure 8), one might ask why the specific failure curves where chosen differently for the model, when there are estimates for the parameters available ( c = 250, and $\mu$ = 0.9). Should be discussed somewhere.**

This value of $q$ (i.e., $\mu$) and $\sigma_c$ was taken by *Weiss et al., 2007* and *Weiss and Schulson, 2009* to draw the Mohr-Coulomb envelope on this figure because it was the one available value, reported by *Schulson et al. 2006a* for the failure envelope of first-year arctic sea ice obtained from biaxial tests in the laboratory at −10 °C. This is now mentioned in a footnote. In the Maxwell-EB model, we use $\mu$ = 0.7, equivalent to an internal friction angle of 35 degrees, a value commonly used for geomaterials and ice (*Byerlee, 1978* and *Jaeger and Cook, 1979*). A lower value of $q$ could also be deduced from figure 8a. Conversely, using $\mu$ = 0.9 (internal friction coefficient of 42 degrees, not shown) does not impact the behaviour of the Maxwell-EB model.

**page 17, l.3: "later" -> more recent ?**

We changed it for the 1990's, which is the correct period reported by *Barber et al., 2001*.

**page 17, l.16: "According with"-> In line with**

We changed it for "in accordance with", as suggested by reviewer 1.

**page 17, l.21: "'differentiated" does not sound right, rephrase if necessary**

We now use "distinguished".

**page 17, l.29-31: "However, in all of the weaker ice cover scenarios (2002-2008 period and/or summer), none of the ice arches formed near the exit of Kane Basin nor secondary arches formed elsewhere sustain the applied wind forcing and all ice bridges eventually collapse." Is there a similar behaviour in observations in this period? Please add a comment.**

Yes, a similar behaviour was observed over the same time period, as discussed at the beginning of section 5.1.3 (first paragraph). For instance, no ice bridge formed between Kane Basin and Smith Sound in the winters of 2007/2008 to 2009/2010, except for a 2 months period (*Munchow et al., 2016*). We have modified this paragraph to include this and a more recent reference (*Ryan et al., 2017*).

Since we perform simulations with an initially uniform ice thickness and simplified wind forcing, i.e., not representative of specific conditions over the period 1979-2001 or 2002 and 2008, we do not think making a direct comparison to ice conditions in the Strait during that period is relevant at this point in the text.

**page 18, l.7: "widely different dynamical behaviours" -> a wide range of dynamical behaviour**

**page 18, l.7-9: The big question remains: how do you determine the appropriate cohesion? It appears to be vital parameter, similar to P\* in Hibler's VP model.**

Cohesion is indeed an important parameter in the model as it controls the shear strength of the ice and as for $C = 0$, the model would not allow any form of tensile strength. However, we do not believe a direct comparison to $P\*$ is relevant. Indeed, in-situ stress measurements do indicate the importance of the cohesion parameter, by the fact that these measurements fit well a Mohr-Coulomb criterion with non-zero cohesion (see figure 8a). On the contrary, these measurements do not support the role of $P\*$, the biaxial compressive strength, as being a relevant parameter to describe the shape of the damage criterion (or yield criterion in the case of the VP model). The measurements do not give an indication of an appropriate value for this parameter either.

As mentioned on page 11, line 17, some studies (e.g., Schulson, 2004; Weiss et al., 2007) assume a scale effect on shear strength, set by the size of the defects (thermal cracks, brine pockets, …) present in the ice cover. According to this scaling, lower values of $C$ are consistent with larger defect sizes and a lower shear strength. It is difficult to infer a proper spatial scale for the in-situ stress measurements reported here (from Weiss et al., 2007), but it should be smaller than the spatial resolution of the present experiments, hence a lower cohesion should be used in the model.

As mentioned in section 4, the *highest* values of $C$ employed here (i.e., the upper bound of the distribution of C in the case of $C_{min} = 30$ kPa, which is 60 kPa) are consistent with the in-situ stress measurements reported by *Weiss et al., 2007* (see figure 8b). We obtained the formation of stable ice bridges in the model for lower values of $C$.

However, the fact that we obtained the formation of a stable ice bridge in the present idealized and realistic simulations using $C_{min} = 20$ kPa does not mean this is the appropriate value of cohesion for sea ice or for the Maxwell-EB model, nor that it is the only value for with the Maxwell-EB model can reproduce the formation of a stable ice bridge between Kane Basin and Smith Sound. As mentioned in lines 7 to 9, this result depends on

- the prescribed initial thickness. Bridges form at lower cohesion for thicker ice. Here we used $h_0 = 1$ m but a higher $h_0$ might be more representative of ice conditions for some years.

- on the specific value used for the Young's modulus. A higher value allows the formation of stable ice bridges for lower values of cohesion. As mentioned in the response to your earlier comment, the value used for $E_0$ is at the lowest bound of the range of reported values.

- the magnitude of the applied wind forcing. In the model simulations, we increase the wind forcing up to 1 Nm$^{-2}$ and hold it constant, which corresponds to a wind speed of 82 km h$^{-1}$ or 22 ms$^{-1}$. While daily-averaged model wind stress values of 0.7–1.0 N m$^{-2}$ have been reported in Nares Strait, see *Samelson et al., 2006*, a uniform, sustained wind stress of 1 Nm$^{-2}$ for *several days* is most probably an overestimation of the reality. Were we made this choice of wind forcing to simplify the analysis.

Therefore, If we were to increase $h_0$, increase $E_0$ and decrease the applied wind forcing, stable ice bridge would be obtained for lower values of $C$, and conversely for a lower $h_0$ and $E_0$ and higher wind forcing. In the passage you are reporting, we therefore made it clear that the goal of these experiments was not to determine an appropriate range of value for the cohesion.

**page 19, l.5-6: "A Lagrangian model would perhaps be more suitable to simulate the edge of the detached ice"; or a better advection scheme with less numerical diffusion (i.e. higher order basis functions in your finite element method)**

The diffusivity of the numerical scheme and order of the polynomial approximations used are described in the sentences above, from p. 18, line 31, to p. 19, line 3. The sentence you are referring to does not refer to diffusion, but to the fact that Lagrangian approaches, i.e., which follow ice particles, are better suited to track the ice edge. The use of higher polynomial approximation does not *change* the numerical scheme.

Also, we have replaced "more suitable" by "a more natural approach".

**page 19, l.13-14: "Nevertheless, at all times the simulated probability density function is strongly asymmetric, consistent with thickness distributions estimated for sea ice with little history of melting (e.g., Haas, 2009)." Please discuss in how far this special experimental geometry with many coastlines and the low Cmin is suitable to compare to observations made for open ocean Arctic sea ice as described in Haas (2009).**

Here, we referred to measurements from the open Arctic ocean with little history of melting specifically because the model does not represent thermodynamic effects and hence the simulated ice thickness distribution and hence a comparison with measurements from a region where the melting signal is important should not be made. Asymmetric thickness distribution have not been obtained from open Arctic ocean measurements only. For instance, *Hass et al., 2006* report an asymmetric thickness distribution with an exponential tail from AEM measurements at the entrance of Nares Strait. We now include this reference in the text.

Concerning the value of cohesion, a higher value (e.g., $C_{min} = 20$ kPa) also give a strongly asymmetric thickness distribution, however, it does not allow the thickness to increase to values as high as in the $C_{min} = 10$ kPa case in the same simulation time, only because ice bridges form and stop the flow of ice through the channel, hence reducing the amount of ice entering the channel that can be

incorporated into ice ridges. This point is now clarified in this section. Moreover, as discussed in the response to your earlier comment, there is no observational nor physical evidences at this point to characterize $C_{min}$ = 10 kPa as a "low" or "too low" cohesion for the ice cover.

**page 19, l.18: This term (7) is not correct and should not be used. See e.g. Schulkes (1995), JGR, for correct equations and a nice explanation of ridging in general.**

This is a typo in the text on the development of the term $\nabla \cdot (h\,u)$ (see response to reviewer 3 and to your earlier comment). This was not an error in the code.

**page 19, l. 23: "Fig. 11b" -> Fig. 10b 10b**

**page 20, l. 2-3: "In coupled thermodynamic and dynamic models, a high density of leads is expected to impact the simulated heat fluxes between the atmosphere, the ice and the ocean (Smith et al., 1990)." This is not really a conclusion, but part of a discussion.**

We agree and move this comment to the discussion part of this section (page 20, end of second paragraph).

**page 20, l. 11-13: "the presence of land fast ice along. . ." This has hardly been discussed and comes as a surprise. Needs more attention in Section 5 if you want to keep this conclusion**

We do not agree with this comment, as the presence of landfast ice is discussed in section 5.1.2 and 5.1.2, along with other features reproduced by the model. This remains in the list of conclusions. We have added additional references on the observed presence of landfast ice in Nares Strait.

**page 20, l. 24: "a process that is known to be underestimated in VP models using a two-level scheme" This is new to me. At correspondingly high resolution I would expect a VP model to behave in a similar manner, see also Losch and Danilov (2012), Fig6. which shows very similar ice thickness distribution in a similar channel experiment.**

The statement made here compares the thickening of the ice cover between a VP model with a two-level versus a multi-categogies thickness redistribution scheme. It is our understanding that in *Losch and Danilov, 2012*, a two-level categories scheme was used as was not compared to a multi-categories scheme. An ice thickness distribution was not computed in this study. The results reported represent a steady state after 10 years of integration and hence would not be directly comparable with the present Maxwell-EB simulations.

This sentence was moved to the discussion of the two-level and multi-categories scheme, section 2.2.

**page 20, l.26-28: See above, I don't think, that you can say this, because you'd have to show that the same model configuration with a VP model would not have your thickness distribution. I am pretty sure that you would get a similar result.**

The sentences you are referring to is:

"In the Maxwell-EB model, this capability of accounting for a sufficient thickening of the ice as well as the spatial localization of extreme thickness values arises from the appropriate description of extreme strain localization. On a mechanical point of view, this may therefore question the relevance of using multi-categories redistribution schemes."

The sentences therefore discusses the capability of the Maxwell-EB model, not the VP model, to represent the localization of increased ice thickness, in relation with the localization of ice deformation. The next sentence questions the use of a multi-categories thickness model versus a simpler thickness redistribution model to obtain this localization of high thickness values. As mentioned an earlier comment, the thickness redistribution scheme is independent of the rheology used and here, the VP model is not mentioned. Therefore this does not prevents us from writing this sentence.

**page 21, l. 14: "later" -> recent**

**page 21, l. 33: "Haas, C.: Dynamics Versus Thermodynamics: The Sea Ice Thickness Distribution, p. 638, Wiley-Blackwell, 2009." Please correct citation as book chapter in Sea Ice (eds D. N. Thomas and G. S. Dieckmann)**

**page 22, l.10: "III, W. D. H.: Modeling a Variable Thickness Sea Ice Cover,. . ." -> wrong name**

**page 25, l.27: "Weiss, J. and Dansereau, V.: Linking scales in sea ice mechanics, Philosophical Transactions A, pp. –, doi:10.1098/XXXX, 2016." Is this a submitted manuscript? If so it is not properly cited.**

**page 26, Figure 5: What is Cmin in this simulations? Did you consider to show a sea ice concentration plot for the idealized experiments as well? That would help to see the arches directly.**

$C_{min}$ = 20 kPa (it is the same simulation as in Figure 4, as mentioned in the caption). This is now also stated at the beginning of section 5.1.1. The corresponding fields of ice concentration are now added to this figure.

**page 29, Figure 8: An indication of the probability of single stress states using a colormap or transparency would be helpful, to get an impression how frequently biaxial tensile states (and all other stress states) occur.**

To indicate the proportion of each types of stress through time, a time series of stress state types (tensile, biaxial tensile-compressive, biaxial compressive) during the corresponding simulation is now included in Figure 8. Figure 8b (now 8c) corresponds to a snapshot at $t$ = 72 hours, when the probability of each stress states and repartition in the principal stresses plane has stabilized. This point is now clarified in the text.

**page 31, Figure 10: Why are the PDFs for x = 4km and 8km given at t=5days, whereas the other results are shown for t=3days?**

Thank you for catching this. This is a typo from an earlier simulation. The PDFs for $x$ = 4 km and 8 km are indeed given for $t$ = 3 days. This has been corrected in the figure caption.

---

## Author Response (AR2)

**Response to Reviewer 2**

**- I would appreciate if the authors could provide the random field used for the computation of the initial condition for cohesion as supporting information along with the manuscript. If published, all information of initial conditions would be provided to repeat the experiments in a different framework and use this configuration as an benchmark experiment.**

**- page 4, l.4: "below its original value (2, Hibler, 1979). to increase the shear and uniaxial compressive strength " please remove the period.**

**- page 27, Fig. 7: as there are no subfigures, "(a)" in the beginning of the caption is not needed**

Thank you very much for reading the paper again and for your additional comments. We corrected the two mistakes you mentioned (as well as a few other typos we found along the way).

Regarding your request for the initial fields of cohesion used, we would indeed encourage using the idealized and realistic Nares Strait configurations as benchmark experiments. However, we do not believe that publishing these two fields is relevant in the context of providing the initial conditions that are essential to repeat the experiments performed here. As explained and demonstrated in our response to your earlier comments, the noise introduced in the field of $C$ does not prescribe the location of the simulated leads and bridges. The results described here do not depend on the exact spatial distribution of cohesion : simulations using the same value of $C_{min}$ but different random spatial distributions of the disorder on $C$ were compared and produced results that were very similar, sometimes even undistinguishable, in all of the aspects discussed in the paper. We now add the following sentence to further stress this point in the first paragraph of page 12, which describes the field of cohesion:
"Model simulations using the same value of $C_{min}$ but different random spatial distributions of the disorder on $C$ produced similar results, in all aspects comparable to those discussed below."

The important point we therefore aim to stress is: to repeat the experiments presented here, it would be sufficient to generate the noise in the field of $C$ following the *same method* we employed. This method is described in the paper (page 12, first paragraph) and consists in (1) drawing randomly a value over each model element from a uniform distribution of values comprised between 1 and 2 and (2) multiplying the resulting random field by the desired value of $C_{min}$.

In other words, we believe that providing the two fields used to initialize the idealized and realistic simulations presented in the paper would convey the idea that using these exact fields is necessary to obtain the results described in the paper, which is not true.

The issue with providing these fields is also somewhat technical. On the one hand, possessing the exact fields of $C$ is not sufficient. One would also need to work with the exact same finite element grid employed here, as interpolation would necessary alter these fields. On the other hand, providing the grid and code for the generation of these fields would allow reproducing fields of cohesion with the right characteristics but not the exact same fields used here, as the code is simply based on a random number generator that is seeded differently every time it is called. However, a field with the same characteristics as used here could be very easily generated for a given finite difference or finite element grid in most programming languages following the method described on on page 12, first paragraph.